# Riverine impact on future projections of marine primary production and carbon uptake

Shuang Gao[1,2], Jörg Schwinger[3], Jerry Tjiputra[3], Ingo Bethke[1], Jens Hartmann[4], Emilio Mayorga[5], Christoph Heinze[1]

[1]Geophysical Institute, University of Bergen, Bjerknes Centre for Climate Research, Bergen, Norway
[2]Institute of Marine Research, Bergen, Norway
[3]NORCE Norwegian Research Centre, Bjerknes Centre for Climate Research, Bergen, Norway
[4]Institute of Geology, Center for Earth System Research and Sustainability (CEN), Universität Hamburg, Hamburg, Germany
[5]Applied Physics Laboratory, University of Washington, Seattle, WA, USA

*Correspondence to*: Shuang Gao (shuang.gao@hi.no)

**Abstract.** Riverine transport of nutrients and carbon from inland waters to the coastal and finally the open ocean alters marine primary production (PP) and carbon (C) uptake regionally and globally. So far, this process has not been fully represented and evaluated in the state-of-the-art Earth system models. Here we assess changes in marine PP and C uptake projected under the Representative Concentration Pathway 4.5 climate scenario using the Norwegian Earth system model, with four riverine transport configurations for nutrients (nitrogen, phosphorus, silicon and iron), carbon and total alkalinity: deactivated, fixed at a recent-past level, coupled to simulated freshwater runoff, and following four plausible future scenarios. The inclusion of riverine nutrients and carbon at 1970's level improves the simulated contemporary spatial distribution of annual mean PP and air-sea $CO_2$ fluxes relative to observations, especially on the continental margins (5.4% reduction in root mean square error [RMSE] for PP) and in the North Atlantic region (7.4% reduction in RMSE for C uptake). While the riverine nutrients and C input is kept constant, its impact on projected PP and C uptake expresses differently in future period from the historical period. Riverine nutrient inputs lessen nutrient limitation under future warmer conditions as stratification increases, and thus lessen the projected decline in PP by up to $0.66 \pm 0.02$ Pg C yr$^{-1}$ (29.5%) globally, when comparing 1950–1999 with 2050–2099 period. The riverine impact on projected C uptake depends on the balance between the net effect of riverine nutrient induced C uptake and riverine C induced $CO_2$ outgassing. In the two idealized riverine configurations the riverine inputs result in a weak net C sink of $0.03$–$0.04 \pm 0.01$ Pg C yr$^{-1}$, while in the more plausible riverine configurations the riverine inputs cause a net C source of $0.11 \pm 0.03$ Pg C yr$^{-1}$. It implies that the effect of increased riverine C may be larger than the effect of nutrient inputs in the future on the projections of ocean C uptake, while in historical period increased nutrient inputs are considered as the largest driver. The results are subject to model limitations related to resolution and process representations that potentially cause underestimation of impacts. High-resolution global or regional models with an adequate representation of physical and biogeochemical shelf processes should be used to assess the impact of future riverine scenarios more accurately.

## 1 Introduction

At global scale, the major sources of both dissolved and particulate materials to the oceans are river runoff, atmospheric deposition and hydrothermal inputs; of these three, river runoff plays an essential role in transporting nutrients into the ocean which stimulate biological primary production (PP) in the ocean (Meybeck, 1982; Smith et al., 2003; Chester, 2012). For some substances riverine transport even acts as the absolutely dominant source, such as total phosphorus (~90%) and total silicon (>70%) (Chester, 2012). River transport of carbon into the ocean influences the air-sea $CO_2$ exchange, local oxygen balance and acidification level, thus further affecting marine ecosystem health (Meybeck and Vörösmarty, 1999; Liu et al., 2021). Despite our limited understanding on the riverine carbon fluxes, they could play an important role in closing the global carbon budget (Friedlingstein et al., 2021). A recent study on global carbon cycle has emphasized the importance of the carbon transport through the land-to-ocean aquatic continuum (Regnier et al., 2022).

With an increasing world population and a perturbed hydrological cycle under climate change, riverine transport of nutrients and carbon from land to oceans has a potentially growing impact on the marine biogeochemistry and ecosystem (Seitzinger et al., 2010; van der Struijk and Kroeze, 2010). Furthermore, the impacts of anthropogenic activity, particularly agriculture (Bouwman et al., 2009; Garnier et al., 2021), wastewater discharges (Van Drecht et al., 2009) and extensive damming (Eiriksdottir et al., 2016; Zhang et al., 2022), have greatly perturbed the riverine transport of nitrogen (N), phosphorus (P) and silicon (Si) to the oceans. Seitzinger et al. (2010) estimated that there was an increase in global riverine fluxes of dissolved inorganic nitrogen (DIN) and phosphorus (DIP) by 35% and 29%, respectively, between 1970 and 2000, and a further possible change of -2% to +29% in DIN and +37% to +57% in DIP between 2000 and 2050, depending on the future scenarios used in their study. Beusen et al. (2016) estimated that river nutrient transport to the ocean increased from 19 to 37 Tg N $yr^{-1}$ and from 2 to 4 Tg P $yr^{-1}$ over the 20$^{th}$ century, taking into account of both increased nutrient input to rivers and intensified retention/removal of nutrients in freshwater systems. The riverine carbon input is highly influenced by the magnitude of continental runoff (Liu et al., 2020; Frigstad et al., 2020), permafrost melting and leaching of post-glacial peat deposits (Wild et al., 2019; Pokrovsky et al., 2020; Mann et al., 2022), all of which are sensitive to climate change. In addition, anthropogenic change, such as land-use and land-cover changes, lake and reservoir eutrophication and sewage emissions of organic material into rivers may become an important factor in the future (Meybeck and Vörösmarty, 1999).

Some regions such as the Arctic Ocean and large river estuaries may receive a higher impact from changes in riverine inputs than other regions. The Arctic Ocean accounts for only 4% of the global ocean area (Jakobsson, 2002), but takes 11% of the global river discharge (McClelland et al., 2012), and it is estimated that about one third of its net PP is sustained by nutrients originated from rivers and coastal erosion (Terhaar et al., 2021). Therefore, one can expect that Arctic PP will be affected by altered riverine transport of nutrients and carbon under future climate changes. Previous studies have shown that enhanced riverine nutrient input increases PP in the Arctic Ocean (Letscher et al., 2013; Le Fouest et al., 2013, 2015, 2018; Terhaar et al., 2019), while large riverine dissolved organic carbon (DOC) delivery reduces $CO_2$ uptake in Siberian shelf seas (Anderson et al., 2009; Manizza et al., 2011). Considering large river estuaries, van der Struijk and Kroeze (2010) have demonstrated potentially higher eutrophication or hypoxia risk in the coastal waters of South America by 2050,

where increasing trends in DIN and DIP are detected. Yan et al. (2010) have reported that anthropogenically enhanced N inputs will continue to dominate river DIN yields in the future and impose a challenge of N eutrophication in Changjiang river basin.

The latest generation of Earth system models (ESMs) have implemented some forms of riverine inputs in their ocean biogeochemistry modules (Séférian et al., 2020). The models that include riverine inputs use different implementations, from constant contemporary fluxes (e.g., IPSL-SM6A-LR and NorESM2; Aumont et al., 2015; Tjiputra et al., 2020), to temporally varying fluxes (CESM2; Danabasoglu et al., 2020), and to interactive with terrestrial nutrient leaching transported by dynamical river routing (e.g., CNRM-ESM2-1 and MIROC-ES2L; Séférian et al., 2019; Hajima et al., 2020), and they typically use the Redfield ratio to convert from one chemical compound to the others. For instance, in the latest version of IPSL model (IPSL-SM6A-LR; Aumont et al., 2015) riverine nutrients (DIN, DIP, Si), dissolved organic nitrogen (DON), dissolved organic phosphorus (DOP), dissolved inorganic carbon (DIC) and total alkalinity (TA) are implemented as constant contemporary fluxes based on data sets from Global NEWS 2 (NEWS 2; Mayorga et al., 2010) and the Global Erosion Model of Ludwig et al. (1996). Further, in the CESM2 (Danabasoglu et al., 2020) DIN and DIP are taken from the Integrated Model to Assess the Global Environment-Global Nutrient Model (IMAGE-GNM; Beusen et al., 2015, 2016) and vary from 1900 to 2005, which is more sophisticated than using constant fluxes. The other riverine nutrients, DIC and TA are held constant using data from NEWS 2 (Mayorga et al., 2010). Some ESMs have implemented interactive riverine nutrients input from terrestrial processes, e.g., in the CNRM-ESM2-1 the riverine DOC is calculated actively from litter and soil carbon leaching in the land model, and the supply of the other nutrients, DIC and TA have been parameterized using the global average ratios to DOC from Mayorga et al. (2010) and Ludwig et al. (1996). In MIROC-ES2L model (Hajima et al., 2020), N cycle is coupled between the ocean and land ecosystems, therefore, the inorganic N leached from the soil is transported by rivers and subsequently as an input to the ocean ecosystem. The riverine P is calculated from N using the Redfield ratio, but riverine carbon input is not implemented. Existing models with interactive riverine inputs typically do not consider biogeochemical processes in the freshwater system such as sedimentation.

A few modelling studies have assessed the impact of riverine nutrients and carbon on marine biogeochemistry. For example, Bernard et al. (2011) and Aumont et al. (2001) evaluated riverine impact on marine Si and carbon cycle, respectively. Lacroix et al. (2020) estimated and implemented pre-industrial riverine loads of nutrients and carbon in a global ocean biogeochemistry model, and concluded that the riverine (mainly inorganic and organic) carbon inputs lead to a net global oceanic $CO_2$ outgassing of 231 Tg C yr$^{-1}$ and an opposing response of an uptake of 80 Tg C yr$^{-1}$ due to riverine nutrient inputs. Additionally, the riverine inputs at pre-industrial level lead to a strong PP increase in some regions, e.g., +377%, +166% and +71% in Bay of Bengal, tropical west Atlantic and the East China Sea, respectively (Lacroix et al., 2020). Tivig et al. (2021), on the other hand, found that riverine N supply alone has limited impact on global marine PP (<+2%) due to the negative feedback of reduced $N_2$ fixation and increased denitrification. This negative feedback could also overcompensate the N addition by river supply locally, e.g., in Bay of Bengal where PP decreased due to riverine N input (Tivig et al., 2021). A couple of modelling studies have also assessed the impact of changing riverine inputs on marine PP and $CO_2$ fluxes. Cotrim da Cunha et al. (2007) assessed riverine impact, using a coarse resolution ocean biogeochemistry model,

with single or combined nutrients from zero input to a high input corresponding to a world population of 12 billion people, and reported changes in PP from −5% to +5% for the open ocean, and from −16% to +5% for the coastal ocean, compared to the present-day simulation. Liu et al. (2021) demonstrated an increase in global coastal net PP of +4.6% response to a half-century (1961–2010) increase in river N loads. In a recent study by Lacroix et al. (2021a) the impact of changing riverine N and P in a historical period (1905–2010) on marine net PP and air-sea $CO_2$ fluxes was investigated by applying an eddy-permitting fine resolution (~0.4˚) ocean biogeochemistry model. Their result revealed an enhancement of 2.15 Pg C $yr^{-1}$ of the global marine PP, corresponding to a relative increase of +5% over the studied period, induced by increased terrigenous nutrient inputs. The PP increase in coastal ocean averaged to 14% with regional increase exceeding 100% and the global coastal ocean $CO_2$ uptake increased by 0.02 Pg C $yr^{-1}$ due to the increased riverine nutrient inputs (Lacroix et al., 2021a). In the Arctic, doubling riverine nutrient delivery increased PP by 11% on average and by up to 35% locally, while the riverine DOC input induced $CO_2$ outgassing resulted in 25% reduction in C uptake in the Arctic Ocean (Terhaar et al., 2019).

Although the historical and contemporary impacts of riverine nutrients and carbon have been considered increasingly, their impacts on future projections of marine biogeochemistry have not been sufficiently addressed. Taking advantage of the latest improvement of global river nutrient/carbon export datasets, e.g., NEWS 2 (https://marine.rutgers.edu/globalnews/datasets.htm) and GLORICH (https://doi.pangaea.de/10.1594/PANGAEA.902360), and responding to the demand of development of ESMs with increasing model resolution, the assessment of the impact of riverine nutrients and carbon on projections of marine biogeochemistry becomes feasible and desired.

In this study, we aim to assess the impact of riverine nutrients and carbon on the projected changes in regional and global marine PP and air-sea $CO_2$ exchange by addressing the following questions:

1) How does the presence of riverine fluxes of nutrient and carbon affect the contemporary representation of marine PP and C uptake in our model?
2) How does the presence of riverine fluxes of nutrient and carbon affect the projections of marine PP and C uptake?
3) How important is the consideration of transient changes in riverine fluxes of nutrient and carbon on the projections?

We explore these questions by performing a series of transient historical and 21st century climate simulations under the RCP 4.5 (middle-of-the-road) scenario with the fully coupled Norwegian Earth system model (NorESM) under four different riverine input configurations. Another objective of the study is to explore the best practical way of implementing riverine inputs into future versions of NorESM. Because of the coarse resolution of the version used here, a series of processes in the coastal zone cannot be represented in our study such as the high accumulation of organic sediment in shallow waters and respective remineralization rates of previously deposited material (Arndt et al., 2013; Regnier et al., 2013). These processes can only be presented in models of much higher spatial resolution, which are at present too costly to be integrated long enough to simulate the large-scale water masses adequately and project long-term scale climatic change. Given missing contributions from unresolved processes, our results are to be interpreted as lower bound estimates.

## 2 Methods

### 2.1 Model description

All simulations in this study have been performed with the Norwegian Earth System Model version 1 (NorESM1-ME, hereafter NorESM) (Bentsen et al., 2013), a climate model that provided input to the Fifth Coupled Model Intercomparison Project (CMIP5) (Taylor et al., 2011). The model is based on the Community Earth System Model version 1 (CESM1) (Hurrell et al., 2013). The atmospheric, land and sea ice components are the Community Atmosphere Model (CAM4) (Neale et al., 2013), the Community Land Model (CLM4) (Oleson et al., 2010; Lawrence et al., 2011) and the Los Alamos National Laboratory sea ice model (CICE4) (Holland et al., 2011), respectively. An interactive aerosol-cloud-chemistry module has been added to the atmospheric component (Kirkevåg et al., 2013). The physical ocean component—the Bergen Layered Ocean Model (BLOM, formerly called NorESM-O) (Bentsen et al., 2013)—is an updated version of the Miami Isopycnic Coordinate Ocean Model (MICOM) (Bleck and Smith, 1990; Bleck et al., 1992) and features a stack of 51 isopycnic layers (potential densities ranging from 1028.2 to 1037.8 kg m$^{-3}$ referenced to 2000 dbar) with a two-layer bulk mixed layer on top. The depth of the bulk mixed layer varies in time and the thickness of the topmost layer is limited to 10 m in order to allow for a faster air-sea flux exchange. The ocean and sea ice components are implemented on a dipolar curvilinear horizontal grid with a 1° nominal resolution that is enhanced at the Equator and towards the poles, and its northern grid pole singularity is rotated over Greenland. The atmosphere and land components are configured on a regular 1.9° x 2.5° horizontal grid.

The ocean biogeochemistry component of NorESM is based on the Hamburg Ocean Carbon Cycle Model (HAMOCC5) (Maier-Reimer et al., 2005). The component has been tightly coupled to NorESM-O such that both components share the same horizontal grid as well as vertical layers and that all tracers are transported by the physical component at model time step (Assmann et al., 2010). Tuning choices and further improvements to the biogeochemistry component are detailed in Tjiputra et al. (2013). Here we only summarise features of particular importance to this study and refer to the HAMOCC version used here as HAMOCC$_{NorESM1}$. The partial pressure of $CO_2$ ($pCO_2$) in seawater is calculated as a function of surface temperature, salinity, pressure, dissolved inorganic carbon (DIC) and total alkalinity (TA). Dissolved iron is released to the surface ocean with a constant fraction (3.5%) of the climatology monthly aerial dust deposition (Mahowald et al., 2005), but only 1% of this is assumed to be bio-available. Nitrogen fixation by cyanobacteria occurs when nitrate in the surface water is depleted relative to phosphate according to the Redfield ratio (Redfield et al., 1934). Phytoplankton growth in the model depends on temperature, availability of light and on the most limiting nutrient among phosphate, nitrate and iron. Constant stoichiometric ratios for the biological fixation of C, N, P and $\Delta O_2$ (122 : 16 : 1 : -172) are prescribed in HAMOCC$_{NorESM1}$, and are extended by fixed Si : P (25 : 1) and Fe : P (3.66 x 10$^{-4}$ : 1) stoichiometric ratios. HAMOCC$_{NorESM1}$ prognostically simulates export production of particulate organic carbon (POC). It is assumed that a fraction of POC production is associated with diatom silica production, and the remaining fraction is associated with calcium carbonate production by coccolithophorides. The fraction of diatom-associated production is calculated from silicate availability, effectively assuming that diatoms are able to out-compete other phytoplankton growth under favorable (high surface silicate concentration) growth conditions. Particles, including

POC, biogenic silica, calcium carbonate and dust are advected by ocean circulation in the model. Those particles sink through the water column with constant sinking speeds and are remineralized at constant rates. HAMOCC$_{NorESM1}$ includes an interactive sediment module with 12 biogeochemically active vertical layers. Permanent burial of particles out of the deepest sediment layer represents a net loss of POC, calcium carbonate and silica from the ocean/sediment system and is compensated by atmospheric and riverine inputs on a time scale of several thousand model-years. More detailed model description and parameters are documented in previous publications (Bentsen et al., 2013; Tjiputra et al., 2013).

## 2.2 Model evaluation

The overall performance of the physical and biogeochemistry ocean components has been evaluated elsewhere (Bentsen et al., 2013; Tjiputra et al., 2013). For example, simulated alkalinity, phosphate, nitrate and silicic acid have been evaluated in previous works (Tjiputra et al., 2013; Tjiputra et al., 2020). Here we only briefly review the model performance of the mostly relevant variables for this study, namely PP and air-sea $CO_2$ fluxes.

The simulated global annual mean PP is 40.1 Pg C yr$^{-1}$ during 2003–2012, which is lower than the satellite-based model estimates, ranging from 55 to 61 Pg C yr$^{-1}$ (Behrenfeld and Falkowski, 1997; Westberry et al., 2008). However, the distribution of annual mean surface PP is generally consistent with the remote sensing-based estimates from Behrenfeld and Falkowski (1997), with the largest model-data deviation in the eastern equatorial Pacific and parts of the Southern Ocean (known as High-Nutrient-Low-Chlorophyll regions), where the model overestimates PP (the Arctic Ocean was not assessed in that study; Tjiputra et al., 2013). Along the continental margins, the simulated PP is generally underestimated compared to the remote sensing-based estimates (Tjiputra et al., 2013), which may relate to the lack of riverine inputs and/or unresolved shelf processes due to coarse model resolution. Additionally, our model simulates a comparable magnitude of projected decrease in PP, by the end of the 21$^{th}$ century compared to historical period, with other global models (see detailed discussion in section 4.1).

In the Arctic Ocean, the simulated PP in our model is biased towards lower values. In the study by Skogen et al. (2018), the NorESM model is compared with a regional model that comprises part of the Arctic region, and it shows that the NorESM simulates too late and too short bloom period than the regional model, hence the annual integrated PP is too low. In a multi-model study (Lee et al., 2016) that assesses the relative skills of 21 regional and global biogeochemical models in reproducing the observed contemporary Arctic PP, the NorESM is shown to have a negative bias of -0.49, but is well within the multi-model mean bias of -0.31±0.39. Many coarse/intermediate resolution global models also show considerably lower net PP in the Arctic (Terhaar et al., 2019). Such common shortcomings in global scale marine biogeochemical models can partly be attributed by the simplified, not regionally adapted ecosystem parameterization, which can be improved through data assimilation (Tjiputra et al., 2007; Gharamti et al., 2017). Additionally, lack of adequate representation of riverine input in some ESMs can also lead to underestimate of PP, since around one third of current Arctic marine PP is sustained by terrigenous nutrient input (Terhaar el al., 2021). Despite the biased low PP under the contemporary climate, the projected absolute change of 70 Tg C yr$^{-1}$ by the end of the 21$^{th}$ century is well within the range estimated from other ESMs (Vancoppenolle et al., 2013).

Tjiputra et al. (2013) also evaluated the simulated mean annual sea-air $CO_2$ fluxes for the 1996–2005 period against observational-based estimates by Takahashi et al. (2009) and concluded that the model broadly agrees with the observations in term of spatial variation, although in the equatorial Indian Ocean and in the polar Southern Ocean (South of 60° S) the model underestimates outgassing and overestimates C uptake, respectively.

**2.3 Riverine data**

The influx of carbon and nutrients from over 6000 rivers to the coastal oceans has been implemented in HAMOCC$_{NorESM1}$ based on previous work of Bernard et al. (2011) but with modifications that are outlined in the following paragraphs.

The riverine influx includes carbon, nitrogen and phosphorus, each in dissolved inorganic, dissolved organic, and particulate forms, as well as TA, dissolved silicon and iron (Fe). Except for DIC, TA and Fe, all data are provided by the NEWS 2 model (Mayorga et al., 2010), which is a hybrid of empirical, statistical and mechanistic model components that simulate steady-state annual riverine fluxes as a function of natural processes and anthropogenic influences. The NEWS 2 data product contains historical (year 1970 and 2000) and future (year 2030 and 2050) estimates of riverine fluxes of carbon and nutrients. The future products are developed based on four Millennium Ecosystem Assessment scenarios (Alcamo et al., 2006): Global Orchestration (GNg), Order from Strength (GNo), Technogarden (GNt) and Adapting Mosaic (GNa). These scenarios represent different focuses of future society on e.g., globalization or regionalization, reactive or proactive environmental management and their respective influences on efficiency of nutrient use in agriculture, nutrient release from sewage, total crop and livestock production along with others (see Table 1 for a brief summary; Seitzinger et al., 2010). The NEWS 2 riverine dataset has been calibrated and assessed against measured yields (Mayorga et al., 2010) and has been widely used and evaluated for different river estuaries (van der Struijk and Kroeze, 2010; Terhaar et al., 2019; Tivig et al., 2021). For example, van der Struijk and Kroeze (2010) compared the NEWS 2 nutrient yields to observed values for South American rivers and indicated that the NEWS 2 models in general perform reasonably well for South American rivers with the variations in yields among rivers described well, although the model performs better for some rivers such as the Amazon than for others. We have compared DIN and dissolved organic nitrogen (DON) from NEWS 2 with measured data from PARTNERS Project (Holmes et al., 2012) for the six largest Arctic rivers around year 2000 (Table C1). The NEWS 2 dataset compares fairly well with the measured data, especially for the Eurasian Arctic rivers with 3.5-28.6% deviation in DIN and 7.3-34.8% in DON, while the discrepancy is larger in the Canadian-Alaska Arctic rivers (i.e., Yukon and Mackenzie rivers) with up to 80.8% and 100% deviation in DIN and DON, respectively.

The DIC and TA fluxes, provided by Hartmann (2009), are produced from a high-resolution model for global $CO_2$ consumption by chemical weathering and are aggregated within catchment basins defined by the NEWS 2 study for each river. Riverine Fe flux is calculated as a proportion of a global total input of 1.45 Tg yr$^{-1}$ (Chester, 1990), weighted by water runoff of each river. Only 1% of the riverine Fe is added to the oceanic dissolved Fe, under the assumption that upto 99% of the fluvial gross dissolved Fe is removed during estuarine mixing (Boyle et al., 1977; Figuères et al., 1978; Sholkovitz and Copland, 1981; Shiller and Boyle, 1991).

At the river mouths, all fluxes are interpolated to the ocean grid in the same way as the freshwater runoff, which
is distributed as a function of river mouth distance with an e-folding length scale of 1000 km and cutoff of 300
km.
In HAMOCC$_{NorESM1}$, there is one dissolved organic pool (DOM) and one particulate organic pool (DET, detritus).
First, we calculate the riverine organic P-N-C ratios for both dissolved and particulate forms, then add the least
abundant species (scaled by the Redfield ratio) to the DOM and DET pools, respectively (see equations below).
$$DOM_{riv} = \min\left(DOP, \frac{DON}{16}, \frac{DOC}{122}\right) \hspace{3cm} (1)$$
$$DET_{riv} = \min\left(POP, \frac{PON}{16}, \frac{POC}{122}\right) \hspace{3cm} (2)$$
POP and PON denote particulate organic phosphorus and particulate organic nitrogen, respectively. The excess
budget from the remaining two species both in dissolved and in particulate forms are assumed to be directly
remineralized into inorganic form and added to the corresponding dissolved inorganic pools (i.e., DIP, DIN, and
DIC) in the ocean.
**2.4 Experimental design**
The fully coupled NorESM model is spun up for 900 years with external forcings fixed at preindustrial year-1850
levels prior to our experiments (Tjiputra et al., 2013). The atmospheric $CO_2$ mixing ratio is set to 284.7 ppm
during the spin-up. Nutrients and oxygen concentrations in the ocean are initialised with the World Ocean Atlas
dataset (Garcia et al., 2013a, b). Initial DIC and TA fields are taken from the Global Data Analysis Project (Key
et al., 2004). After 900 years, the ocean physical- and biogeochemical tracer distributions reach quasi-equilibrium
states. We extended the spin-up for another 200 years with riverine input for each experiment (except for the
reference run) and then performed a set of transient climate simulations for the industrial era and the 21$^{st}$ century
(1850-2100). The simulations use external climate forcings that follow the CMIP5 protocol (Taylor et al., 2011).
For the historical period (1850-2005), observed time-varying solar radiation, atmospheric greenhouse gas
concentrations (including $CO_2$), natural and anthropogenic aerosols are prescribed. For the future period (2006-
2100), the Representative Concentration Pathway (RCP) 4.5 (van Vuuren et al., 2011) is applied. Here, we
consider RCP4.5 as the representative future scenario following the $CO_2$ emission rate based on the submitted
Intended Nationally Determined Contributions, which projects a median warming of 2.6–3.1°C by 2100 (Rogelj
et al., 2016). The riverine input configurations employed in this study are summarized in Figure 1. The evolution
of global total fluxes of each nutrient/carbon species are shown in Figure 2. The experiment configurations are
described as follows:
●    REF: Reference run. Riverine nutrient and carbon supply is deactivated.
●    FIX and FIXnoc: Fixed at recent-past level. FIX: A constant riverine nutrient and carbon supply,

289           representative for the year 1970 as provided by NEWS 2, is applied to the model throughout the whole

290           experiment duration. FIXnoc: As FIX but only with nutrients supply, all carbon (DIC, DOC, POC) and TA

291           fluxes are deactivated.

●    RUN: Coupled to simulated freshwater runoff. Riverine nutrient and carbon supply representative for the

293           year 1970 is linearly scaled with the on-line simulated freshwater runoff divided by the climatological mean

294           runoff over 1960-1979 of the model. Thus, the inputs follow the seasonality and long-term trend of the

simulated runoff. We assume that the nutrient and carbon concentrations in the rivers are constant at the
level of 1970, but the fluxes fluctuate with freshwater runoff.
•    GNS: Four different transient inputs following future projections of NEWS 2. A constant riverine nutrient
and carbon supply representative for year 1970 has been applied from year 1850 to 1970. Between year 1970,
2000, 2030 and 2050 the annual riverine supply is linearly interpolated. From year 2050 to 2100 the annual
riverine supply is linearly extrapolated. From year 2000, riverine supplies of the four NEWS 2 future
scenarios (GNa, GNg, GNo and GNt) are applied.
By comparing FIX versus REF, we assess how the presence of riverine inputs affect the contemporary marine PP
and C uptake representation and also the projected changes. By comparing RUN versus FIX we assess the
potential effects of riverine nutrient and carbon long-term trends associated with an intensifying global
hydrological cycle on marine PP and C uptake. RUN represents a first step towards coupling riverine nutrient and
carbon fluxes to the simulated hydrological cycle. By comparing the GNS configurations versus FIX we assess
how plausible, realistic future evolutions in riverine nutrient and carbon fluxes may impact marine PP and C
uptake projections. We span the uncertainty in future riverine nutrient and carbon fluxes by considering multiple
NEWS 2 scenarios.

## 3 Results

### 3.1 Effect of including riverine inputs on contemporary marine PP and C uptake

We start with assessing how the inclusion of riverine nutrients and carbon affects the contemporary representation
of the global marine PP and C uptake in our model by comparing the annual mean output over the years 2003–
2012 between the REF and FIX experiments. We also compare with satellite and observational based estimates
to see if the inclusion of riverine nutrients and carbon improves the marine PP and C uptake representation in our
model. The spatially integrated values presented in this and following sections are summarized and supplemented
with statistical robustness information in Tables B1 and B2 in Appendix B.
The annual net primary production (PP) is 40.1 and 43.0 Pg C yr$^{-1}$ in the REF and FIX experiments, respectively.
The increase of PP in FIX occurs along continental margins (where seafloor is shallower than 300 m) and also in
the North Atlantic region (0°N-65°N, 0°W-90°W), accounting for 15.4% and 24.9% of the global total increase,
respectively (Figure 3c). The simulated global total PP in both REF and FIX are lower than the satellite-based
model estimates, including Vertically Generalized Production Model (VGPM), Eppley-VGPM and Carbon-based
Production Model (CbPM) over the same time period (data source:
http://www.science.oregonstate.edu/ocean.productivity), ranging from 55 to 61 Pg C yr$^{-1}$ (Behrenfeld and
Falkowski, 1997; Westberry et al., 2008). Although the total PP in FIX is still considerably lower than the satellite-
based estimates, the inclusion of riverine nutrients and carbon does slightly improve the distribution of PP
especially on continental margins (Figure 3), according to our area-weighted root mean square error (RMSE)
analysis. The RMSE of REF relative to mean observational estimates (mentioned above) averages 10.7 mol C m$^{-2}$
yr$^{-1}$ globally, while the value of FIX is 10.3 mol C m$^{-2}$ yr$^{-1}$, which is reduced by 3.7%. For the continental margins,
the RMSE is reduced by 5.5% from 29.0 mol C m$^{-2}$ yr$^{-1}$ in REF to 27.4 mol C m$^{-2}$ yr$^{-1}$ in FIX.
The ocean annual net uptake of $CO_2$ is 2.8 and 2.9 Pg C $yr^{-1}$ in REF and FIX, respectively, with a FIX-REF
difference of 0.1 Pg C $yr^{-1}$ equivalent to 3.1% relative change, which is statistically significant (see Table B2). In
FIX the ocean carbon uptake is generally enhanced everywhere except for the upwelling regions of the Southern
Ocean and in the subpolar North Atlantic between approximately 50°N-65°N and 60°W-10°W (Figure 4c). To
isolate the impact of riverine nutrients input from carbon input, an additional experiment (FIXnoc) was conducted,
where the nutrient fluxes are implemented the same as in FIX, while all carbon (DIC, DOC, POC) and TA fluxes
are eliminated. As shown in Figure 4d, the nutrients input results in more $CO_2$ uptake not only at large river
estuaries but also in the subtropical gyres due to enhanced primary production. In the subpolar North Atlantic and
in the Southern Ocean upwelling region, the addition of riverine nutrients leads to enhanced outgassing. The
riverine carbon input, on the other hand, leads to $CO_2$ outgassing mainly at river estuaries (Figure 4e), but also in
a band along the gulf stream extending into the North Atlantic, where it accounts for 18.1% of the $CO_2$ outgassing
in the subpolar region (50°N-65°N, 60°W-10°W). Along the continental margins the nutrients input increases the
$CO_2$ uptake, while the carbon input has an opposite effect which induces more outgassing. The net effect of both
nutrient and carbon inputs shows that the uptake of $CO_2$ dominates over the outgassing, along the continental
margins and in subtropical gyres (Figure 4c). Compared to the observational based estimates of Landschützer et
al. (2017) (Figure 4a) and according to our RMSE analysis, the inclusion of riverine nutrients and carbon does
not improve the simulated air-sea $CO_2$ fluxes globally. The RMSE of FIX relative to observational estimates
averages to 0.83 mol C $m^{-2}$ $yr^{-1}$ globally, which does not differ much from the value of REF (0.84 mol C $m^{-2}$ $yr^{-1}$
). However, there is a distinguishable improvement of the distribution of air-sea $CO_2$ fluxes in the subpolar North
Atlantic (RMSE is reduced by 8.2%, from 0.73 mol C $m^{-2}$ $yr^{-1}$ in REF to 0.67 mol C $m^{-2}$ $yr^{-1}$ in FIX), with slight
degradations in some other regions (Figure 4c).

**3.2 Effect of including contemporary riverine inputs on projections of marine PP and C uptake**

We now address how the inclusion of riverine nutrient and carbon fluxes affects projections of marine PP and C
uptake by comparing the average output between a future period (2050–2099) and a historical period (1950–1999)
of FIX versus REF.
In both experiments the projections of global PP averaged over the years 2050–2099 are lower than their
corresponding 1950–1999 averages (Figure 5a). However, when riverine input of nutrient and carbon is included,
the projected decrease of global PP is mitigated from -2.2 Pg C $yr^{-1}$ in REF to -1.9 Pg C $yr^{-1}$ in FIX (by 13.6%).
Spatially, the decrease of PP in REF occurs largely in upwelling regions such as the tropical eastern Pacific and
tropical Atlantic, as well as along a latitude band around 40ºS (Figure 6a). The riverine inputs alleviate the
projected PP decrease in those regions (see further discussion in Section 4.2) and reinforce the projected PP
increase in high latitudes (Figure 6b, c). The projections of PP in the Arctic Ocean show significant increases in
both REF and FIX. Climate change alone (REF, without riverine inputs) almost doubles the simulated PP in the
Arctic from 0.08 Pg C $yr^{-1}$ during 1950–1999 to 0.15 Pg C $yr^{-1}$ in 2050–2099 (Figure 5b), likely as a consequence
of sea ice retreat. FIX, which includes riverine inputs, exhibits a slightly larger (but significant, see Table B1)
absolute Arctic PP increase (from 0.10 to 0.18 Pg C $yr^{-1}$) in its future projection than REF.
For global net uptake rate of $CO_2$, both experiments (REF and FIX) project a significant increase under the RCP4.5
(Figure 7a). The inclusion of riverine inputs leads to a slightly higher (but significant, see Table B2) (2.4%)
projected increase of 1.28 Pg C yr$^{-1}$ in FIX compared with 1.25 Pg C yr$^{-1}$ in REF. The increase rate of $CO_2$ uptake
in the Arctic closely follows the global trend (Figure 7b). Spatially, there is a widespread simulated increase in
ocean uptake of $CO_2$ under future climate change except in the subtropical gyres (Figure 8a). Riverine nutrients
input slightly increases the projected carbon uptake at large river estuaries, while decreases the projected uptake
in subpolar North Atlantic (Figure 8d).

**3.3 Effect of future changes in riverine inputs on marine PP and C uptake projections**

Finally, we address how future changes in riverine fluxes of nutrients and carbon affect marine PP and C uptake
by comparing the projected changes for the time period 2050–2099 relative to 1950–1999 among FIX, RUN and
the four GNS experiments.
The future projected decrease of PP in the four GNS averages to -1.6 Pg C yr$^{-1}$, which is less in magnitude
compared to FIX (-1.9 Pg C yr$^{-1}$) and RUN (-1.8 Pg C yr$^{-1}$) (Figure 5a). Spatial distributions of projected PP
changes in GNS and their respective differences relative to FIX are shown in Figure 9. The latter occur
predominantly on the continental shelf in Southeast Asia, where the future projected increase in riverine nutrient
load is the largest in the world in GNS (Seitzinger et al., 2010). Interestingly, the projected increase in PP in
Southeast Asia, induced by riverine nutrient inputs in GNS, is of the same order of magnitude as the projected
decrease in PP due to future climate change in REF. Thus, in GNS the PP are projected to slightly increase on the
continental shelf of Southeast Asia (Figure 9a-d). The riverine nutrient induced PP increase in FIX or RUN is not
large enough to compensate the PP decline due to climate change, since the projected changes in riverine nutrient
inputs are not taken into account in FIX or locally underestimated in RUN.
On the other hand, the future projected global uptake of $CO_2$ in GNS (1.13 Pg C yr$^{-1}$ in average) is reduced
compared to REF (1.25 Pg C yr$^{-1}$), which shows an opposite change than FIX (1.28 Pg C yr$^{-1}$) and RUN (1.29 Pg
C yr$^{-1}$). The changes in riverine inputs in GNS emerge along continental margins, especially around large river
estuaries (Figure 10e-h), where the dissolved organic matter (DOM), that is projected to increase in GNS, enters
the ocean and releases $CO_2$ to the atmosphere (Seitzinger et al., 2010).
Despite the regional differences, there is no significant difference in the projected changes in either globally
integrated PP or $CO_2$ uptake among the four GNS in our model (Figures 5 and 7, see further discussion in Section
395 4.3).

**4 Discussion**

**4.1 Projected marine PP and C uptake under climate change**

In our model, PP is roughly linearly related to the concentrations of the most limiting nutrient (Nut), light intensity
(I), temperature (T) and the available phytoplankton concentration (Phy), i.e., PP ~ Nut · I ·f(T) · Phy. It is shown
in Figure 6a that under climate change the projected decrease in PP occurs mainly in low- and mid-latitudes.
Nitrate is the limiting nutrient (in REF) in almost everywhere except in the Central Indo-Pacific region, in the
South Pacific subtropical gyre, in the Bering Sea and part of the Arctic, where Fe is limiting (Figure A1). Projected
reduction in surface nitrate concentrations (Figure A2b), which is tightly linked to the upper ocean warming and
increased vertical stratification (Bopp et al., 2001; Behrenfeld et al., 2006; Steinacher et al., 2010; Cabré et al.,
2015), contributes to the projected decrease in PP in our model. The simulated global mean PP over 2050–2099
is 38.9 Pg C $yr^{-1}$ in REF, which is 2.24 Pg C $yr^{-1}$ lower than the value over 1950–1999. This -5.4% projected
change in PP is comparable with the multi-model mean estimate of projected change of -3.6 ± 5.7% in the 2090s
relative to the 1990s for RCP4.5 (Bopp et al., 2013) and sits in the range of 2-13% decrease projected by four
ESMs over the 21st century under the SRES A2 scenario (Steinacher et al., 2010). It is also still within the range
of the 13 multi-model mean projected PP change of -1.13 ± 5.81% under the CMIP6 Shared Socioeconomic
Pathways SSP2-4.5 when comparing mean values in 2080–2099 relative to 1870–1899 (Kwiatkowski et al., 2020),
given that the inter-model uncertainties in projected PP have increased in CMIP6 compared to CMIP5 (Tagliabue
et al., 2021). In contrast to the global PP, there are considerable increases in the future projected PP in the Arctic
in REF (Figure 5b). In polar regions light and temperature are the primary limiting factors for phytoplankton
growth, therefore PP increases when light and temperature become more favourable owing to sea-ice melting
under warmer conditions (Sarmiento et al., 2004; Bopp et al., 2005; Doney, 2006; Steinacher et al., 2010). On the
other hand, the fresher and warmer surface water increases stratification, prohibiting nutrients upwelling
(Vancoppenolle et al., 2013; Figure A2b), which counteracts the increase in PP.
The ocean annual net uptake of $CO_2$ increases significantly during 2050–2099 compared with the uptake during
1950–1999 in REF (Figure 7a), which is mainly driven by increasing difference in air-sea partial pressure of $CO_2$.

## 4.2 Changes in projected marine PP and C uptake due to riverine input

When riverine nutrient fluxes are added into coastal surface waters in FIX, the PP is higher in both historical and
future periods compared to REF (Figure 5a), due to alleviated nutrient limitation. Interestingly, the effect of
riverine inputs on PP for the historical and future time periods is not the same, suggesting a different nutrient
depletion level (Figure A2b). The projected decrease in PP is lessened from -5.4% in REF to -4.4% in FIX. It
implies that during 1950–1999 the riverine nutrients are not depleted by primary producers, while during 2050–
2099 the riverine nutrients are utilized to a greater extent due to the exacerbated nutrient limitation (Figure A2b)
and potentially to higher phytoplankton growth rate in warmer climate. Figure 12 illustrates this in a schematic
diagram that shows the impact of riverine nutrients on projected PP in low- and mid-latitudes. Moreover, the
inclusion of constant riverine inputs (FIX) can potentially explain one tenth of the ~10% (2-13%, Steinacher et
al., 2010) inter-model spread. In RUN and GNS, the projected decline in PP is further alleviated to -4.1% and -
3.6% (averaged over four GN scenarios), respectively, compared to -4.4% in FIX, owing to the varying (mostly
increase) nutrients input. In the Arctic, when riverine nutrients input is present in the model, it helps to sustain the
projected PP increase against the stronger stratification under future climate warming, although this effect is only
minor (Figure 5b).
The riverine inputs have a two-fold effect on the ocean C uptake. It is the competition between the riverine
(inorganic and organic) nutrients input induced $CO_2$ uptake and the riverine carbon input induced $CO_2$ outgassing,
which determines whether the shelf is a C sink or a C source. However, the composition of the riverine organic
matter (i.e., carbon to nutrient ratio) and the degradation timescales which are the key factors, have been debated
over the last three decades (Ittekkot, 1988; Hedges et al., 1997; Cai, 2010; Bianchi, 2011; Blair and Aller, 2011;
Lalonde et al., 2014; Galy et al., 2015). It is generally agreed that the riverine organic carbon to nutrient ratio is
high (e.g., C:P weight ratio larger than 700, Seitzinger et al., 2010) and the degradation and resuspension rates in
shallow shelf seas/sediment are higher than the open ocean (Krumins et al., 2013). It suggests that at shallow and
near-shore areas the riverine carbon input usually results in a $CO_2$ source to the atmosphere, while at deeper outer
shelf areas the riverine nutrient input causes PP increase and a $CO_2$ sink, and the magnitudes of the C source and
sink on the continental shelves almost compensate each other. This phenomenon has been discussed by both
measurement-based studies (Borges and Frankignoulle, 2005; Chen and Borges, 2009) and modelling studies (e.g.,
Lacroix et al., 2020). However, the spatial resolution in our model is not fine enough to differentiate the near-
shore and outer shelf processes. This partly contributes to comparable $CO_2$ outgassing near shore (due to riverine
C) and $CO_2$ ingassing on outer shelves (due to riverine inorganic and organic nutrients input), leading to a globally
weak integrated C sink on the continental margins in FIX and RUN experiments for both historical and future
time periods. Although the riverine input of nutrients and C are constant for both time periods in FIX, the riverine
induced C uptake is slightly (but significantly) bigger (0.03 Pg C yr$^{-1}$) during 2055–2099 compared to 1950–1999,
which indicates that the riverine nutrients input is slightly dominant over riverine C input in FIX, and the riverine
nutrients are utilized more in the future period. A recent modelling study (Lacroix et al., 2021a) with improved
shelf processes, has also reported a 0.03 Pg C yr$^{-1}$ increase in global C uptake induced by temporally varying
terrestrial nutrients input during 1905–2010. They conclude that due to large historical perturbation, the increased
nutrient inputs are the largest driver of change for the $CO_2$ uptake at the regional scale. In GNS, on the other hand,
the riverine inputs reduce globally integrated C uptake for both historical and future time periods, but not equally
(Figure 7a). It reduces more in the future period (2050–2099) than the historical period (1950–1999), which
implies that the effect of riverine C input in the future scenarios are more dominant over nutrients input.
Simulations with high-resolution global or regional models with more realistic representation of shelf processes
are required to accurately assess the impact of riverine inputs on carbon cycling in the coastal ocean.

## 4.3 Sensitivity of projected marine PP and C uptake to riverine configuration

By exploring different riverine configurations (FIX, RUN, GNS) we investigate how uncertainties in future
riverine fluxes translate into uncertainties in projected PP and C uptake changes. In RUN we assume constant
concentrations (at 1970's level) of riverine nutrient and carbon over time and couple them to the simulated
freshwater runoff. Thus, the annual global total fluxes of nutrient and carbon vary with time following the
variability of runoff (Figure 2), in contrast to the constant fluxes in FIX. The global total simulated runoff, under
RCP4.5 in our model, is on average higher during 2050–2099 than the runoff during 1950–1999, indicating an
intensified hydrological cycle under future climate change. Hence, the global riverine fluxes of nutrient and carbon
during 2050–2099 are higher than those during 1950–1999 in RUN. However, the temporal changes in global
riverine fluxes in RUN are relatively small compared with the absolute flux values in FIX, which explains the
slightly larger projected changes in global PP and ocean carbon uptake in RUN compared to FIX. It is noteworthy
that the large inter-annual variability in the riverine fluxes of nutrient and carbon in RUN does not increase the
inter-annual variability in simulated PP and ocean carbon uptake either globally or on the continental margins
(Figure 11), something that warrants further investigation. The approach of RUN serves as a trial to introduce
seasonal and inter-annual variability in riverine nutrient and C inputs that is linked to hydrological variability. It
should be explored in future works if RUN and GNS can be integrated to produce more realistic long-term trends
in riverine nutrient and C inputs as well as short-term variability. Although the RUN approach is more
sophisticated when compared to FIX, it employs a linear relationship between the future riverine nutrient and C
fluxes and the simulated hydrological cycle, which is a highly simplified assumption (see discussion in section
483      4.4).

Figure 2 shows that the inputs of DIN and DIP are considerably lower, while the dissolved silicon (DSi) and
particulate organic matter (POM) are higher in the future period in RUN compared to GNS. This is because many
anthropogenic processes that are important for determining the future riverine fluxes are not considered in RUN,
but are considered in NEWS 2 model system, from which the GNS' future scenarios are simulated. For example,
the nutrient management in agriculture, the sewage treatment and phosphorus detergent use, and the increased
reservoirs from global dam construction in river system (Seitzinger et al., 2010; Beusen et al., 2009) are the key
factors affecting future riverine fluxes of DIN, DIP, and DSi/POM, respectively. Therefore, it is worth exploring
the merits of using GNS in future projections of marine biogeochemistry. The four future scenarios provide a
range of potential outcomes resulting from different choices tending toward either globalization or regional
orientation, either reactive or proactive approach to environmental threats (see Table 1). A large range of the
riverine inputs in GNS, e.g., temporal changes in DIN fluxes across scenarios ranging 24.8-63.0% of the annual
flux in FIX, do not transfer to large uncertainties in future projections of global marine PP in our model, which
can primarily be attributed to unresolved shelf processes due to coarse model resolution. However, the scenario
differences might be of importance in regional projections, such as in seas surrounded by highly populated nations
and near river estuaries. Simulations with high-resolution global or regional models with a good representation of
shelf processes are required to accurately assess the local impact of riverine inputs.
**4.4 Limitations and uncertainties**
We acknowledge several limitations of our study, particularly related to the resolution and complexity of our
model. Firstly, coarse-resolution models tend to underestimate PP along the coast. Such well-known model issues
may offset the impact induced by riverine inputs. Secondly, shelf processes, which are not well represented in our
model due to coarse resolution, modify a large fraction of some riverine species, e.g., conversion of organic carbon
to $CO_2$ occurs rapidly via remineralization in estuaries before they are transported to the open ocean. Further,
some simplified processes of the model may introduce bias in the results, e.g., how the model deals with the
riverine dissolved organic and particulate matter. In our model, there is only one dissolved organic pool (DOM)
and one particulate organic pool (DET), and the Redfield ratio (P-N-C) needs to be kept. Therefore, the P-N-C
ratios of riverine input for both dissolved organic matter (including DON, DOP and DOC) and particulate
(inorganic and organic) matter (including particulate nitrogen, particulate phosphorus and POC) are calculated,
then the least abundant species (scaled by the Redfield ratio) are added to the DOM and DET pools, respectively.
The excess budget from the remaining two species (of P, N or C) are assumed to be directly remineralized into

inorganic form and added to the corresponding dissolved inorganic pools (i.e., DIP, DIN, or DIC) in the ocean. This simplification may result in overestimation of riverine dissolved inorganic nutrients and thereby riverine induced PP enhancement. Especially, in NEW 2 dataset particulate P is typically dominated by inorganic forms (Mayorga et al., 2010), which means that it is likely not directly bio-available. Therefore, we have assessed the bias due to the direct remineralization of the riverine dissolved organic and particulate matter. We calculated firstly the proportion of directly remineralized matter from the total riverine dissolved organic matter (DOM) and particulate (inorganic and organic) matter (PM) by using the following equation, i.e., $[X/(DOM_{riv}+PM_{riv})*100\%]$ (X is the directly remineralized dissolved organic and particulate matter). The directly remineralized part on average accounts for 64.8%, 27.8% and 62.8% of the total riverine organic and particulate matter of P, N and C, respectively. In a recent study by Lacroix et al. (2021b) who used an enhanced version of HAMOCC (horizontal resolution of ~0.4°) with improved representation of riverine inputs and organic matter dynamics in the coastal ocean, they quantified that around 50% of the riverine DOM and 75% of the POM are mineralized in global shelf waters. Therefore, our model assumption is on track with the finer-resolution-model estimates and this direct remineralization compensates to some extent the under-represented organic matter degradation rate on the ocean shelf. This bias in riverine dissolved nutrient input may further lead to bias in the enhanced PP. We calculated the contribution of the directly remineralized part on the enhanced PP, by comparing X with the corresponding total riverine dissolved nutrient additions as $[X/(X+DIX_{riv})*100\%]$ ($DIX_{riv}$ denotes the corresponding riverine dissolved nutrient additions), which accounts for 80.5%, 33.3%, and 41.1% for P, N, and C, respectively. Assuming that all coastal regions are nutrient limited, this direct remineralization could be theoretically responsible for 33.3%-80.5% of the enhanced PP, depending on which nutrient species is limiting the PP. In our model, phosphate is rarely limiting (Figure A1), therefore, the impact of this direct remineralization on PP is likely on the lower end of this range (33.3%-80.5%). Given that the proportion of the direct remineralized organic N (27.8%, see the calculation above) in our model is comparable to or lower than the reported values by field studies (~38.8% of DON decomposed during transition from Arctic rivers to coastal ocean; Kattner et al., 1999; Lobbes et al., 2000; Dittmar et al., 2001), which indicates that the bias on enhanced PP is likely less than 33.3%.

Some approximation and assumption in the experimental setup may also induce uncertainties in our results. Our spin-up experiment uses riverine nutrient and carbon inputs fixed at 1970 levels, as provided by NEWS 2. As a caveat, our post-1970 simulated changes in marine PP and $CO_2$ fluxes miss out any legacy effects from riverine input changes that occurred before 1970. The fixed inputs likely overestimate the accumulated inputs prior 1970, causing potential underestimation of the projected change impacts. However, Beusen et al. (2016) found that changes in riverine N and P are relatively small before 1970 compared to changes after 1970. Therefore, we expect the impact due to missing legacy effects to be minor. Moreover, in FIX we applied riverine inputs at 1970 level over available inputs at 2000 level, because the former are more representative for the 1950–1999 baseline period. However, the use of 1970 level input is suboptimal when evaluating simulated PP and $CO_2$ fluxes against observations obtained after 2000. Beusen et al. (2016) have shown that the riverine N and P has increased by ~40.0% and 28.6%, respectively, from 1970 to 2000. Therefore, the riverine impact may be underestimated when comparing with the observations during 2003–2012. In RUN, we assume constant concentrations of riverine nutrient and carbon over time and the fluxes vary with freshwater runoff. This may be applicable for some

nutrients such as DIN or within a certain limit of runoff change such as for dissolved Si (Figure A3). However,
this may not be appropriate for all nutrient/carbon species. Furthermore, the variability of runoff is subject to
inter-annual to decadal climate variability, which partially masks the centennial trend. This caveat can be
overcome through performing multi-realization ensemble simulations.
Lastly, riverine Fe flux is weighted by water runoff of each river and integrated globally as a total input of 1.45
Tg yr$^{-1}$ (Chester, 1990). To the best of our knowledge, the available global riverine iron dataset is rare. Previous
studies have used various approximation approaches, e.g., constant Fe to dissolved inorganic carbon (DIC) ratio
(Aumont et al., 2015), Fe to phosphorus ratio (Lacroix et al., 2020). In the study by Aumont et al. (2015), the Fe:
DIC ratio is determined so that the total Fe supply also equals 1.45 Tg Fe yr$^{-1}$ as estimated by Chester (1990). We
are aware that our approximation likely has bias in regional scales, especially in Fe limiting regions like the Arctic.
However, it has likely a minor impact on the projected PP, since light rather than riverine nutrients input is the
primary control of the projected Arctic PP in our model. Also, we have conducted all simulations only under one
IPCC representative concentration pathway scenario (the intermediate RCP 4.5), which may lead to a narrower
possible range of the riverine fluxes induced impact on the projected marine PP and C uptake.
**5 Conclusions**
In this study, we apply a fully coupled Earth system model to assess the impact of riverine nutrients and carbon
delivery to the ocean on the contemporary and future marine PP and carbon uptake. We also quantify the effects
of uncertainty in future riverine fluxes on the projected changes, using several riverine input configurations.
Compared to satellite- and observation-based estimates, the inclusion of riverine nutrients and carbon improves
the contemporary spatial distribution only slightly for PP (3.6% reduction in RMSE) and insignificantly for ocean
carbon uptake (0.1% reduction in RMSE) on a global scale, with larger improvements on the continental margins
(5.4% reduction in RMSE for PP) and the North Atlantic region (7.4% reduction in RMSE for carbon uptake).
Concerning future projected changes, decline in nutrients supply in tropical and subtropical surface waters, due
to upper ocean warming and increased vertical stratification, is projected by our model to reduce PP over the 21$^{st}$
century. Riverine nutrient inputs into surface coastal waters alleviate the nutrient limitation and considerably
lessen the projected future decline in PP from -5.4% without riverine inputs to -4.4%, -4.1% and -3.6% in FIX,
RUN and GNS (averaged over GNa, GNg, GNo and GNt), respectively. Different from the global value, the
projected PP in the Arctic increases considerably, because light and temperature—the primary limiting factors for
phytoplankton growth in polar regions—become more favorable due to sea-ice melting under warmer future
conditions. When riverine nutrient inputs are presented in the model, they further enhance the projected increase
in PP in the Arctic, counteracting the nutrient decline effect due to stronger stratification in the fresher and warmer
surface water.
Depending on the riverine scenarios, where the riverine nutrients input dominates over the C input, the projected
net uptake of $CO_2$ further enhances along continental margins via photosynthesis process. Conversely, where the
riverine C input is dominant over the nutrients input, the projected net uptake of $CO_2$ is reduced, especially at
large river estuaries, due to higher $CO_2$ outgassing.
We have explored a range of riverine input configurations from temporally constant fluxes (FIX), to idealized
time-varying fluxes following variations in simulated hydrological cycle (RUN), to plausible future scenarios
(GNS) from a set of global assumptions. The large range of the uncertainty of the riverine input does not transfer
to large uncertainty of the projected global PP and ocean C uptake in our simulations likely due to model
limitations related to resolution and shelf process representations. Our study suggests that applying transient
riverine inputs in the ESMs with coarse or intermediate model resolution (~1°) does not significantly reduce the
uncertainty in global marine PP and C uptake projections, but it may be of importance for regional studies such
as in the North Atlantic and along the continental margins.
Future modelling studies that include riverine input to the ocean can benefit from using high or at least adequate
model resolution, so that shelf processes, such as realistic remineralization rate for riverine organic matter in the
coastal water and shelf sediment as well as lateral transport, can be better resolved. Better constraints on riverine
C to nutrient ratios are needed to accurately assess the net riverine impact on ocean C uptake. Further exploration
of various future scenarios of riverine input is clearly warranted in order to better assess projected changes in
ocean PP and C uptake, especially in regional scales.
**Appendix A**

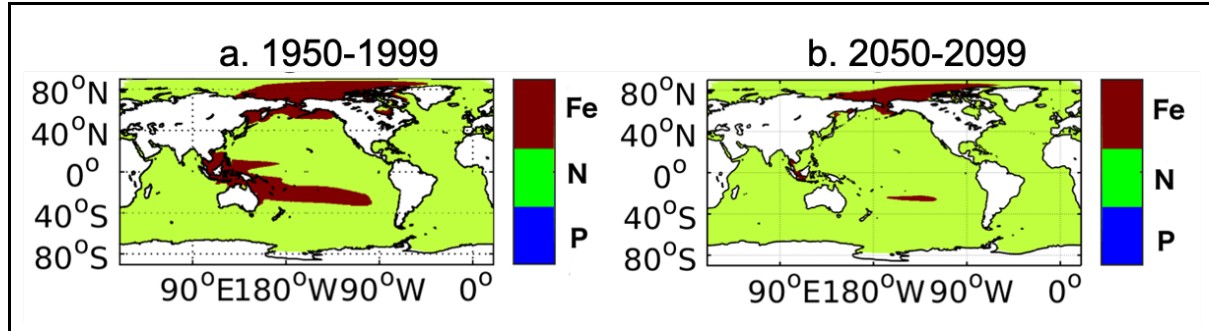

**Figure A1: The limiting nutrient among iron, nitrate and phosphate in REF during (a) 1950–1999 and (b) 2050–2099.**


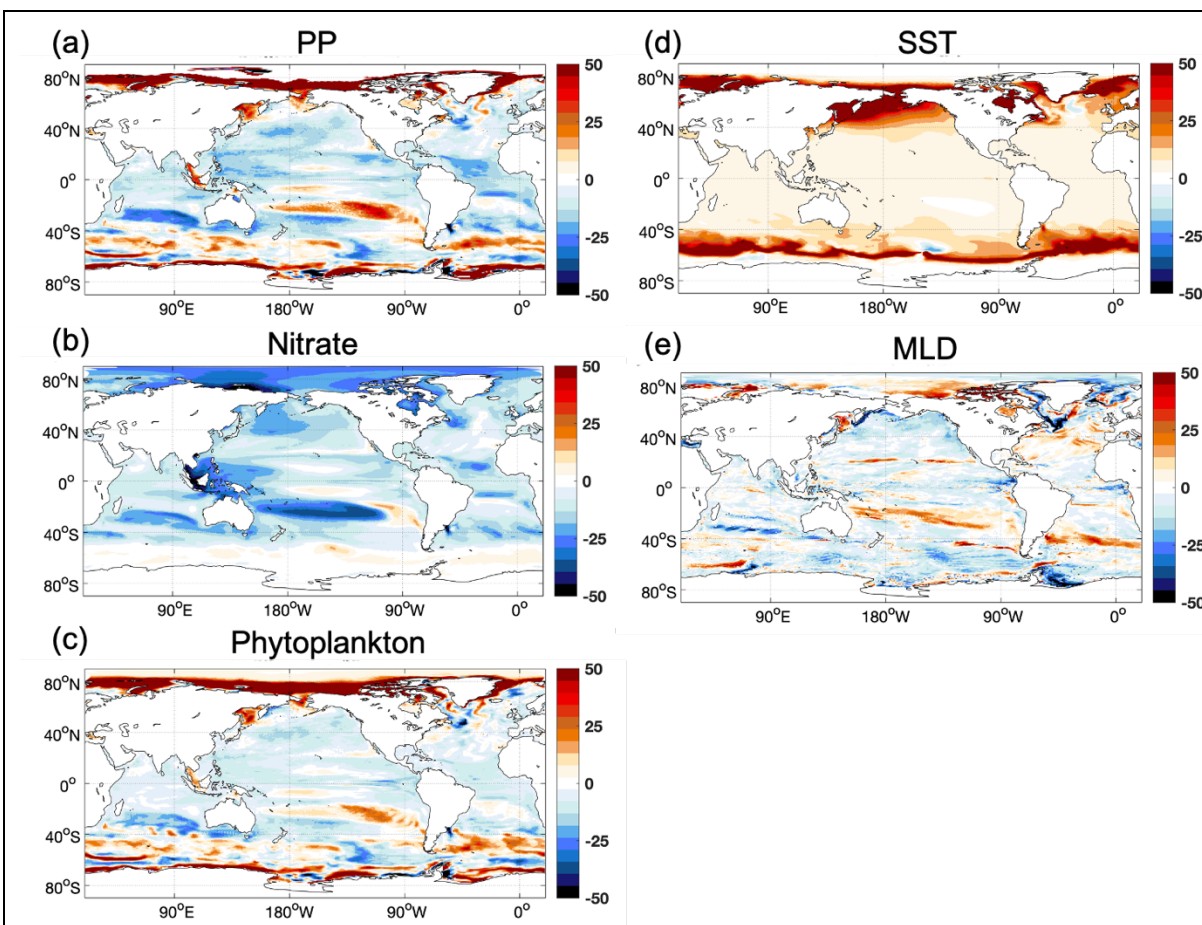

**Figure A2: The relative changes in projected (a) primary production, (b) nitrate concentration, (c) phytoplankton concentration, (d) sea surface temperature and (e) annual mean maximum mixed layer depth in REF (2050–2099 compared to 1950–1999).**


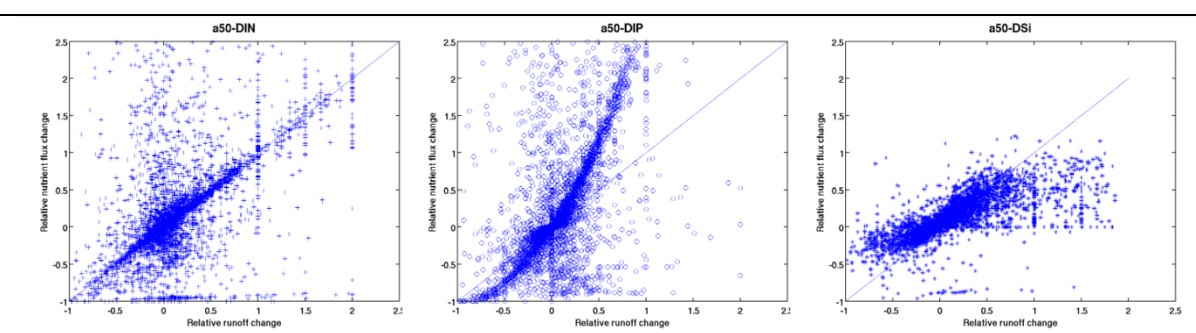

**Figure A3. The relationship between relative changes in freshwater runoff and relative changes in nutrient fluxes (dissolved inorganic nitrogen, phosphorus and silicon) in 2050 according to the Adapting Mosaic future scenario in NEW2 dataset.**

**Appendix B – Robustness of results to sampling error**
Time-averaged quantities and their differences—like the ones considered in this study—are subject to temporal
sampling uncertainty arising from the presence of internal climate variability and associated biogeochemistry
variability. We evaluated the statistical robustness of our results with respect to temporal sampling uncertainty as
outlined in the following.
We assessed statistical significance of time-averaged differences using Student's t-test. We performed the test on
annual data with $\alpha$ set to 0.05 and N set to the number of years in the respective average period, assuming the
internal climate variability exhibits most power on interannual and shorter timescales. We removed the main part
of the externally forced signal by subtracting the linear trend of the annual timeseries prior to performing the t-
test if the timeseries contained more than 20 years. For shorter time series, we therefore did not remove the linear
trend as it potentially has a large internal variability component.
All differences presented in the main text, summarized in Tables B1 and B2, were found to be statistically
significant and the plots feature only differences for which the t-test locally rejected the null-hypothesis. We found
even small inter-simulation differences statistically significant because these differences were less affected by
internal variability. In our model setup, the marine biogeochemistry does not feedback on the physical climate.
Consequently, the climate variability and climate trends are the same in all experiments and the interannual
variability in the biogeochemical parameters—which is predominantly driven by the physical climate
variability—is also virtually the same. As illustrated in Figure B1, any uncertainty related to internal climate
variability is effectively removed in the computation of the inter-experiment differences. In this manner, we were
able to obtain statistically robust results for short time-slices without having to perform multi-member simulation
ensembles for each experiment.
Detectability of inter-simulation differences does, however, not guarantee that the differences are large enough to
be competitive with real-world internal variability to have real-world implications. Therefore, we additionally
compared the inter-simulation differences against the internal variability of the absolute field (i.e., not the
difference field). We estimated the joint internal variability of the absolute field for N-year time averages as
$$\sigma_{\mu_{AB}} = \frac{\sqrt{\sigma_A{}^2 + \sigma_B{}^2}}{\sqrt{2N}}$$

where $\sigma_A$ and $\sigma_B$ are the interannual standard deviations for experiment A and B, respectively. As for the t-test,
we removed the externally forced signal by subtracting the linear trend of the annual timeseries prior to computing
standard-deviations if N>20. On all difference plots we marked the areas where inter-simulation differences
exceed $\sigma_{\mu_{AB}}$ and thus are large enough to have real-world implications.

**Table B1: Global and regional statistics of simulated primary production. Shown are the time-mean $\mu$ and twice its**
**standard-deviation $\sigma_\mu$ (rounded up to two decimals) derived from annual values. The $t_{his}$ and $t_{fut}$ denote the time**
**periods 1950–1999 and 2050–2099, respectively.**

| Variable | Experiment | Period | Region | $\mu \pm 2\,\sigma_\mu$ |
|---|---|---|---|---|
| RMSE of PP (mol C m$^{-2}$ yr$^{-1}$) | REF | 2003–2012 | Global | $10.70 \pm 0.18$ |
| | | | Continental margins | $28.96 \pm 0.18$ |
| | FIX | | Global | $10.31 \pm 0.21$ |
| | | | Continental margins | $27.43 \pm 0.19$ |
| | FIX-REF | | Global | $-0.39 \pm 0.04$ |
| | | | Continental margins | $-1.52 \pm 0.04$ |
| PP (Pg C yr$^{-1}$) | REF | 2003–2012 | Global | $40.06 \pm 0.50$ |
| | FIX | | | $42.99 \pm 0.51$ |
| | FIX-REF | | | $2.93 \pm 0.02$ |
| PP projection (Pg C yr$^{-1}$) | REF | $t_{his}$ | Arctic | $0.08 \pm 0.01$ |
| | | $t_{fut}$ | | $0.15 \pm 0.01$ |
| | | $t_{fut}$-$t_{his}$ | | $0.07 \pm 0.01$ |

| | | | | |
|---|---|---|---|---|
| | FIX | $t_{his}$ | | $0.10 \pm 0.01$ |
| | | $t_{fut}$ | | $0.18 \pm 0.01$ |
| | | $t_{fut}$-$t_{his}$ | | $0.08 \pm 0.01$ |
| | FIX-REF | $t_{fut}$-$t_{his}$ | | $0.01 \pm 0.01$ |
| PP projection (Pg C yr$^{-1}$) | REF | $t_{his}$ | Global | $41.14 \pm 0.26$ |
| | | $t_{fut}$ | | $38.90 \pm 0.23$ |
| | | $t_{fut}$-$t_{his}$ | | $-2.24 \pm 0.37$ |
| | FIX | $t_{his}$ | | $43.99 \pm 0.26$ |
| | | $t_{fut}$ | | $42.06 \pm 0.24$ |
| | | $t_{fut}$-$t_{his}$ | | $-1.93 \pm 0.38$ |
| | RUN | $t_{fut}$-$t_{his}$ | | $-1.82 \pm 0.38$ |
| | GNS | | | $-1.57 \pm 0.38$ |
| | FIX-REF | | | $0.31 \pm 0.01$ |
| | GNS-REF | | | $0.66 \pm 0.02$ |

**Table B2: Global and regional statistics of simulated ocean carbon uptake. Shown are the time-mean μ and twice its**
**standard-deviation $\sigma_\mu$ (rounded up to two decimals) derived from annual values. The $t_{his}$ and $t_{fut}$ denote the time**
**periods 1950–1999 and 2050–2099, respectively. Values in brackets denote relative changes in percentage.**

| Variable | Experiment | Period | Region | $\mu \pm 2\,\sigma_\mu$ |
|---|---|---|---|---|
| RMSE of C uptake (mol C m$^{-2}$ yr$^{-1}$) | REF | 2003–2012 | Global | $0.84 \pm 0.05$ |
| | | | Subpolar North Atlantic | $0.73 \pm 0.09$ |
| | FIX | | Global | $0.83 \pm 0.05$ |
| | | | Subpolar North Atlantic | $0.67 \pm 0.08$ |
| | FIX-REF | | Global | $-0.01 \pm 0.01$ |
| | | | Subpolar North Atlantic | $-0.06 \pm 0.01$ ($8.2 \pm 0.1\%$) |
| C uptake (Pg C yr$^{-1}$) | REF | 2003–2012 | Global | $2.77 \pm 0.06$ |
| | FIX | | | $2.86 \pm 0.07$ |
| | FIX-REF | | | $0.09 \pm 0.01$ ($3.1 \pm 0.1\%$) |
| C uptake projection (Pg C yr$^{-1}$) | REF | $t_{fut}$-$t_{his}$ | Global | $1.25 \pm 0.03$ |
| | FIX | | | $1.28 \pm 0.04$ |
| | RUN | | | $1.29 \pm 0.04$ |
| | GNS | | | $1.13 \pm 0.04$ |
| | FIX-REF | | | $0.03 \pm 0.01$ |
| | GNS-REF | | | $-0.11 \pm 0.03$ |


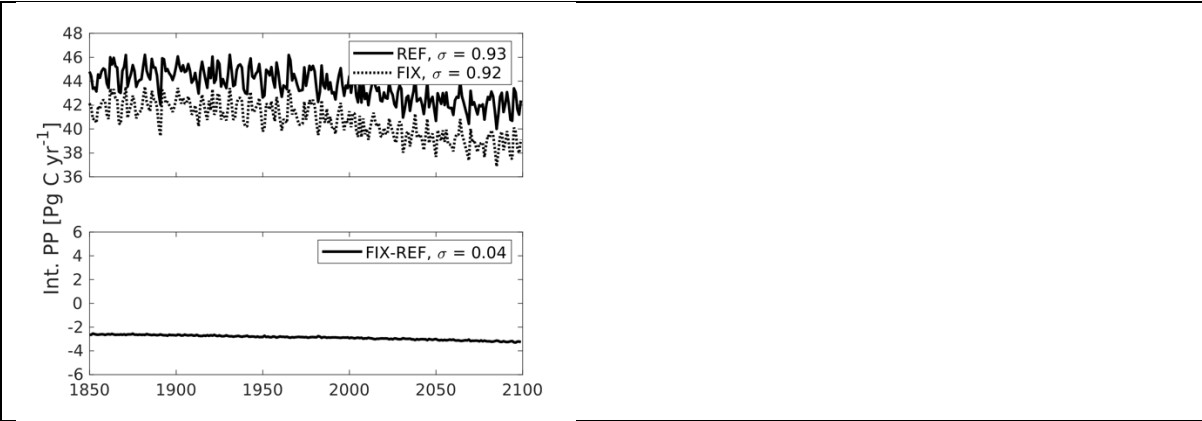

**Figure B1. Global integrated primary production (PP) time-series from single experiments (top) versus difference**
**between two experiments (bottom). The PP variability of REF and FIX closely follow each other because the**
**simulations feature the exact same physical variability. As a result, the interannual variability largely cancels out**
**in the computation of FIX-REF differences and the FIX-REF difference times-series exhibits a standard-deviation**
**that is an order of magnitude smaller than the standard-deviations of REF and FIX.**

**Appendix C – Comparison between NEWS 2 dataset and measurement-based riverine data**

**Table C1: Comparison between NEWS 2 dataset (Mayorga et al., 2010) and measurement-based (provided by PARTNERS Project; Holmes et al., 2012) riverine dissolved inorganic nitrogen (DIN) and dissolved organic nitrogen (DON) in the 6 largest Arctic rivers around year 2000.**

| River | DIN ($Pg\ N\ yr^{-1}$) | | DON ($Pg\ N\ yr^{-1}$) | |
|---|---|---|---|---|
| | NEWS 2 | Measurement | NEWS 2 | Measurement |
| Ob | 89 | 86 | 102 | 110 |
| Yenisei | 47 | 51 | 132 | 111 |
| Lena | 30 | 33 | 88 | 135 |
| Kolyma | 9 | 7 | 21 | 17 |
| Yukon | 5 | 26 | 14 | 47 |
| Mackenzie | 22 | 27 | 62 | 31 |

Note that the data from NEWS 2 are for the year 2000, while measured data from PARTNERS Project are calculated over 1999–2008 (missing discharge data restricted the Yukon estimates to 2001–2008).

**Code and data availability**

The model code, input data, output data and scripts used for producing the results and figures in the study are available at the NIRD Research Data Archive via https://doi.org/10.11582/2022.00072 with CC BY 4.0 license (Gao, 2022).

**Author contribution**

SG and IB designed the model experiments and SG developed the model code and performed the simulations with the help from IB. JS and JT contributed to the interpretation and analyzation of the results. JS, JT, IB and CH contributed to editing the manuscript. CH supervised the project work. JH and EM provided riverine data and consultation. SG prepared the manuscript with contributions from all co-authors.

**Competing interests**

The authors declare that they have no conflict of interest.

**Disclaimer**

This article reflects only the authors' view – the funding agencies as well as their executive agencies are not responsible for any use that may be made of the information that the article contains.

**Acknowledgement**

This work was supported through project CRESCENDO (Coordinated Research in Earth Systems and Climate: Experiments, kNowledge, Dissemination and Outreach; Horizon 2020 European Union's Framework Programme for Research and Innovation, grant no. 641816, European Commission). Computing and storage resources have been provided by UNINETT Sigma2 (nn2345k, nn2980k, ns2345k, ns2980k). JT acknowledge

the Research Council Funded project Downscaling Climate and Ocean Change to Services (CE2COAST; 318477).
IB received funding from the Trond Mohn Foundation through the Bjerknes Climate Prediction Unit
(BFS2018TMT01) and NFR Climate Futures (309562). JH benefited from financial support from the Deutsche
Forschungsgemeinschaft (DFG, German Research Foundation) under Germany's Excellence Strategy – EXC
2037 'Climate, Climatic Change, and Society' – project number 390683824, contribution to the Center for Earth
System Research and Sustainability (CEN) of Universität Hamburg. We gratefully acknowledge that the
schematic diagram of Figure 12 has been prepared by Jadelynn Fong. We thank Fabrice Lacroix and one
anonymous reviewer for their constructive comments which improved the manuscript greatly.

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

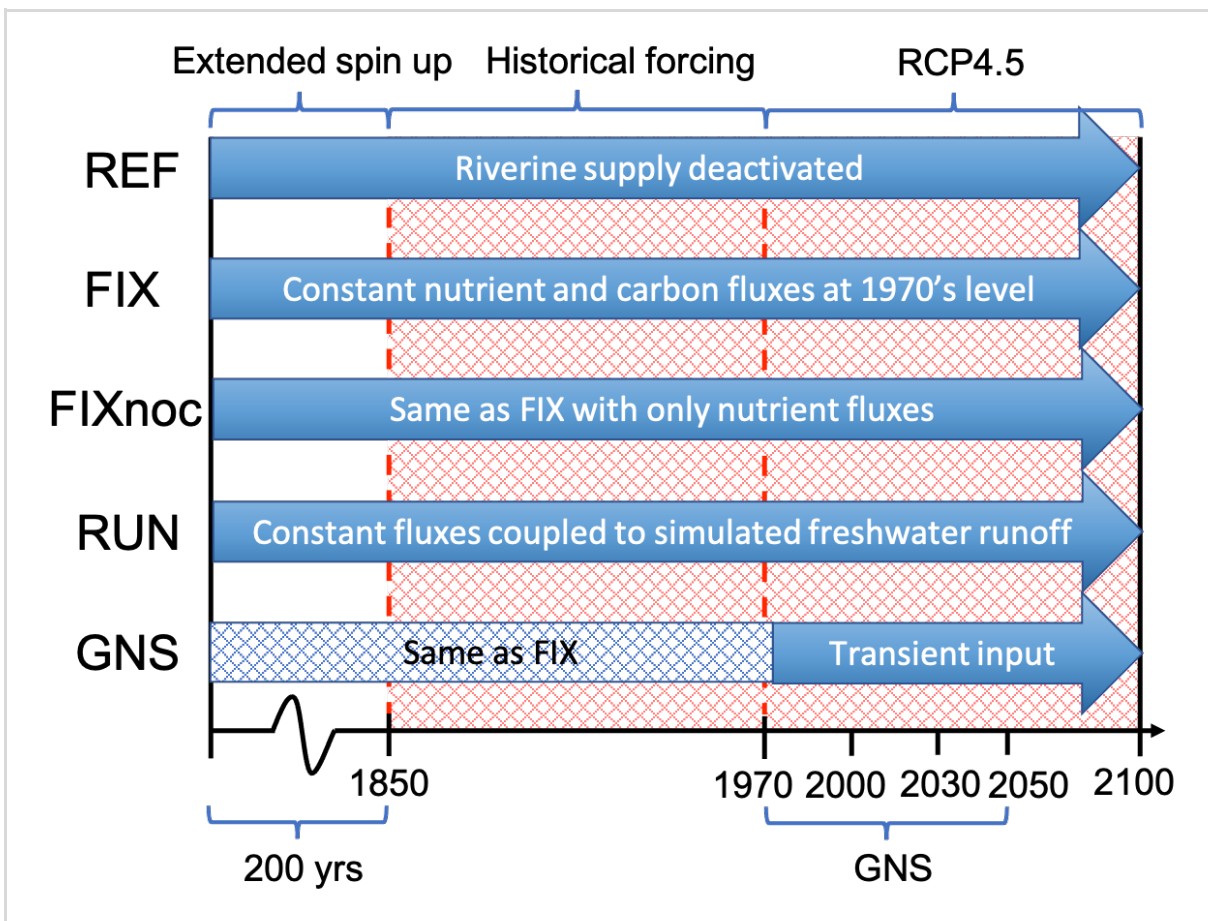

**Figure 1: Schematic illustration of the spin-up and integration procedure following the experimental design described in Section 2.4.**


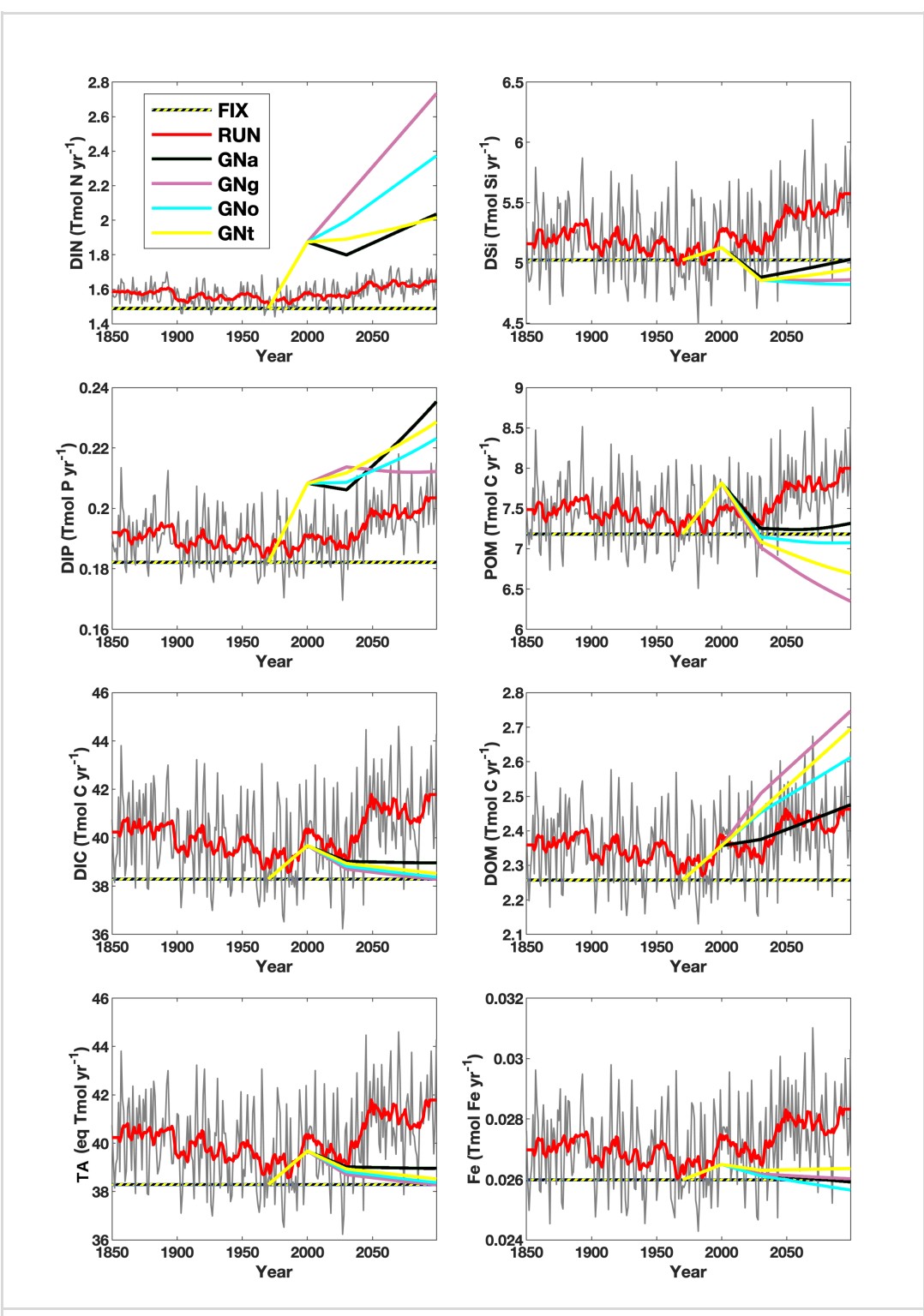

**Figure 2: Time series of global riverine fluxes of nutrient and carbon to the ocean according to the configuration of six model experiments (FIX, RUN, GNa, GNg, GNo and GNt). The thin grey and thick red curves are the annual and 11-year running mean fluxes, respectively, in RUN. DIN: dissolved inorganic nitrogen; DIP: dissolved inorganic phosphorus; DIC: dissolved inorganic carbon; TA: alkalinity; DSi: dissolved silicon; POM: particulate organic matter; DOM: dissolved organic matter; Fe: dissolved iron.**


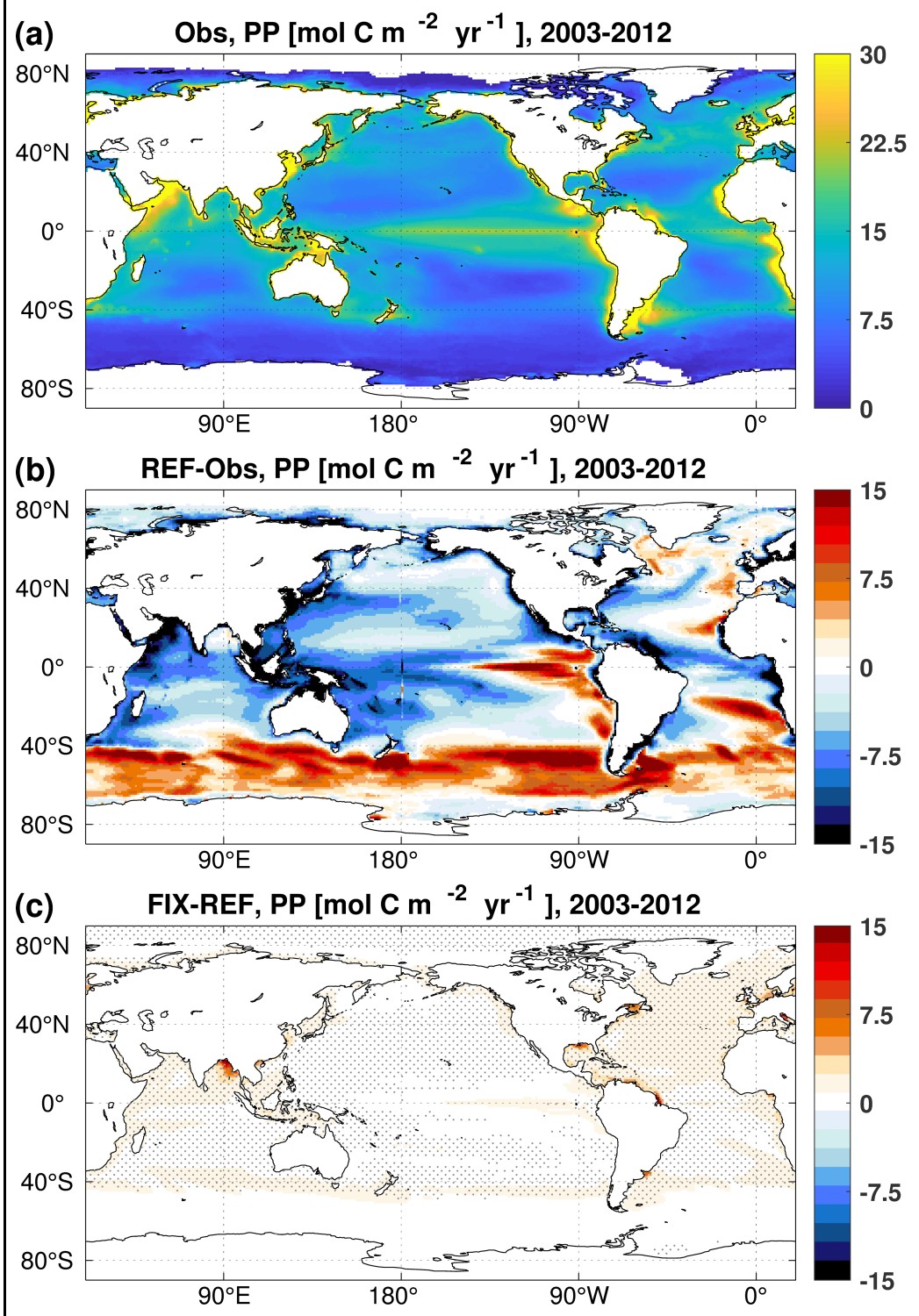

**Figure 3: Vertically integrated primary production averaged over the 2003–2012 period of (a) the mean of three satellite-based climatologies derived from MODIS retrievals, (b) the difference between REF and satellite-based estimates, (c) the difference between FIX and REF. In panel c, only significant differences are plotted, and dots denote areas where the signal is larger than the standard-deviation of the absolute field (see details in Appendix B).**

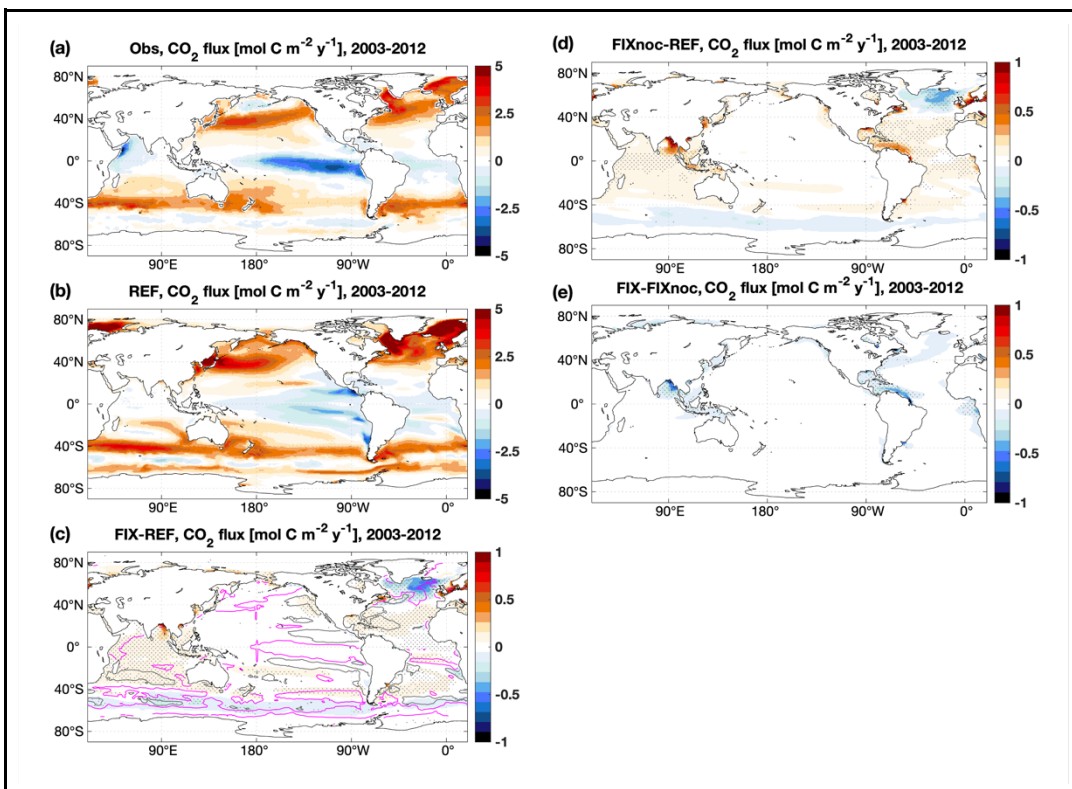

**Figure 4: Annual mean air-sea CO₂ fluxes over the 2003–2012 period of (a) the observational based estimates of Landschützer et al. (2017), (b) REF, (c) the difference between FIX and REF, (d) the difference between FIXnoc and REF, and (e) the difference between FIX and FIXnoc. Contour lines in (c) are the differences between REF and Obs, purple lines (0.6 mol C m⁻² yr⁻¹) indicate where REF overestimates C uptake compared to Obs and grey lines (-0.6 mol C m⁻² yr⁻¹) indicate the opposite. In panels c-e, only significant differences are plotted, and dots denote areas where the signal is larger than the standard-deviation of the absolute field (see details in Appendix B).**


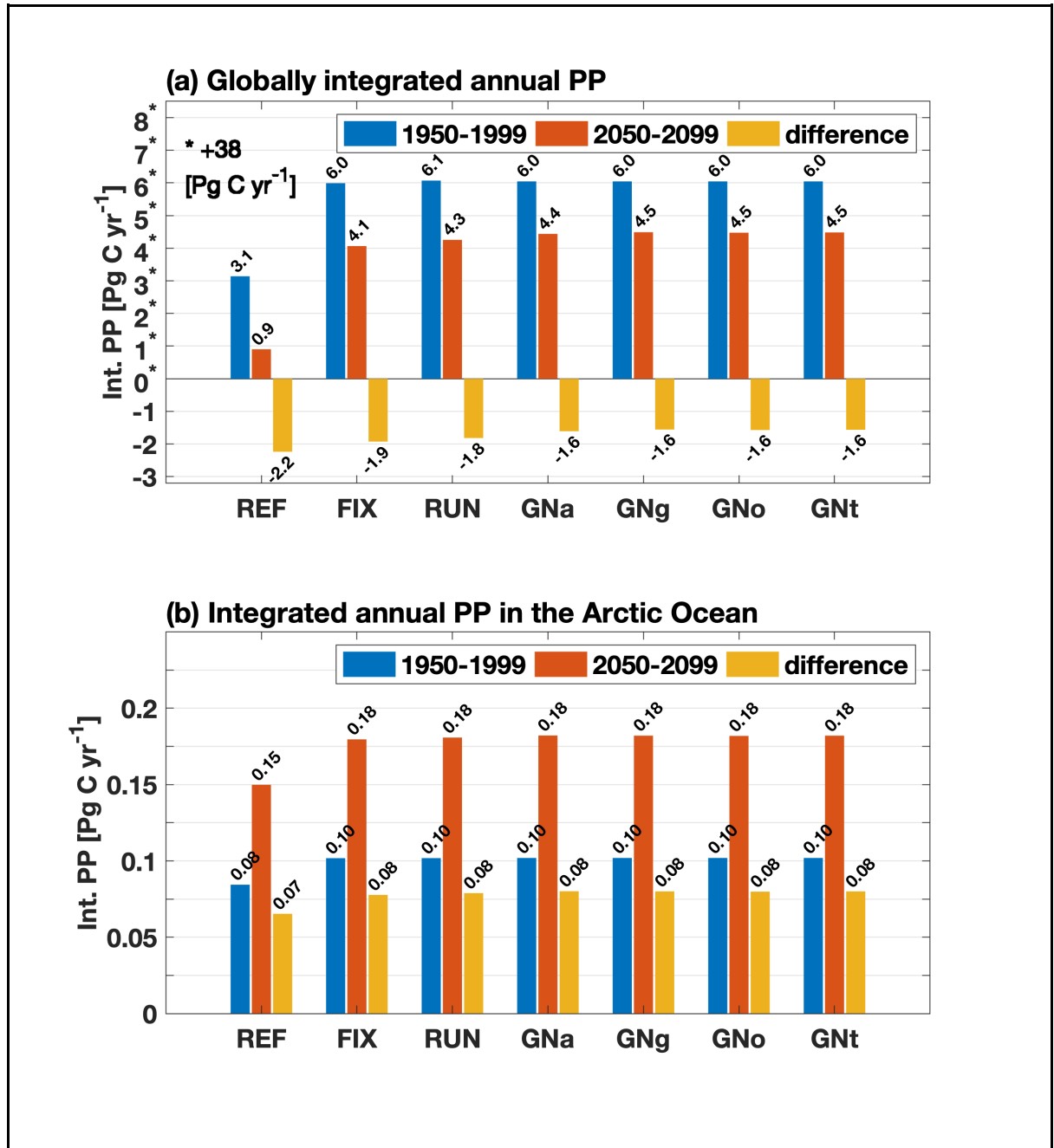

**Figure 5: (a) Globally integrated annual mean primary production over 1950–1999 (blue), over 2050–2099 (red) and the differences between these two time periods (yellow) for all experiments; note that the positive numbers in the y axis (marked with stars) are scaled by minus 38 Pg C yr⁻¹ so that the negative numbers are visible; (b) Same as 8a) but for the Arctic Ocean (ocean area north of the Bering Strait on the Pacific side and north of 70°N on the Atlantic side) for the same time periods as in (a).**


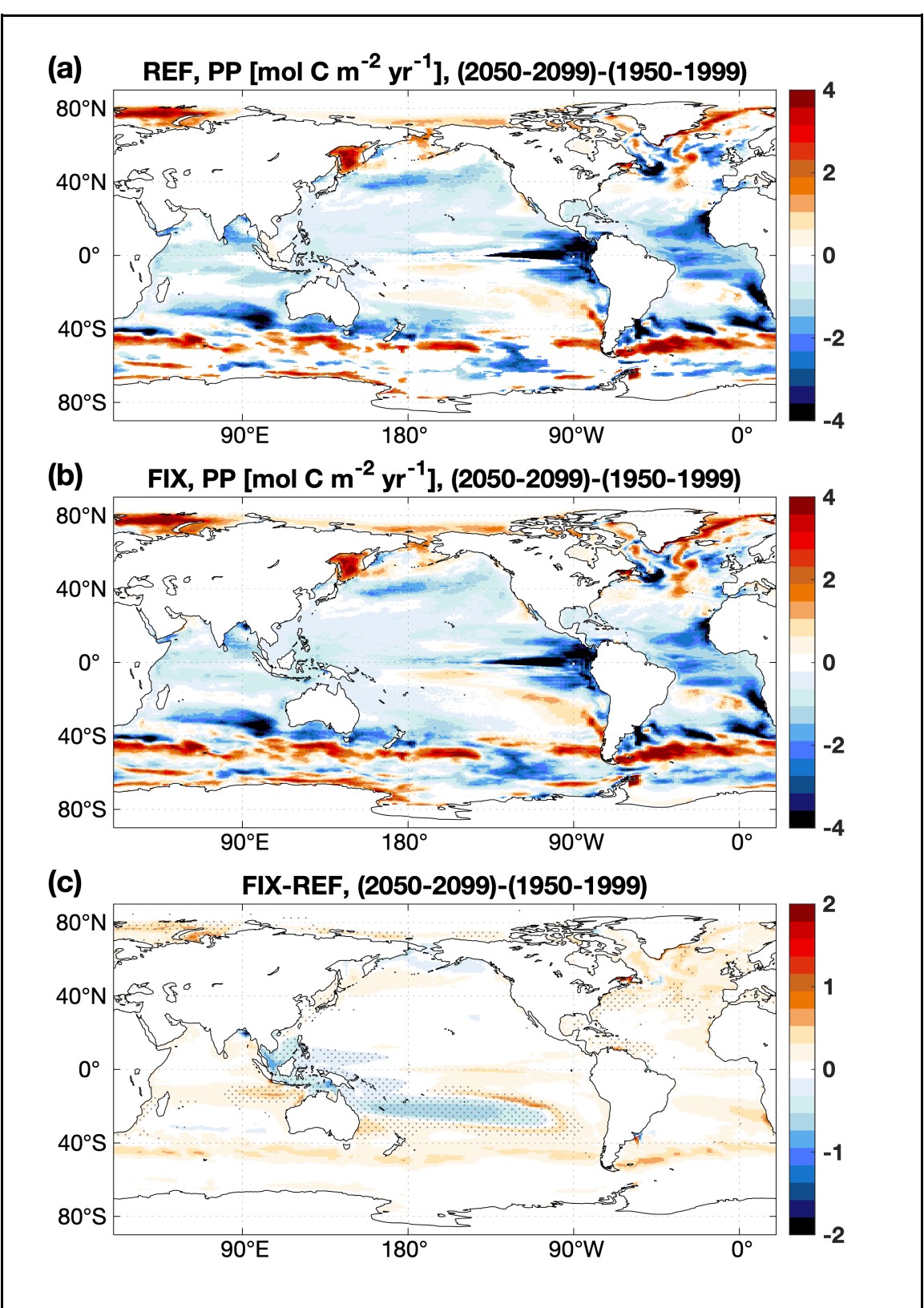

**Figure 6: The difference in vertically integrated primary production between 2050–2099 and 1950–1999 time periods in (a) REF, (b) FIX, and (c) the difference between (b) and (a). In panel c, only significant differences are plotted, and dots denote areas where the signal is larger than the standard-deviation of the absolute field (see details in Appendix B).**

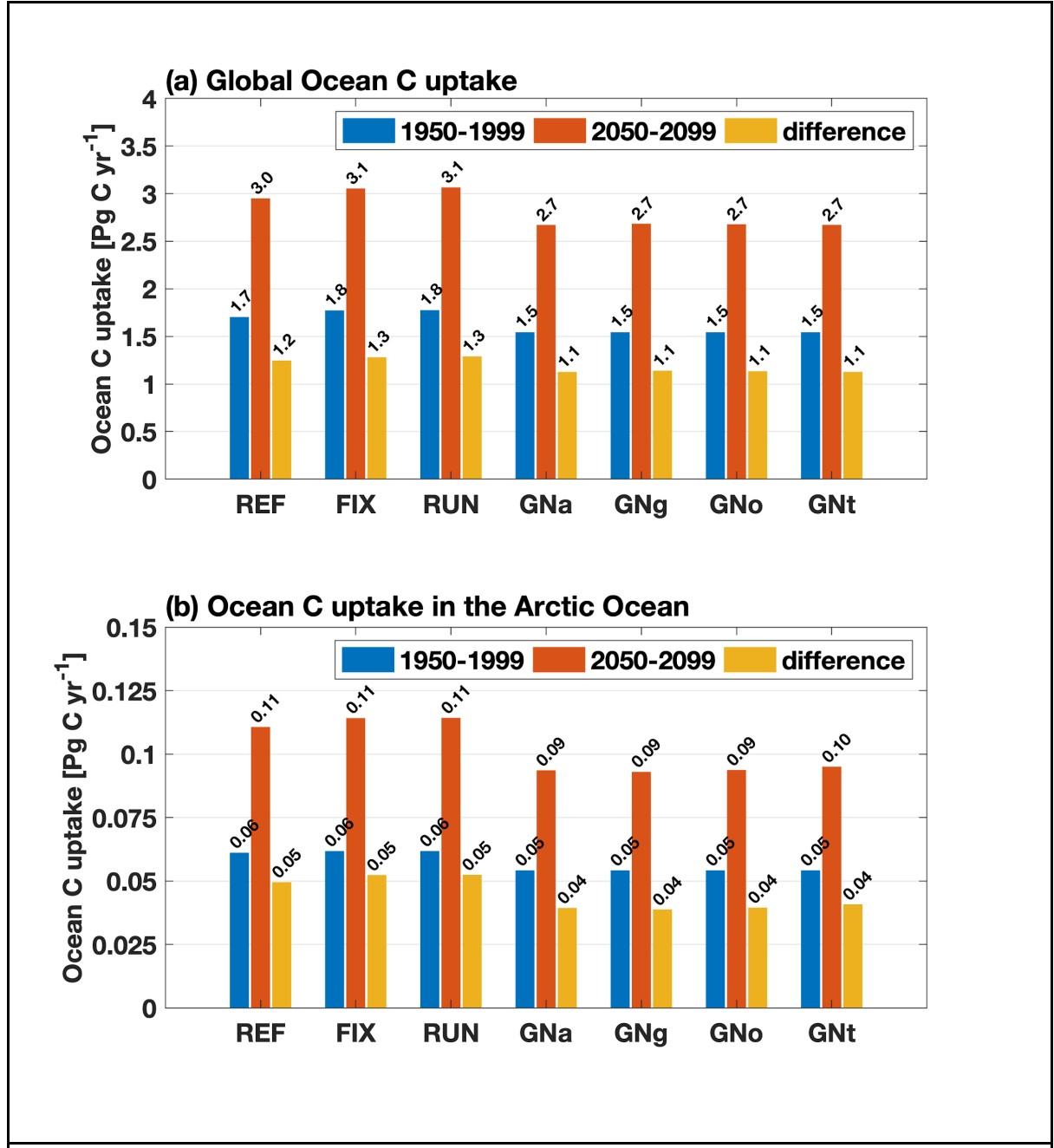

**Figure 7: (a) Globally integrated annual mean ocean carbon uptake over 1950–1999 (blue), over 2050–2099 (red) and the differences between these two time periods (yellow) for all experiments; (b) Same as (a) but for the Arctic Ocean (ocean area north of the Bering Strait on the Pacific side and north of 70°N on the Atlantic side) for the same time periods as in (a).**

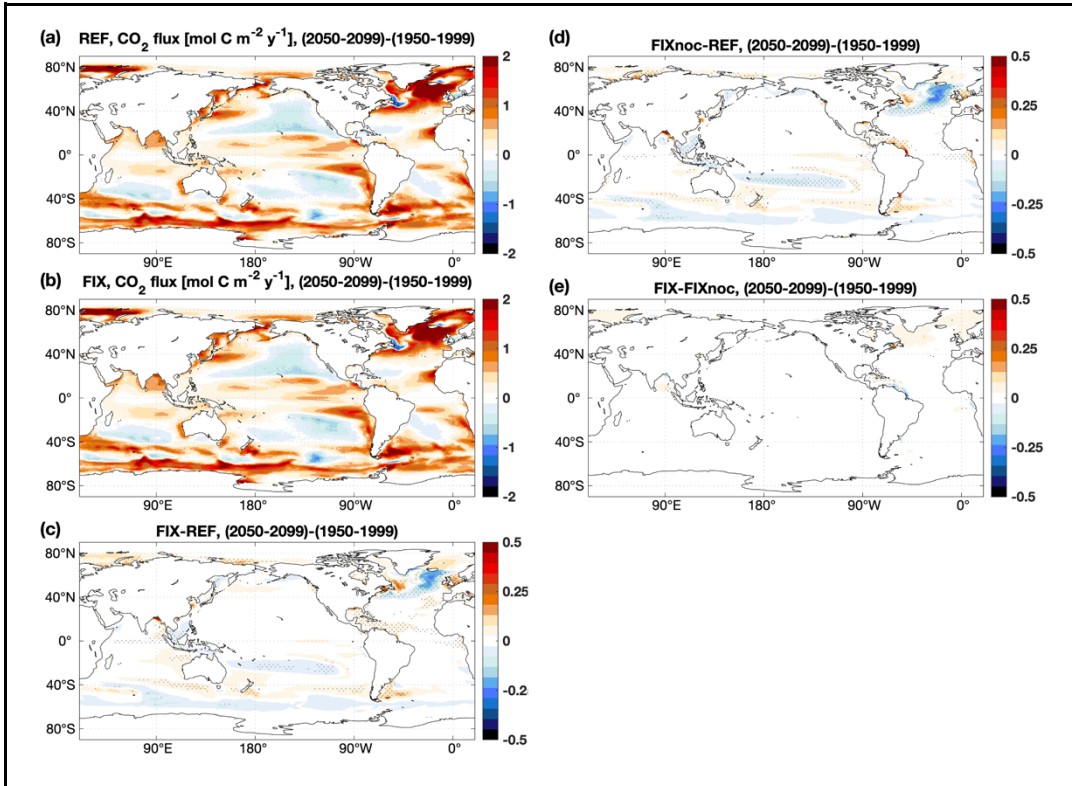

**Figure 8: Differences in annual mean air-sea CO₂ fluxes (mol C m⁻² yr⁻¹) between 2050–2099 and 1950–1999 periods in (a) REF, (b) FIX, (c) the difference between FIX and REF, (d) the difference between FIXnoc and REF, and (e) the difference between FIX and FIXnoc. In panels c-e, only significant differences are plotted, and dots denote areas where the signal is larger than the standard-deviation of the absolute field (see details in Appendix B).**


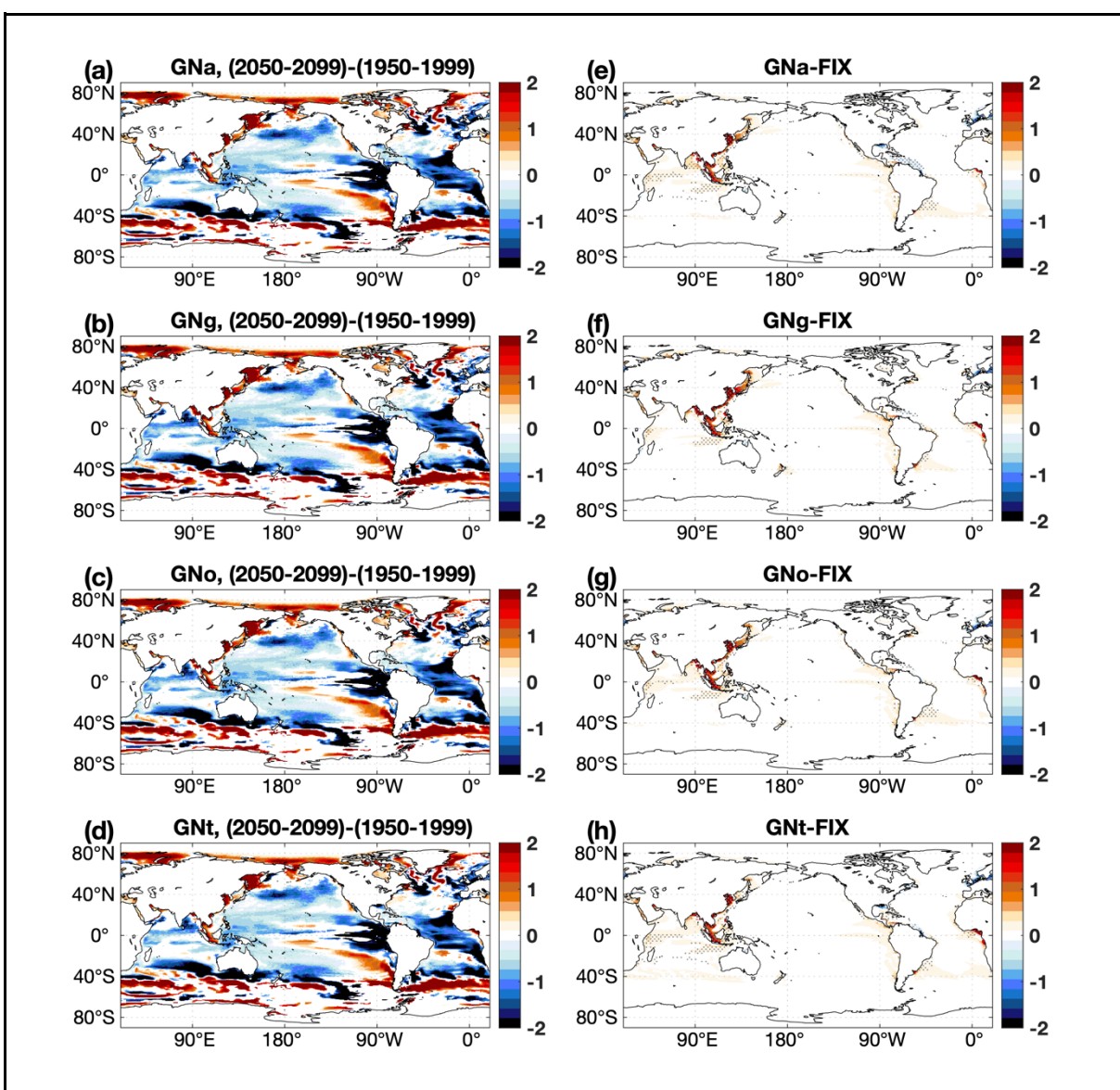

**Figure 9: (a-d) Projected changes in vertically integrated primary production (mol C m⁻² yr⁻¹) in four GNS experiments between 2050–2099 and 1950–1999 periods; (e-h) The difference in projected changes in vertically integrated primary production (mol C m⁻² yr⁻¹) between each GNS experiment and FIX. In panels e-h, only significant differences are plotted, and dots denote areas where the signal is larger than the standard-deviation of the absolute field (see details in Appendix B).**


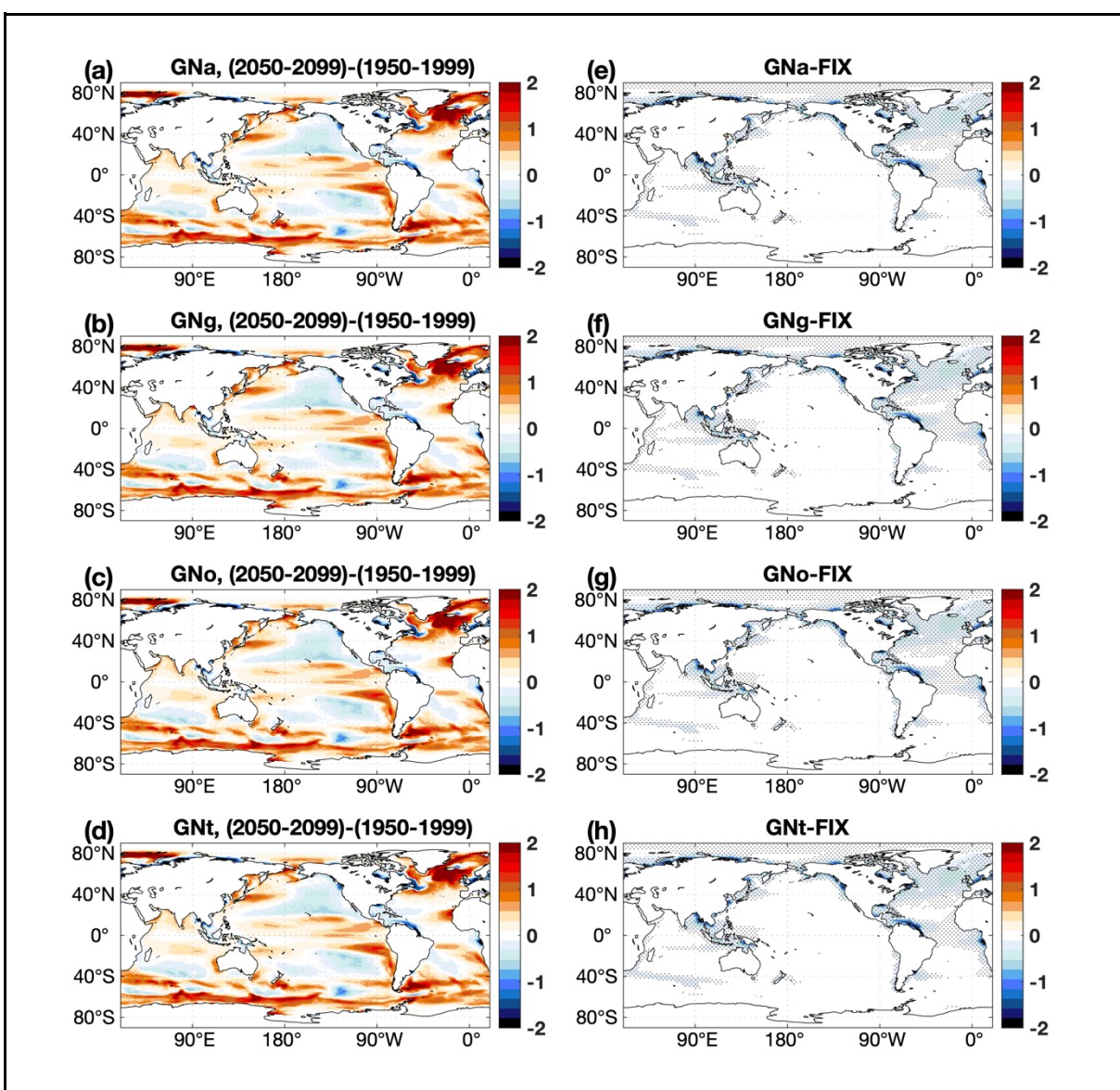

**Figure 10: Projected changes in annual mean air-sea CO$_2$ fluxes (mol C m$^{-2}$ yr$^{-1}$) in four GNS experiments between 2050–2099 and 1950–1999 periods; (e-h) The difference in projected changes in annual mean air-sea CO$_2$ fluxes (mol C m$^{-2}$ yr$^{-1}$) between each GNS experiment and FIX. In panels e-h, only significant differences are plotted, and dots denote areas where the signal is larger than the standard-deviation of the absolute field (see details in Appendix B).**


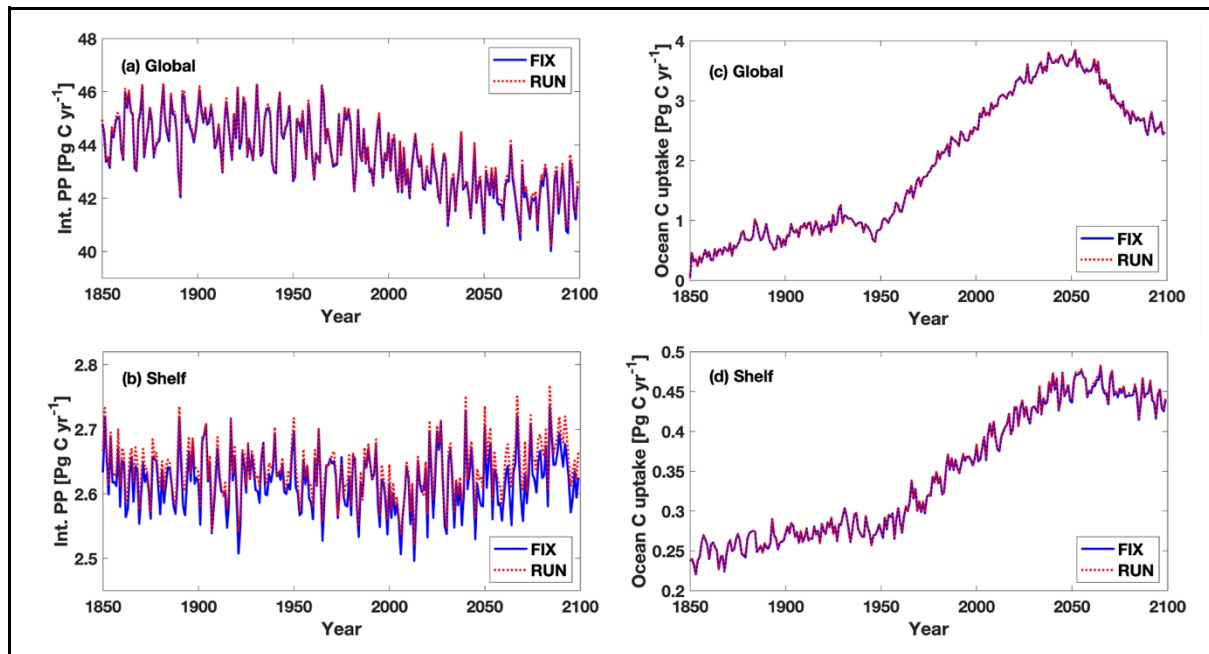

**Figure 11: Time-series of integrated annual primary production and ocean carbon uptake during 1850–2099 in FIX and RUN (a, c) globally and (b, d) on continental shelves.**

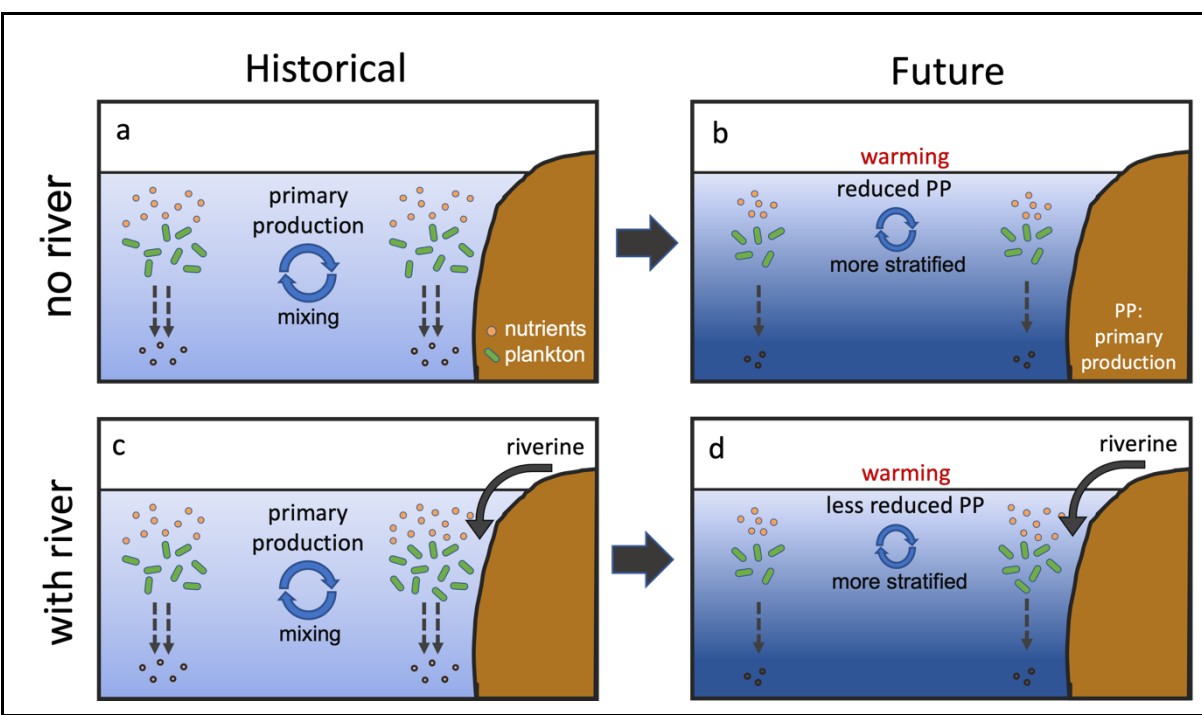

**Figure 12: Schematic drawing of impact of riverine nutrients input on future projections of marine primary production. (a, b) Decline in nutrients supply into subtropical surface waters, due to the upper ocean warming and increased vertical stratification, which is projected by models to reduce primary production over the 21st century. (c, d) Riverine nutrients input into surface coastal waters alleviates the nutrient limitation and lessen the projected future decline in primary production.**

**Table 1. Brief introduction to future scenarios for river nutrient export used in Global NEWS 2 (Seitzinger et al.,**
**2010)**

| Scenario | Agricultural trends | Sewage |
|---|---|---|
| Adapting Mosaic (GNa) a world with a focus on regional and local socio-ecological management | -medium productivity increase -2% of cropland area for energy crops -fertilizer efficiency: moderate increase in N and P fertilizer use in all countries; better integration of animal manure and recycling of human N and P from households with improved sanitation but lacking a sewage connection | -constant fraction of population with access to sanitation and sewage connection -moderate increase in N and P removal by wastewater treatment |
| Global Orchestration (GNg) a globalized world with an economic development focus and rapid economic growth | -high productivity increase -4% of cropland area for energy crops -fertilizer efficiency: no change in countries with a soil nutrient surplus; rapid increase in N and P fertilizer use in countries with soil nutrient depletion | -towards full access to improved sanitation and sewage connection -rapid increase in N and P removal by wastewater treatment |
| Order from Strength (GNo) a regionalized world with a focus on security | -low productivity increase -1% of cropland area for energy crops -fertilizer efficiency: no change in countries with a soil nutrient surplus; moderate increase in N and P fertilizer use in countries with soil nutrient depletion | -same as GNa |
| Technogarden (GNt) a globalized world with a focus on environmental technology | -medium-high productivity increase -28% of cropland area for energy crops -fertilizer efficiency: rapid increase in N and P fertilizer use in countries with a soil nutrient surplus; rapid increase in countries with soil nutrient depletion | -same as GNg |
