# Peer review of "Riverine impact on future projections of marine primary 1"

_Biogeosciences, 2021_

## Author Comment (AC1)

**Reply to Anonymous Referee #2**

**We are grateful for the comments and edits of the anonymous reviewer who invested time for the review of our manuscript. The reviewer's text is reported in Italic and our responses in roman.**

*Using a global ocean biogeochemical model, the authors assess the future impacts of changing riverine inputs while performing simulations with several scenarios of riverine inputs. In the paper, the authors show a slight reduction in root mean square error of primary production with respect to observations on the continental shelf and in parts of the ocean highly affected by rivers. They show that, overall, riverine nutrients may alleviate part of projected primary production decline. The topic is very relevant and potentially important for fully constraining the ocean carbon cycle. However, while the modelling work is very sound in its closed framework, I believe the author's review of literature seems to be lacking, which leads to, at times, questionable assumptions and too high confidence in a model setting not necessarily adequate for investing impacts of riverine tracers and their fate in the ocean. I do think the study should be published, but the authors should discuss the following points and, in my opinion, clearly point out uncertainties related to their framework in the abstract.*

*General Comments*

*It is debatable that the model configuration used here is actually adequate for the research question addressed in this paper. The model is limited in terms of resolution (~1 degree), and this is a strong constraint for representing fine-scale circulation features that are thought to be of particular importance in the coastal ocean. This is a topic that many studies have discussed previously, and these should be considered. Secondly, the model doesn't consider specific biogeochemical processes relevant to the coastal ocean. For instance, organic matter decomposition rates are shown to be much higher in the coastal sediment than in the open ocean (Ardnt et al., 2013), bed shear stress also re-suspend biogeochemical from the shelf seafloor. I think omitting both of these physical and biogeochemical limitations lead to the important underestimation of primary production on river-dominated continental shelves shown in this study, which seems to indicate the model is underrepresenting. This also affect exports of riverine biogeochemical compounds to the open ocean, and thus this has very important consequences for the main outcomes presented in this study. The authors do mention these limitations briefly in the limitations section, but this should be considered omni-present in attempts to interpretate the results, and at the least mentioned in the abstract.*

Thank you very much for the comment. We are fully aware that in a coarse resolution model, not all physical and biogeochemical processes can be reproduced in detail and this has been already mentioned in the limitations section. However, we think that the study is useful nevertheless, in order to see the effect of an addition (or omission) of respective land-sea fluxes of chemical compounds within the coarse resolution (~1 degree) context. Especially since such models are still regularly employed in IPCC reports on projecting future ocean carbon cycle response to climate change. We will clarify that the goal of this study is not to simulate the distributions of geochemical tracers near the coast correctly, but rather to assess the large-scale impact of adding another layer of (relatively simplistic) continental margin-to-open ocean

biogeochemical process. Further, to explore the best practical way of implementing the riverine inputs for modelling groups is also one of the aims of this work.

We plan to carry out two amendments, one in the abstract and one in the main text.

We will change the last sentence of the abstract to: "Simulations with high-resolution global or regional models with an adequate representation of shelf processes are required to accurately assess the impact of future riverine scenarios."

At the end of the introduction, we will add at line 69: "Because of the coarse resolution of the model, a series of processes in the coastal zone cannot be represented in our study such as the high accumulation of organic sediment in shallow waters and respective remineralisation rates of previously deposited material (Ardnt et al., 2013; Regnier et al., 2013). These processes can only be presented in a model of much higher spatial resolution, which on the other hand cannot be integrated long enough to simulate the large scale water masses adequately and project long-term scale climatic change."

*A second, perhaps less central point, but still relevant to the study, is that the authors spin-up their model to present day fluxes, whereas these are actually more strongly perturbed over the historical time period (Beusen et al., 2016), than what is projected in terms of their future changes. Since time-scales of the ocean carbon cycle are notably long, this historical perturbation could have important legacy effects propagating into the future, potentially enhancing the primary production more than is estimated here. This should, in my opinion, also be discussed in the limitations section.*

Thank you for the comment. In the revised manuscript, we will be more clear about using the recent-past riverine fluxes, i.e., constant fluxes at 1970's level, to spin-up the model, rather than present day fluxes. Beusen et al. (2016) have shown in their Figure 3 that the nutrient fluxes did not vary much before 1970. However, using constant riverine fluxes rather than transient fluxes to spin-up the model can potentially have legacy effects on the results. We will discuss this in the limitations section.

*Specific Comments:*

*Abstract*

- *L16 "With four riverine configurations: deactivated, fixed at a contemporary level, coupled to simulated freshwater runoff, and following four plausible future scenarios." Are only the nutrients (and if yes which ones) changing, or also carbon and alkalinity? This should be stated here.*

We will change it to "...with four riverine transport configurations for nutrients (nitrogen, phosphorus, silicon and iron), carbon and total alkalinity: deactivated, fixed at a contemporary level, coupled to simulated freshwater runoff, and following four plausible future scenarios."

- *L17 "The inclusion of riverine nutrients and carbon…" Those numbers are valid for contemporary I guess?*

We will change it to "The inclusion of riverine nutrients and carbon at 1970's level improves the modelled contemporary spatial distribution relative to the observations…"

- *L20 "Riverine nutrient inputs alleviate nutrient limitation,…" Should be reformulated, since riverine nutrient inputs are unlikely alleviated nutrient limitation in general, but reduce (?) it in some regions (?).*

We will change it to "Riverine nutrient inputs lessen nutrient limitation under future warmer conditions as stratification increases, and thus lessen the projected future decline in PP…". It is not feasible to put detailed regions in the abstract due to word limit, but we have explained the regions in detail in section 3.2.

*Introduction*

- *In general, there are very little citations in the introduction, and often the same ones are used repeatedly. There are some recent modeling studies of implications of riverine inputs in the ocean that would be very relevant for this study. These should, in my opinion, be considered in the introduction:*

*Lacroix, F., Ilyina, T., Mathis, M., Laruelle, G. G., & Regnier, P. (2021). Historical increases in land-derived nutrient inputs may alleviate effects of a changing physical climate on the oceanic carbon cycle. Global Change Biology, 27, 5491– 5513. https://doi.org/10.1111/gcb.15822*

*Liu, X., Stock, C. A., Dunne, J. P., Lee, M., Shevliakova, E., Malyshev, S., & Milly, P. C. D. (2021). Simulated global coastal ecosystem responses to a half-century increase in river nitrogen loads. Geophysical Research Letters, 48, e2021GL094367. https://doi.org/10.1029/2021GL094367*

*I would furthermore suggest citing some regional-scale studies that investigate implications of riverine inputs on biogeochemistry of specific shelves, literature is abundant here. In addition, I would read and refer to the last 2-3 Global Carbon Budget studies for potential importance of riverine carbon fluxes for the ocean.*

Thank you very much for the suggestion and the references. We will include the above mentioned literature as well as some other ones (e.g. Tivig et al., 2021), and will also add more references on regional studies, e.g. Arctic (Siberia's river basin) and Amazon river estuary (Drake et al., 2021).

- *L25 "The large range of the riverine input across our four riverine 26 configurations does not transfer to a large uncertainty of the projected global PP and ocean C uptake…" In terms of global PP, one could argue this could be due to the representation of continental shelf in the model, which leads to heavily underestimated continental shelf PP.*

We agree completely with the reviewer on this point and we will change this sentence to "Simulations with high-resolution global or regional models with an adequate representation of shelf processes are required to accurately assess the impact of future riverine scenarios."

We are fully aware of the underrepresented shelf process issue and the underestimated coastal PP in coarse-resolution models. We have discussed those issues in section 4.3 (Firstly, poorly represented physical shelf processes, as well as uncertainties in biogeochemical dynamic. For example, conversion of organic to inorganic carbon occurs rapidly via remineralization in estuaries before they are transported to the open ocean. Secondly, coarse-resolution models tend to underestimate primary production along the coast. Such well-known model issues may limit the impact induced by riverine inputs). We have pointed out in section 4.2 that "However, the scenario differences might be of importance in regional projections, such as in seas surrounded by highly populated nations and/or near river estuaries. Simulations with high-resolution global or regional models with an adequate representation of shelf processes are required to accurately assess the local impact of riverine inputs."

- *L35 "Although riverine carbon only plays a minor role in the global carbon cycle, …" Recent Global Carbon Budget publications disagree with this (Friedlingstein et al., 2021). If the higher estimates of outgassing of riverine carbon are true (up to 0.8 Pg C yr-1), they could potentially play a large role in explaining discrepancies between CO2 estimates arising observation-based products and model-based results.*

We will change this sentence to "Despite our limited understanding on the riverine carbon fluxes, they could play an important role in closing the global carbon budget (Friedlingstein et al., 2021) and could be very sensitive to regional and global changes such as weathering, land cover and climate (Meybeck and Vörösmarty, 1999)."

- *L44 Maybe add the more recent Beusen et al. (2016) estimates to this for the historical time period?*

We will add the following sentence in the revised version: "Beusen et al. (2016) estimated that river nutrient transport to the ocean increased from 19 to 37 Tg N $yr^{-1}$ and from 2 to 4 Tg P $yr^{-1}$ over the 20th century, taking into account of both increased nutrient input to rivers and intensified retention/removal of nutrients in freshwater systems."

*Methods*

- *L118 "The riverine influx includes carbon, nitrogen and phosphorus, each in dissolved inorganic, dissolved organic, and particulate forms, as well as alkalinity (ALK), dissolved silicon and iron (Fe)." Are there specific ocean variables for terrestrial dissolved and particulate organic matter? If not how does not model deal with organic P-N-C ratios that differ from those of the Redfield ratio? This is an important point to clarify because high C-to-nutrient ratios are thought to be largely responsible for ocean outgassing.*

In the model there is one dissolved organic pool (DOC) and one particulate organic pool (DET, detritus). First, we calculate the riverine organic P-N-C ratios for both dissolved and particulate forms, then add the least abundant species (scaled by the open ocean Redfield ratio) to the DOC and DET pools, respectively. The excess budget from the remaining two species both in dissolved and in particulate forms are assumed to be directly remineralized into inorganic form and added to the corresponding dissolved inorganic pools (i.e., PO4, NO3, and DIC) in the ocean. We will elaborate on this, based on the text in line 137-140, in the revised version.

- *L140 "Any remaining riverine organic matter is then added to its inorganic pool" This is not really clear. Is excess organic carbon is added to the DIC pool? If not I think this might be the reason why river inputs cause a net sink in the model, and not source as is relatively well acknowledged (see e.g., Global Carbon Budget, 2021). Also keep in mind that organic carbon mineralization has a small effect on alkalinity (which I don't think would have a huge impact here).*

Please see the response to the last point. Excess DOC is indeed added to the DIC pool, but yet the riverine fluxes in the model lead to a net C sink, which might be potentially due to the overestimation of the organic-to-inorganic conversion of excess nutrients.

*Also, maybe more important here: are you assuming the large particulate fluxes (particulate P and N) from NEWS2 are organic? Because from my understanding, these can be inorganic (for P bedrock erosion, occluded etc..), and this would not at all be bio-available in the coastal ocean.*

Although the Global NEWS2 data provide the total particulate N and P rather than differentiated inorganic and organic particulate forms, particulate N occurs largely as organic matter while particulate P is typically dominated by inorganic forms (Mayorga et al., 2010). The reviewer is completely correct in stating that particulate P is mostly inorganic and not directly bio-available. Thus, adding the remaining particulate P (after calculating the least abundant species according to the Redfield ratio) into dissolved inorganic P pool may lead to bias in the enhanced primary production. Along the same line, adding the remaining riverine dissolved organic matter into the corresponding dissolved inorganic pool may also partly lead to bias in the enhanced primary production. Therefore, we have calculated the upper range of the impact of the directly remineralized dissolved organic and particulate matter on the enhanced primary production, by comparing with the corresponding riverine dissolved nutrients [X/(X+DIXriv)*100%] (X is the directly remineralized dissolved organic and particulate matter). Assuming that all coastal regions are nutrient limited, this direct remineralization is responsible for approximately 33.3-80.5% of the enhanced primary production.

We will add a paragraph in the discussion on this point in the revised version.

- *L156 "REF: Reference run. Riverine nutrient and carbon supply is deactivated." Are there other sources of nutrients and carbon in the model? If riverine nutrients and carbon were the only inputs to the ocean model and their sediment loss is non-zero, I would expect all related variables to thrive to zero, which does not make a very*

*interesting reference run. In the case there are other inputs, they should be given in numbers and explained.*

The only external inputs of nutrients are from aerial dust (iron) deposition and nitrogen fixation.

The REF simulates primary production and ocean $CO_2$ uptake evolution under climate change, without riverine input. It is interesting to see that the effect of riverine inputs on primary production is different between the historical and future time period (due to a different nutrient depletion level). This assessment is only possible to make by subtracting the climate effect on primary production in REF.

- *L179-L185 In my opinion the authors don't need to specifically defend themselves on this particular point, at least not to this extent. I would consider shortening or removing.*

We prefer to keep it unshortened, but will separate it from the other content of the section. Following the suggestion of the other reviewer to add more information on the statistical significance of our results, we plan to add a new sub-section on statistical robustness, where we will include this part and additional signal-to-noise assessment that measures the riverine impact against the magnitude of simulated inter-annual ocean biogeochemistry variability. For more details, please see our response to the other reviewer's comments on the significance of the results.

*Results*

- *L198 "Although the total PP in FIX is still considerably lower than the satellite-based estimates, the inclusion of riverine nutrients and carbon does slightly improve the distribution of PP especially on continental margins (Figure 3), according to our area-weighted root mean square error (RMSE) analysis. „*

*Figure 3 really shows that a large part of the underestimation of PP is originating from the continental shelf, in particular regions of riverine inputs. The improvement is minor compared to the actual bias. In my opinion, this actually shows that the model underestimates the impacts of rivers on PP, which does have a strong implication for the conclusions of this paper, and should be assessed somehow.*

We agree with the reviewer that the riverine nutrient input (at 1970's level) does not significantly improve the simulated contemporary primary production along the continental shelf. The reasons could be:

1) The underestimation of primary production on continental margins due to coarse model resolution and unresolved shelf processes is larger than the impact of riverine nutrient input. Therefore, it could not be compensated.

2) The time period of contemporary primary production that we look into is 2003-2012, but the riverine nutrient input in the model is at 1970's level. Beusen et al. (2016) have shown that the riverine nitrogen and phosphorus has increased by ~40.0% and 28.6% from 1970 to 2000. Therefore, the riverine impact might be higher.

We will add this into discussion.

*Figure 4: It's a bit concerning to me that considering riverine inputs lead to a sink of carbon in the ocean. It is relatively well acknowledged that river inputs are thought to cause a source of carbon (e.g., Regnier et al., 2013; Resplandy et al., 2018). The reason for this is that carbon to nutrient ratio of the (bio-available) terrestrial inputs is larger than the Redfield ratio. I guess the fact that most particulate P and N is thrown in as dissolved inorganic species might be the explanation for this. How is the alkalinity to DIC ratio of riverine inputs constrained? Either way, either explain the reason for this or I would consider not discussing the CO2 flux for the "unperturbed" river simulation.*

Firstly, the riverine DIC to alkalinity ratio that we have applied is 1:1.

Our model shows the ocean as a weak carbon sink when including riverine, which can be partly due to our assumption that the excess dissolved organic and particulate matter immediately becomes remineralized bioavailable inorganic nutrients from rivers. This is because we have only one dissolved organic pool (DOC) and one particulate organic pool (DET) in the model, and the Redfield ratio needs to be kept. We have assessed the impact of this approach on our results (please see the response to the comments in the Method section).

Borges and Frankignoulle (2005) stated that "marginal seas act as a strong sink of $CO_2$ of about $-0.45$ Pg C $yr^{-1}$. This sink could be almost fully compensated by the emission of $CO_2$ from the ensemble of near-shore coastal ecosystems of about 0.40 Pg C $yr^{-1}$. Although this value is subject to large uncertainty, it stresses the importance of the diversity of ecosystems, in particular near-shore systems, when integrating CO2 fluxes at global scale in the coastal ocean." Chen and Borges (2009) demonstrated that " the available data of pCO2 measurements in about 60 continental shelves of the world allows the conclusion that continental shelves are indeed sinks for atmospheric carbon….The concept of marginal seas as sinks and near-shore coastal ecosystems as sources of atmospheric CO2 allows reconciling diverging views on carbon cycling in the coastal ocean. The fact that the inputs of terrestrial/riverine organic carbon would be in excess of carbon burial in marine sediments does not necessarily imply a net heterotrophy of marginal seas that is in contradiction with the high offshore export rates of POC and DOC consistently reported across continental margins. …Hence, inner estuaries and near-shore ecosystems are effective filters for terrestrial/riverine organic inputs and impose a by-pass of carbon towards the atmosphere for the global carbon cycle."

In our model, the shelf processes are not well represented due to model resolution. However, the processes that the riverine input of nutrients and carbon causes CO2 outgassing near coastal regions and CO2 ingassing on continental shelves lead to an globally integrated overall net weak carbon sink in the continental margins in the model. Simulations with high-resolution global or regional models with a good representation of shelf processes are required to accurately assess impact of riverine inputs on carbon cycling in the coastal ocean.

We will summarise this into a paragraph and add it to discussion.

- *L305 "Our experiments show that riverine nutrient inputs have a dominant role over the organic matter inputs in FIX, enhancing CO2 uptake along continental margins via sustaining PP in both historical and future time periods." This is however purely a consequence of the ratio of (bio-available) nutrients to organic matter that is added to the ocean, which as mentioned, I don't think is completely correct from a process-based perspective. In fact, I think most river-dominated shelves show C outgassing, see regional CO2 fluxes from Chen and Borges (2009) or regional-based studies.*

Please see the response to the last comment.

- *L337 "…do not transfer to large uncertainties in future global marine biogeochemistry projections in NorESM." Yes, but if you would have taken uncertainties related to the coastal ocean into account, through e.g. sensitivity analysis of sediment degradation, I wonder if the conclusions would be different here, I would assume so.*

We will change the sentence to "A large range of the riverine inputs in GNS, e.g., temporal changes in DIN fluxes across scenarios ranging 24.8-63.0% of the annual flux in FIX, do not transfer to large uncertainties in future projections of global marine primary production in our model, which can be primarily attributed to unresolved shelf processes due to coarse model resolution."

*Minor edits*

- *L22 "and thus lessen the projected future decline in PP by up to 0.6 PgC yr-1 22 (27.3%) globally depending on the riverine configuration." -> , globally,*

We will change it accordingly.

- *L55 "Taking the advantage of the latest improvement of global" Remove "the".*

We will change it accordingly.

- *L171 "By comparing FIX versus REF…" Add comma here.*

We will change it accordingly.

- *L332 ". Therefore, it is worth exploring the merits of using GNS in future projections of marine biogeochemistry." Not really sure what is meant here.*

This sentence was to express that using future scenarios of transient riverine fluxes can be explored by future modelling studies on projection of marine biogeochemistry, especially on high-resolution regional scales. We will remove this sentence in the revised version, since it has been explained at the end of this paragraph (Line 338-340).

**Reference**

Arndt S., B.B. Jørgensen, D.E. LaRowe, J.J. Middelburg, R.D. Pancost, P. Regnier,Quantifying the degradation of organic matter in marine sediments: A review and synthesis,Earth-Science Reviews, Volume 123, 2013, Pages 53-86, ISSN 0012-8252, https://doi.org/10.1016/j.earscirev.2013.02.008.

Beusen, A. H. W., Bouwman, A. F., Van Beek, L. P. H., Mogollón, J. M., and Middelburg, J. J.: Global riverine N and P transport to ocean increased during the 20th century despite increased retention along the aquatic continuum, Biogeosciences, 13, 2441–2451, https://doi.org/10.5194/bg-13-2441-2016, 2016.

Borges, A. V., Delille, B., and Frankignoulle, M.: Budgeting sinks and sources of CO2 in the coastal ocean: Diversity of ecosystems counts, Geophysical Research Letters, 32, https://doi.org/10.1029/2005GL023053, 2005.

Chen C.-T. A., Borges A.V., Reconciling opposing views on carbon cycling in the coastal ocean: Continental shelves as sinks and near-shore ecosystems as sources of atmospheric CO2, Deep Sea Research Part II: Topical Studies in Oceanography, Volume 56, Issues 8–10, 2009, Pages 578-590, https://doi.org/10.1016/j.dsr2.2009.01.001.

Drake, T. W., Hemingway, J. D., Kurek, M. R., Peucker-Ehrenbrink, B., Brown, K. A., Holmes, R. M., Galy, V., Moura, J. M. S., Mitsuya, M., Wassenaar, L. I., Six, J., and Spencer, R. G. M.: The Pulse of the Amazon: Fluxes of Dissolved Organic Carbon, Nutrients, and Ions From the World's Largest River, Global Biogeochemical Cycles, 35, e2020GB006895, https://doi.org/10.1029/2020GB006895, 2021.

Friedlingstein, P., Jones, M. W., O'Sullivan, M., Andrew, R. M., Bakker, D. C. E., Hauck, J., Le Quéré, C., Peters, G. P., Peters, W., Pongratz, J., Sitch, S., Canadell, J. G., Ciais, P., Jackson, R. B., Alin, S. R., Anthoni, P., Bates, N. R., Becker, M., Bellouin, N., Bopp, L., Chau, T. T. T., Chevallier, F., Chini, L. P., Cronin, M., Currie, K. I., Decharme, B., Djeutchouang, L., Dou, X., Evans, W., Feely, R. A., Feng, L., Gasser, T., Gilfillan, D., Gkritzalis, T., Grassi, G., Gregor, L., Gruber, N., Gürses, Ö., Harris, I., Houghton, R. A., Hurtt, G. C., Iida, Y., Ilyina, T., Luijkx, I. T., Jain, A. K., Jones, S. D., Kato, E., Kennedy, D., Klein Goldewijk, K., Knauer, J., Korsbakken, J. I., Körtzinger, A., Landschützer, P., Lauvset, S. K., Lefèvre, N., Lienert, S., Liu, J., Marland, G., McGuire, P. C., Melton, J. R., Munro, D. R., Nabel, J. E. M. S., Nakaoka, S. I., Niwa, Y., Ono, T., Pierrot, D., Poulter, B., Rehder, G., Resplandy, L., Robertson, E., Rödenbeck, C., Rosan, T. M., Schwinger, J., Schwingshackl, C., Séférian, R., Sutton, A. J., Sweeney, C., Tanhua, T., Tans, P. P., Tian, H., Tilbrook, B., Tubiello, F., van der Werf, G., Vuichard, N., Wada, C., Wanninkhof, R., Watson, A., Willis, D., Wiltshire, A. J., Yuan, W., Yue, C., Yue, X., Zaehle, S., and Zeng, J.: Global Carbon Budget 2021, Earth Syst. Sci. Data Discuss., 2021, 1-191, 10.5194/essd-2021-386, 2021.

Mayorga, E., Seitzinger, S. P., Harrison, J. A., Dumont, E., Beusen, A. H. W., Bouwman, A. F., Fekete, B. M., Kroeze, C., and Van Drecht, G.: Global Nutrient

Export from WaterSheds 2 (NEWS 2): Model development and implementation, Environmental Modelling & Software, 25, 837-853, https://doi.org/10.1016/j.envsoft.2010.01.007, 2010.

Meybeck, M. and Vörösmarty, C.: Global transfer of carbon by rivers, Global Change Newsletter, 37, 18-19, 1999.

Regnier, P., P. Friedlingstein, P. Ciais, F.T. Mackenzie, N. Gruber, I.A. Janssens, G.G. Laruelle, R. Lauerwald, S. Luyssaert, A.J. Andersson, S. Arndt, C. Arnosti, A.V. Borges, A.W. Dale, A. Gallego-Sala, Y. Goddéris, N. Goossens, J. Hartmann, C. Heinze, T. Ilyina18, F. Joos, D.E. LaRowe, J. Leifeld, F.J. R. Meysman, G. Munhoven, P. A. Raymond, R. Spahni, P. Suntharalingam, and M. Thullner, 2013, Anthropogenic perturbation of the carbon fluxes from land to ocean, *Nature Geoscience*, 6(8), 597-607. DOI: 10.1038/NGEO1830

Tivig, M., Keller, D. P., and Oschlies, A.: Riverine nitrogen supply to the global ocean and its limited impact on global marine primary production: a feedback study using an Earth system model, Biogeosciences, 18, 5327-5350, 10.5194/bg-18-5327-2021, 2021.

---

## Author Comment (AC2)

**Reply to Anonymous Referee #1**

**We are grateful for the comments and edits of the anonymous reviewer who invested time for the review of our manuscript. The reviewer's text is reported in Italic and our responses in roman.**

*The research topic is very timely, the manuscript is well written, and the model simulates seem to be well done. Furthermore, the authors have really made a lot of simulations and obviously went to great lengths. However, many things need to be addressed before the paper can be accepted. The most important points are: (a) Discussing Lacroix et al. (2021), (b) adding many more references to support the text, (c) evaluating the model results with observations and show where the model is good or bad, (d) evaluating the global news present-day river fluxes to understand how realistic they are, (e) discussing the far too low NPP in the Arctic Ocean, and (f) testing if results are statistically significant. In this state, the main conclusion that rivers might be of importance in coastal region has no underlying prove and large parts of the manuscript are not possible to review. I would focus more on regional coastal areas and discuss and analyze these further, but this is up to you.*

*Major comments*

- *References are often missing. One of the most important references missing is probably Lacroix et al. (2021) who performed very similar simulations. Comparison to their results would be essential. Please revise the entire manuscript for the many missing references. Here some examples:*
  o *Lines 30/31: no references for this statement*

This sentence together with the sentence after is one complete statement, and the references are therefore after the second sentence in Line 32.

  o *Lines 33/34: no references for this statement*

Reference (Chester, 2012) will be added.

  o *Lines 34/35: no references for this statement*

This is our own statement.

  o *Lines 37-48: Almost no references here!*

Line 37-38: Our own statement.

Line 39-41: Reference (Seitzinger et al., 2010) will be added.

Line 41-44: Reference was given.

Line 44-46: Reference (Meybeck and Vörösmarty, 1999) will be added.

  o *Lines 49/50: No references*

That statement is according to our own knowledge, and it is supported by examples written in Line 50-55.

○    …
•    *Lines 49-54: A more detailed explanation is missing on how ESMs simulate rivers. Seferian et al. (2020) gives a good overview. There were already 5 ESMs in CMIP5 that simulated riverine C, N, and P (3 also simulated Fe) fluxes and there are now 8 in CMIP6. Some of them even simulate dynamically changing riverine fluxes.*

We will add more information on the riverine implementation, especially with respect to new developments in some of the CMIP6 models (e.g. CESM2, CNRM-ESM2-1, MIROC-ES2L, IPSL-CM6A-LR):

"The latest generation of ESMs have implemented some forms of riverine inputs in their ocean biogeochemistry modules (Seferian et al., 2020). In general, those that implement riverine inputs, do it differently from constant contemporary fluxes based on data from GlobalNEWS (IPSL-SM6A-LR and CESM2; Aumont et al., 2015; Danabasoglu et al., 2020) to fully interactive with terrestrial nutrient leaching transported by dynamical river routing (CNRM-ESM2-1 and MIROC-ES2L; Seferian et al., 2019; Hajima et al., 2020). Models with interactive riverine transports do not consider biogeochemical processes in the freshwater, hence tracers are treated as passive tracers. Redfield ratio is typically used to convert from one chemical compound to the others."

•    *In the main manuscript, a large space is given to the Arctic Ocean. Please introduce the Arctic Ocean accordingly and explain in the Introduction already why it might matter if you want to keep it as one of the regional seas that you want to discuss (see Terhaar et al., 2019 & 2021 and citations within).*

Thank you for the suggestion and references. We will add the paragraph below in the introduction after the first paragraph.

The Arctic Ocean, which accounts for only 4% of the global ocean area (Jakobsson, 2002), takes 11% of the global river discharge (McClelland et al., 2012), and it is estimated that about one third of its net primary production is sustained by nutrients originated from rivers and coastal erosion (Terhaar et al., 2021). Therefore, there is no surprise that Arctic primary productivity will be affected by altered riverine transport of nutrients and carbon under future climate changes. Previous studies have shown that enhanced riverine nutrient input increases primary production in the Arctic Ocean (Letscher et al., 2013; Le Fouest et al., 2013, 2015, 2018; Terhaar et al., 2019), while large riverine DOC delivery reduces $CO_2$ uptake in Siberian shelf seas (Anderson et al., 2009; Manizza et al., 2011).

•    *The model is not evaluated. Only a reference to previous publications is mentioned (lines 111-113). However, to understand and discuss the changes in the future and the sensitivity to riverine fluxes a much more detailed model evaluation is needed. For example, NPP is far too low in the Arctic Ocean: for the 2nd half of the 20th century the simulated NPP is around 100 TgC/yr. However, the observation-based NPP in the last years of the 20th century is slightly above 450 Tg C/yr (Arrigo and van Dijken, 2015). If the model is not capable of simulating NPP in a part of the*

*ocean, why should we trust any of the projections done by that model? Having demonstrated that this is the case in the Arctic Ocean, I cannot trust the other numbers. Especially, given that the model-obs differences (Fig 3b) are so much larger than the differences between rivers and no rivers (Fig. 3c). Please make a thorough comparison and evaluate your results on the background of the model performance and tell the reader about the models' strong points and weaknesses when it comes to ocean biogeochemistry.*

Thank you for the comment. The NorESM model has been evaluated in different publications. For the Arctic domain, its skill in simulating the observed primary production was done in Lee et al. (2016) and the reviewer is correct that the NorESM is biassed low against observations. However, it is on par with other global ocean models. For instance, in their paper, Lee et al. (2016) assessed the relative skills of 21 regional and global biogeochemical models in reproducing the observed contemporary Arctic primary production. The NorESM has a negative bias of -0.49, and is well within the multi-model mean bias of -31+/-0.39. In another study, the NorESM model is compared with a regional model that comprises part of the Arctic region, and it shows that the NorESM simulates too late and too short bloom period than the regional model (Skogen et al., 2018), hence the annual integrated primary production is too low. Global high resolution models also show considerably lower NPP in the Arctic (e.g., 165Tg C/yr; Terhaar et al., 2019). Such common shortcomings in global scale marine ecosystem models can partly be attributed by the simplified parameterization, which can be improved through data assimilation (Tjiputra et al., 2007; Gharamti et al., 2017).

We want to show the direction of improvement toward observation by showing the differences in PP between with rivers and no rivers (Fig. 3c). Despite the model-obs differences in the Arctic Ocean PP, we would like to see whether there is a difference in the sensitivity of PP and carbon uptake to the riverine input at global scale and in the Arctic (Figs. 5 and 7).

We will add discussion on model confidence in the Arctic Ocean in the 'Limitations and Uncertainties' section.

•    *Four different scenarios for the future riverine fluxes are introduced in line 125. However, it is impossible to know which scenario is which. Please introduce the scenarios carefully so that the reader knows these scenarios are.*

We will add a table similar to Figure 2 in the paper by Seitzinger et al. (2010) to introduce these four scenarios.

•    *Riverine data in general: How good is the data that you use? I would like to see a comparison to observations of larger important rivers such as the Amazon or the rivers in the Arctic that are observed by ArcticGRO (https://arcticgreatrivers.org/). Nutrient fluxes from Global News 2 in the Arctic can be off by 300% (Kaiser et al., 2017; Thibodeau et al., 2017; Terhaar et al., 2019). Without knowing the quality of the present-day riverine fluxes, it is not possible to evaluate the results.*

We didn't find any evaluation of the Global NEW 2 data in the publications that the reviewer mentioned here (Kaiser et al., 2017; Thibodeau et al., 2017; Terhaar et al., 2019). However, the Global NEWS 2 riverine dataset has been calibrated and

assessed against measured yields (Mayorga et al., 2010) and has been widely used and evaluated for different river estuaries (A number of publications can be found in a special section of Global Biogeochemical Cycles: https://agupubs.onlinelibrary.wiley.com/doi/toc/10.1002/(ISSN)1525-2027.NUTRIENT1). For example, van der Struijk et al. (2010) compared the Global NEWS nutrient yields to observed values for South American rivers. They stated that "For some rivers (such as the Amazon), the model performs better than for others. In general, the model seems to do better for DIN, DON and DOC than for DIP, although for the Amazon also modeled DIP yields also compare well to measured values. The variations in yields among rivers are described well by the NEWS models… Nevertheless, we may argue that the NEWS models in general perform reasonably well for South American rivers."

Although the evaluation of the riverine data is out of the focus of our paper, we will mention the evaluation of the dataset in section 2.2 and refer the readers to relevant publications.

- *Some of the organic nutrients are remineralized directly due to the fixed stoichiometric ratio in the marine organic matter (line 140). Please tell the reader how much is remineralized directly and discuss later if that influences the results. The lability of terrestrial organic matter is an important factor for the impact of riverine nutrient fluxes on NPP and carbon fluxes on air-sea CO2 fluxes (Terhaar et al., 2021).*

We have calculated the proportion of directly remineralized matter, including dissolved organic matter (DOM) and particulate (inorganic and organic) matter (PM), i.e., $X/(DOM_{riv}+PM_{riv})$ (X is the directly remineralized dissolved organic and particulate matter). The directly remineralized part on average accounts for 64.8%, 27.8% and 62.8% of the total riverine organic and particulate matter of phosphorus, nitrogen and carbon, respectively. This approach may lead to bias in the enhanced primary production. We further calculated the contribution of the directly remineralized part on the enhanced primary production, by comparing X with the corresponding total riverine-induced dissolved nutrient additions [$X/(X+DIXriv)*100\%$], which accounts to 80.5%, 33.3%, and 41.1% for phosphate, nitrate, and carbon, respectively. Assuming that all coastal regions are nutrient limited, this direct remineralization could be responsible for 33.3%-80.5% of the enhanced primary production, depending on which nutrient species is limiting the primary production.

- *Is there a particular reason why you use fluxes from 1970 for the FIX run? Later you compare the NPP results to observation-based NPP from after 2000. Wouldn't it be better to use the 2000 fluxes for the FIX run?*

In Section 3.3, we assess the effect of future changes in riverine inputs comparing the future period 2050–2099 to the recent past reference period 1950–1999. This assessment was our main objective when we designed the experiments. We chose the 1970 fluxes because they are more representative for the 1950–1999 period than the 2000 fluxes (Beusen et al., 2016).

We agree with the reviewer that the use of 2000 fluxes for the FIX run would have been preferred when comparing to observation-based NPP from after 2000. In hindsight, choosing 1950–1999 as a present day reference was not an optimal

choice considering better availability of observations estimates during the early 21st century. We will mention this in the discussion of caveats in the revised manuscript version.

- *As mentioned above, nutrient fluxes often do not scale at all with runoff as concentrations can decrease strongly when discharge increases. Furthermore, apart from DOM and DIP the global news scenarios all give very different future scenarios compared to RUN. I think the simulation RUN hence really makes very little sense and I do not know what its value is. I would certainly not make such strong statements in the Discussion (line 321). I am happy to be convinced otherwise.*

We agree with the reviewer that the RUN experiment has its uncertainties. It is an idealistic scenario, which assumes that future changes in riverine carbon and nutrient transports are directly linked to changes in riverine freshwater transports, ignoring other anthropogenic effects related to land surface processes. The comparison of RUN to the GNS experiments demonstrates that the anthropogenic and natural changes in nutrients and carbon on land are equally or more important than the direct effect of the changes in the hydrological cycle. Including RUN along with the more realistic GNS scenarios thus still provide valuable information, which e.g., may caution other modelling groups against adopting an over-simplified coupling of riverine nutrient and carbon transports to the hydrologic cycle. Another motivation for RUN was to introduce seasonal and interannual variability in nutrient and carbon transports that is linked to variability in riverine freshwater transport. Future work should explore if GNS and RUN can be integrated to produce more realistic long-term trends in riverine nutrient and carbon transports as well as short-term variability.

We will clarify that to explore the best practical way of implementing the riverine inputs for modelling groups is one of the aims of this work. We will delete the sentence in line 321.

- *It seems that large changes are always simulated in the Black Sea. However, results from ESMs in enclosed or semi-enclosed seas usually make no sense. Did you mask these seas, including the Mediterranean Sea? If not, how does masking these seas change your results?*

The reviewer is right that in global models the results in those enclosed or semi-enclosed seas are often largely biassed. We did not mask those seas during simulation, but we masked them for plotting.

- *I am really struggling with the significance of the results. For example, in line 192, I would like to see the inter-annual variability as a measure of the standard deviation to see how much they are really different. Similar, is there an uncertainty estimate for the observation-based estimates in line 197? Overall, the differences in annual NPP and RMSE seem to be so small that I am not sure if it makes sense to use terms such as 'better' or 'improve'. Can you find any statistical way to evaluate if the changes are significant? This comment should be addressed to other numbers throughout the manuscript. Please be also careful with the word 'significant' as used in line 238 if it has no statistical meaning.*

We thank the review for the comment. We will follow the suggestion and add more information on the robustness of changes in the revised manuscript. We will test the statistical significance with respect to sampling error resulting from interannual variability and add this information in the text and on the figures. In addition to formal significance, we will introduce a signal-to-noise measure that provides further information on the importance of the signal in a real-world context (more on that below). We will detail the significance assessment in a new section "2.4 Statistical robustness of riverine impacts" and move the last paragraph of section 2.3 to the new section.

As stated in section 2.3, we expect even small values to be statistically significant because the interannual climate variability is the same in all simulations and thus most of the interannual signal is removed in the computation of the experiment differences. We will illustrate this behaviour with a Supplementary Figure showing two time series from two different experiments in one panel and the time-evolving difference of these series in another panel, together with confidence intervals of the time-means. Differences between two temporal periods (i.e., future minus past) are still affected by interannual variability, but the sampling effect will be similar for all experiments and therefore should not impact much the comparison of the temporal differences between the experiments.

In the revised manuscript, we will formally test the statistical significance of changes by doing the following: 1) construct time series of differences (either from two periods or two experiments), 2) compute the standard-deviation of the mean from the time-evolving differences, 3) compute the p-value from the time-mean difference and the standard-deviation of the mean, 4) reject null-hypothesis if p<alpha. This will test only for local significance and the probability of falsely rejecting the null-hypothesis somewhere on the map is generally higher than alpha. While performing more rigorous multi-hypothesis and field significance testing would be desired, it is beyond the scope of this study and we have not observed it in similar studies. Despite limitations, the local significance test provides basic information on statistical robustness.

Small but statistically significant differences between the experiments (where the effect of interannual variability has been removed) are not necessarily large enough to have real-world implications and be detectable in observations. Therefore, we will introduce a second measure. We will define a signal-to-noise ratio S2N as the time-mean difference over the standard-deviation of the mean of the original field (and not the difference field with the interannual variability removed). On our difference maps, we will mark areas with S2N>1 as regions where the signal emerges from interannual noise, indicating that the signal is large enough to have real-world implications.

In addition to including robustness information in the difference maps, we will add a table that summarises all globally integrated changes mentioned in text and figures. We will add statistical significance (expressed as p-values) and S2N values to this table.

- *Please refrain from making strong claims about the Arctic. Indicating a ~76% increase in NPP is misleading giving how bad the model simulates the present-day NPP. Based on an observation-based NPP of 450 TgC/yr, a change of 70-80 TgC/yr is only an increase in 17%. Moreover, the very low present-day NPP suggests either*

*strong light or strong nutrient limitation. If it is strong nutrient limitation, riverine fluxes would have an overly strong effect because all nutrients would be used immediately. So maybe even the 17% are still too high. This goes back to the point that the reader must know how good the model performs locally.*

We agree that stating a relative change (in percentage) could be misleading given the biassed low primary productivity simulated in our model. As with other Earth system models, the increased Arctic primary production in NorESM, which is not strongly nutrient limited, is associated with sea-ice loss. Our projected PP increase in the reference run is roughly 70 Tg C yr-1 by the end of the 21st century. This is in the same order of magnitude estimated from 11 ESMs with mean change of 59Tg C yr-1 (individual ESM ranges from -110 to +253 Tg C yr-1; Vancopenolle et al., 2013). We will include the following statements in the revised version.

We note that the relative change in PP in the Arctic is likely to be overestimated since the NorESM simulates a biassed low PP under the contemporary climate. Nevertheless, the projected absolute change of 70Tg C yr-1 is well within the range estimated from other Earth system models (Vancopenolle et al., 2013).

*General comments*

- *Often 'biogeochemistry is used as a synonym for PP and air-sea CO2 fluxes. But the word biogeochemistry also includes acidification, carbon and nutrient cycles, and other things. Please just say PP and air-sea CO2 fluxes. ( for example line 250).*

We will make respective specifications in order to be clear in these occasions.

- *Please adhere to the best practice guide (https://www.ncei.noaa.gov/access/ocean-carbon-data-system/oceans/Handbook_2007/Guide_all_in_one.pdf) and use CT and AT instead of DIC and ALK.*

The best practices guide by Dickson et al. (2007) refers to best practices for measurements. The symbols AT and CT for alkalinity and total dissolved inorganic carbon are not sacrosanct. However, we will follow this recommendation and replace DIC by CT and ALK by AT.

- *I find the name 'reference run' misleading. It is rather a control run. Reference should be the best case or something.*

The term "reference experiment" is widely used in the Earth system modelling community in the way used in our study. It is standard for ESM modelling studies to have a "reference" experiment and one or several "sensitivity" experiments. The response is typically evaluated as the difference sensitivity minus reference. The term "control experiment" is typically reserved for a simulation where external forcings are fixed at pre-industrial (or some other) levels. However, the external forcings used in this study are transient ones, therefore using "control" may confuse readers.

Furthermore, the original NorESM1-ME (Tjiputra et el., 2012) and many other ESMs do not include riverine nutrient and carbon transports to the oceans. Therefore, we

find the use of "reference" for the experiment without transports appropriate in our study and prefer to keep it in the revised manuscript version.

- *Significant digits should always be the same. For example, in lines 204 and 205 you cite air-sea CO2 fluxes and use different number of digits.*

We will check and change all numbers accordingly.

*Minor comments*

- *Lines 19/20: Can one speak of improve based on such small changes in the RMSE? Is it significant?*

Referring to the response to major comment on the significance, we will do more analysis and edit it accordingly.

- *Line 13: Suggest changing "not only regionally but also globally" to "regionally and globally"*

We will change it accordingly.

- *Line 18: Suggest changing "modelled" to "simulated"*

We will change it accordingly.

- *Line 22: Unclear what you mean by depending on the riverine configuration. There is no range in the numbers given in this sentence.*

We meant that by adding the riverine nutrients input, the projected future decline in PP can be alleviated maximum by 0.6 PgC yr-1 (27.3%) globally in our experiments. The range is given in Conclusion in line 367-369 (Riverine nutrient inputs into surface coastal waters alleviate the nutrient limitation and considerably lessen the projected future decline in PP from -5.4% without riverine inputs to -4.4%, -4.1% and -3.6% in FIX, RUN and GNS (averaged over four scenarios), respectively. )

- *Lines 23/24: the last part of the sentence should be rewritten*

We will change it to "The riverine impact on projected C uptake depends on the net effect of riverine nutrient induced PP increase and riverine C input induced outgassing."

- *Lines 22-25: A lot of words that do not tell much. Nutrients increase CO2 uptake, CT fluxes decreases it. But where does it increase it and where does it decrease it? Maybe shorten this or explain.*

Please see the response to the last comment.

- *Line 26: Can you be more quantitative?*

We will change this sentence to "Simulations with high-resolution global or regional models with an adequate representation of shelf processes are required to accurately assess the impact of future riverine scenarios."

- *Line 31: Not sure if you can count runoff.*

We will change it to "river runoff plays an essential role in …".

- *Lines 31/32: transporting nutrients where? Suggest adding "into the ocean" after "transporting nutrients".*

We will add it accordingly.

- *Line 34: What do you mean by "absolutely dominant" source? More than 50%? Please be clear.*

We will add the information as follows. "For some substances such as total phosphorus (~90.0%) and total silicon (>70.0%), riverine input even acts as the absolutely dominant source."

- *Line 34: Suggest adding "into the ocean" after "transport of carbon".*

We will add it accordingly.

- *Line 34: Suggest writing air-sea CO2 exchange instead of air-sea C exchange. It CO2 and not C that is exchanged across the air-sea interface.*

We will change it accordingly.

- *Line 36: What is "it"?*

We will change this sentence to "Despite our limited understanding on the riverine carbon fluxes, they could play an important role in closing the global carbon budget (Friedlingstein et al., 2021) and could be very sensitive to regional and global changes such as weathering, land cover and climate (Meybeck and Vörösmarty, 1999)." They refer to riverine carbon fluxes.

- *Line 36: Do you mean global ocean carbon cycle or really global carbon cycle?*

We have removed this term. Please see above.

- *Line 36: Global and regional changes of what?*

Please see the response above.

- *Line 37: Suggest starting a new paragraph here*

We will follow the suggestion in the revised version.

- *Line 55: In the Arctic, Terhaar et al. (2019) started to assess future changes.*

Thank you for the reference. We will include it in the introduction.

- *Line 56: Please say which datasets you are referring to.*

We will add "Global NEWS2 dataset" in this sentence.

- *Line 58: Why does more data make the impact study more 'desirable'?*

"Taking the advantage of the latest improvement of global river nutrient/carbon export datasets and responding to the demand of development of ESMs with increasing model resolution, the assessment of the impact of riverine nutrients and carbon on future projections of marine biogeochemistry becomes feasible and desired, especially for impact studies along continental margins."

Here more data makes the assessment feasible, and increasing model resolution makes the assessment desirable.

- *Line 58: Please say why it is now feasible? One could argue that the CMIP6 horizontal resolution is still not good enough to resolve the global ocean.*

Please see the response above.

- *Line 68: Please already say here why you use RCP4.5.*

We will change it to "the RCP4.5 (middle-of-the-road) scenario" here. We think section 2.3 Experimental design is a suitable place to state the detailed reason for the choice of future scenario. We will also add references to support our choice. Please see the response to comment on Line 153 below.

- *Line 84: Configured sounds strange here.*

We will change "configured" to "implemented".

- *Line 88: 'd' is missing in 'based'.*

We will add it.

- *Lines 88-113: Does the ocean biogeochemical component also have a name? It is very confusing that you say it is based on HAMOCC and then you only describe HAMOCC. That makes it impossible to understand what has changed.*

In the NorESM1-ME, the biogeochemical model is still called HAMOCC5, despite some development from the original HAMOCC5 version, which was developed in Hamburg. However, in the newer version of NorESM2, the biogeochemical model is renamed as iHAMOCC.

- *Line 129: What is the motivation to use CT and AT data from Hartmann (2009)? What kind of data is that? Modeled, observed, extrapolated? What is the underlying data? Please explain what you use.*

The reason that we used CT and AT data from Hartmann (2009) is that the Global NEWS2 dataset does not include CT and AT data. The CT and AT data from Hartmann (2009) are produced from a high-resolution model for global $CO_2$-consumption by chemical weathering. The dataset contains different forms of riverine carbon (dissolved and particulate, inorganic and organic), implemented to the Global NEWS2 river basin map.

- *Lines 129-133: Why do you use iron riverine fluxes from 1990? Is there nothing newer including observations since 1990? Does it make sense to weight iron river fluxes by runoff? Often nutrients do not scale at all with runoff (Holmes et al., 2012), so I would like to see some support for this assumption.*

To the best of our knowledge, the available global riverine iron dataset is rare. Previous studies have used various approximation approaches, e.g., constant Fe to dissolved inorganic carbon (DIC) ratio (Aumont et al., 2015), Fe to phosphorus ratio (Lacroix et al., 2020). In the study by Aumont et al. (2015), the Fe: DIC ratio is determined so that the total Fe supply equals 1.45 Tg Fe yr−1 as estimated by Chester (1990). We are aware that our approximation likely has bias in regional scales, and we will discuss this in the Limitation section.

- *Line 135: Is there a reason why you use 1000 km and 300 km here?*

The short answer is that NorESM1 is based on NCAR's Community Earth System Model (CESM; Hurrell et al., 2013). The configuration for distributing riverine runoff into ocean grid cells along with the 1-degree resolution ocean grid have been both inherited from CESM without modification.

The exact reasoning behind the NCAR's choice of using a 1000 km e-folding length scale and 300 km cutoff value is not known to us. The effect of using an e-folding length scale that is considerably larger than the cutoff value is that approximately equal weights are used when distributing the runoff to ocean grid cells that lie within the cutoff range of the river mouth. The cutoff value must be large enough that at least one ocean grid cell midpoint lies within the range. For a 1-degree resolution ocean grid, a value of 100 km should be sufficient to satisfy this. Possibly, the large value of 300 km was chosen to also satisfy coarser grids, such as the 3-degree resolution grid used in NCAR's computationally efficient model configuration, and to avoid numerical instabilities by more smoothly distributing the runoff.

In this study, we ensured that riverine nutrients and carbon are distributed in the same way as riverine freshwater. We considered this important especially for the RUN configuration that couples the variability of the riverine nutrient and carbon transports to the variability of the riverine freshwater transport. How sensitive the ocean biogeochemistry impacts are to the details of how riverine runoff is distributed into coastal grid cells and generally how well ocean shelf processes are represented warrants further investigation.

We will add some text summarising the above to the discussion section or the Supplementary Information of the revised manuscript.

- *Line 145: Any reason why not GLODAPv2 is used?*

Our NorESM1-ME model configuration was finalised in the early 2010s, and GLODAPv2 was not available at that time. While there are improvements in the GLODAPv2 due to higher volume of data and improved interpolation scheme (Lauvset et al., 2016), since the model was then spun up for nearly 1000 years, we expect that these differences in initialization will not be significant for our purpose.

- *Line 147: Is the additional spin-up of 200 years is sufficient to get into a new equilibrium.*

The nutrient drift after 200 years spin-up is small (in the order of 1%/100 years).

- *Line 153: In what sense is RCP4.5 the most representative scenario? Most likely? Based on what?*

We will add the following sentence here. "Here, we consider RCP4.5 as the representative future scenario following the $CO_2$ emission rate based on the submitted Intended Nationally Determined Contributions (INDC), which projects a median warming of 2.6-3.1 degC by 2100 (Rogelj et al., 2016)."

- *Line 160: What means not considered? Deactivated?*

We will change "not considered" to "deactivated".

- *Lines 168-170: It is not entirely clear that you make 4 simulations. Can you be a little bit more explicit?*

We will change Line 166 to "GNS: Four different transient inputs following future projections of NEWS 2."

- *Line 194: In the figure it does not look as if only 15% of the increase is in the coastal shelf seas. What do you mean by predominantly, can you be quantitative here?*

The number 15% is calculated from continental shelf, where seafloor is shallower than 300m. The increased primary production occurs on the continental shelves in the North Atlantic will be quantified in the revised version.

• *Line 214: Is this a result or a speculation. If you cannot prove it, I suggest to either add it to the Discussion and add literature that supports this point or delete it.*

This is not a direct result. We will remove it.

• *Line 244-245: Does it really make sense to say slightly higher if the difference is that small?*

We will add more information on the robustness of changes in the revised manuscript. Please also see the response to the comment on the significance of the results.

• *Line 272: Please be quantitative*

We will change it to "The projection of global total PP shows up to 27.3% less decrease, if riverine inputs are present in the model."

• *Line 279: Is not nitrogen the limiting nutrient in the Arctic Ocean (Tremblay et al., 2015)?*

Our model shows that a large area of the Arctic Ocean is nitrogen limiting, while some part is iron limiting.

• *Line 286: Why do you not add CMIP6 data?*

We will add comparison with CMIP6 results (Kwiatkowski et al., 2020; Tagliabue et al., 2021) in the revised version.

• *Why do you speculate in the Discussion in lines 290 to 293: You can show that with your model results.*

Thank you for the comment. The statement was inferred from model results (i.e., FIX simulation produces higher NPP than REF simulation, which suggests that riverine-induced nutrient addition alleviate the stronger nutrient limitation in the future. We have rephrased the sentence to clarify this.

• *Figure 4c is almost impossible to read.*

We will improve this figure by smoothing the contour lines and using different colours.

• *Figure 5a: Could you highlight the +38 more? I was confused first.*

We will highlight it in the revised version.

**Reference**

Anderson, L. G., Jutterström, S., Hjalmarsson, S., Wåhlström, I., & Semiletov, I. P. (2009). Outgassing of CO2 from Siberian Shelf seas by terrestrial organic matter decomposition. *Geophysical Research Letters*, 36, L20601. https://doi.org/10.1029/2009GL04004

Aumont, O., Ethé, C., Tagliabue, A., Bopp, L., and Gehlen, M.: PISCES-v2: an ocean biogeochemical model for carbon and ecosystem studies, Geosci. Model Dev., 8, 2465-2513, 10.5194/gmd-8-2465-2015, 2015.

Chester, R.: The transport of material to the oceans: the river pathway, in: Marine Geochemistry, Springer, Dordrecht, https://doi.org/10.1007/978-94-010-9488-7_3, 1990.

Chester, R.: The Input of Material to the Ocean Reservoir, in: Marine Geochemistry, Wiley Online Books, 7-10, https://doi.org/10.1002/9781118349083.ch2 https://doi.org/10.1002/9781118349083.ch3, 2012.

Danabasoglu, G., Lamarque, J. F., Bacmeister, J., Bailey, D. A., DuVivier, A. K., Edwards, J., Emmons, L. K., Fasullo, J., Garcia, R., Gettelman, A., Hannay, C., Holland, M. M., Large, W. G., Lauritzen, P. H., Lawrence, D. M., Lenaerts, J. T. M., Lindsay, K., Lipscomb, W. H., Mills, M. J., Neale, R., Oleson, K. W., Otto-Bliesner, B., Phillips, A. S., Sacks, W., Tilmes, S., van Kampenhout, L., Vertenstein, M., Bertini, A., Dennis, J., Deser, C., Fischer, C., Fox-Kemper, B., Kay, J. E., Kinnison, D., Kushner, P. J., Larson, V. E., Long, M. C., Mickelson, S., Moore, J. K., Nienhouse, E., Polvani, L., Rasch, P. J., and Strand, W. G.: The Community Earth System Model Version 2 (CESM2), Journal of Advances in Modeling Earth Systems, 12, e2019MS001916, https://doi.org/10.1029/2019MS001916, 2020.

Dickson, A.G.; Sabine, C.L. and Christian, J.R. (eds) (2007) Guide to best practices for ocean CO2 measurement. Sidney, British Columbia, North Pacific Marine Science Organization, 191pp. (PICES Special Publication 3; IOCCP Report 8). DOI: https://doi.org/10.25607/OBP-1342

Friedlingstein, P., Jones, M. W., O'Sullivan, M., Andrew, R. M., Bakker, D. C. E., Hauck, J., Le Quéré, C., Peters, G. P., Peters, W., Pongratz, J., Sitch, S., Canadell, J. G., Ciais, P., Jackson, R. B., Alin, S. R., Anthoni, P., Bates, N. R., Becker, M., Bellouin, N., Bopp, L., Chau, T. T. T., Chevallier, F., Chini, L. P., Cronin, M., Currie, K. I., Decharme, B., Djeutchouang, L., Dou, X., Evans, W., Feely, R. A., Feng, L., Gasser, T., Gilfillan, D., Gkritzalis, T., Grassi, G., Gregor, L., Gruber, N., Gürses, Ö., Harris, I., Houghton, R. A., Hurtt, G. C., Iida, Y., Ilyina, T., Luijkx, I. T., Jain, A. K., Jones, S. D., Kato, E., Kennedy, D., Klein Goldewijk, K., Knauer, J., Korsbakken, J. I., Körtzinger, A., Landschützer, P., Lauvset, S. K., Lefèvre, N., Lienert, S., Liu, J., Marland, G., McGuire, P. C., Melton, J. R., Munro, D. R., Nabel, J. E. M. S., Nakaoka, S. I., Niwa, Y., Ono, T., Pierrot, D., Poulter, B., Rehder, G., Resplandy, L., Robertson, E., Rödenbeck, C., Rosan, T. M., Schwinger, J., Schwingshackl, C., Séférian, R., Sutton, A. J., Sweeney, C., Tanhua, T., Tans, P. P., Tian, H., Tilbrook,

B., Tubiello, F., van der Werf, G., Vuichard, N., Wada, C., Wanninkhof, R., Watson, A., Willis, D., Wiltshire, A. J., Yuan, W., Yue, C., Yue, X., Zaehle, S., and Zeng, J.: Global Carbon Budget 2021, Earth Syst. Sci. Data Discuss., 2021, 1-191, 10.5194/essd-2021-386, 2021.

Hajima, T., Watanabe, M., Yamamoto, A., Tatebe, H., Noguchi, M. A., Abe, M., Ohgaito, R., Ito, A., Yamazaki, D., Okajima, H., Ito, A., Takata, K., Ogochi, K., Watanabe, S., and Kawamiya, M.: Development of the MIROC-ES2L Earth system model and the evaluation of biogeochemical processes and feedbacks, Geosci. Model Dev., 13, 2197-2244, 10.5194/gmd-13-2197-2020, 2020.

Hartmann, J.: Bicarbonate-fluxes and CO2-consumption by chemical weathering on the Japanese Archipelago — Application of a multi-lithological model framework, Chemical Geology, 265, 237-271, https://doi.org/10.1016/j.chemgeo.2009.03.024, 2009.

Hurrell et al. (2013). The Community Earth System Model: A Framework for Collaborative Research, Bulletin of the American Meteorological Society, 94(9), 1339-1360, https://doi.org/10.1175/BAMS-D-12-00121.1.

Kwiatkowski, L., Torres, O., Bopp, L., Aumont, O., Chamberlain, M., Christian, J. R., Dunne, J. P., Gehlen, M., Ilyina, T., John, J. G., Lenton, A., Li, H., Lovenduski, N. S., Orr, J. C., Palmieri, J., Santana-Falcón, Y., Schwinger, J., Séférian, R., Stock, C. A., Tagliabue, A., Takano, Y., Tjiputra, J., Toyama, K., Tsujino, H., Watanabe, M., Yamamoto, A., Yool, A., and Ziehn, T.: Twenty-first century ocean warming, acidification, deoxygenation, and upper-ocean nutrient and primary production decline from CMIP6 model projections, Biogeosciences, 17, 3439-3470, 10.5194/bg-17-3439-2020, 2020.

Lacroix, F., Ilyina, T., & Hartmann, J. (2020). Oceanic $CO_2$ outgassing and biological production hotspots induced by pre-industrial river loads of nutrients and carbon in a global modeling approach. *Biogeosciences*, **17**, 55– 88. https://doi.org/10.5194/bg-17-55-2020

Lauvset, S. K., Key, R. M., Olsen, A., van Heuven, S., Velo, A., Lin, X., Schirnick, C., Kozyr, A., Tanhua, T., Hoppema, M., Jutterström, S., Steinfeldt, R., Jeansson, E., Ishii, M., Perez, F. F., Suzuki, T., and Watelet, S.: A new global interior ocean mapped climatology: the 1° × 1° GLODAP version 2, Earth Syst. Sci. Data, 8, 325-340, 10.5194/essd-8-325-2016, 2016.

Le Fouest, V., Babin, M., & Tremblay, J.-E. (2013). The fate of riverine nutrients on Arctic shelves. *Biogeosciences*, 10(6), 3661–3677. https://doi.org/10.5194/bg-10-3661-2013

Le Fouest, V., Manizza, M., Tremblay, B., & Babin, M. (2015). Modelling the impact of riverine DON removal by marine bacterioplankton on primary production in the Arctic Ocean. *Biogeosciences*, 12(11), 3385–3402. https://doi.org/10.5194/bg-12-3385-2015

Le Fouest, V., Matsuoka, A., Manizza, M., Shernetsky, M., Tremblay, B., & Babin, M. (2018). Towards an assessment of riverine dissolved organic carbon in surface waters of the western Arctic Ocean based on remote sensing and biogeochemical modeling. *Biogeosciences*, 15(5), 1335–1346. https://doi.org/10.5194/bg-15-1335-2018

Lee, Y.J., P.A. Matral, A.M. Friedrichs, V.S. Saba, O. Aumont, M. Babin, E. Buitenhius, M. Chevallier, L. de Mora, M. Dessert, D. Feldman, R. Frouin, M. Gehlen, T. Gorgues, T. Ilyina, M. Jin, J. John, J. Lawrence, C. Perruche, V. Le Fouest, E. Popova, A. Romanou, A. Samuelsen, J. Schwinger, R. Séférian, C. Stock, J. Tjiputra, B. Tremblay, K. Ueyoshi, M. Vichi, A. Yool, and J. Zhang, 2016: Net primary productivity estimates and environmental variables in the Arctic Ocean: An assessment of coupled physical-biogeochemical models. *J. Geophys. Res. Oceans*, **121**, no. 12, 8635-8669, doi:10.1002/2016JC011993.

Letscher, R. T., Hansell, D. A., Kadko, D., & Bates, N. R. (2013). Dissolved organic nitrogen dynamics in the Arctic Ocean. *Marine Chemistry*, 148, 1–9. https://doi.org/10.1016/j.marchem.2

Manizza, M., Follows, M. J., Dutkiewicz, S., Menemenlis, D., McClelland, J. W., Hill, C. N., Peterson, B. J., & Key, R. M. (2011). A model of the Arctic Ocean carbon cycle. *Journal of Geophysical Research*, 116, C12020. https://doi.org/10.1029/2011JC006998

Meybeck, M. and Vörösmarty, C.: Global transfer of carbon by rivers, Global Change Newsletter, 37, 18-19, 1999.

McClelland, J. W., Holmes, R. M., Dunton, K. H., & Macdonald, R. W. (2012). The Arctic Ocean estuary. Estuaries Coasts, 35(2), 353–368. https://doi.org/10.1007/s12237-010-9357-3

Rogelj, J., den Elzen, M., Höhne, N., Fransen, T., Fekete, H., Winkler, H., Schaeffer, R., Sha, F., Riahi, K., and Meinshausen, M.: Paris Agreement climate proposals need a boost to keep warming well below 2 °C, Nature, 534, 631-639, 10.1038/nature18307, 2016.

Seitzinger, S. P., Mayorga, E., Bouwman, A. F., Kroeze, C., Beusen, A. H. W., Billen, G., Van Drecht, G., Dumont, E., Fekete, B. M., Garnier, J., and Harrison, J. A.: Global river nutrient export: A scenario analysis of past and future trends, Global Biogeochemical Cycles, 24, https://doi.org/10.1029/2009GB003587, 2010. Jakobsson, M. (2002). Hypsometry and volume of the Arctic Ocean and its constituent seas. Geochemistry, Geophysics, Geosystems, 3(5), 1028. https://doi.org/10.1029/2001GC000302

Séférian, R., Nabat, P., Michou, M., Saint-Martin, D., Voldoire, A., Colin, J., Decharme, B., Delire, C., Berthet, S., Chevallier, M., Sénési, S., Franchisteguy, L., Vial, J., Mallet, M., Joetzjer, E., Geoffroy, O., Guérémy, J.-F., Moine, M.-P., Msadek, R., Ribes, A., Rocher, M., Roehrig, R., Salas-y-Mélia, D., Sanchez, E., Terray, L.,

Valcke, S., Waldman, R., Aumont, O., Bopp, L., Deshayes, J., Éthé, C., and Madec, G.: Evaluation of CNRM Earth System Model, CNRM-ESM2-1: Role of Earth System Processes in Present-Day and Future Climate, Journal of Advances in Modeling Earth Systems, 11, 4182-4227, https://doi.org/10.1029/2019MS001791, 2019.

Séférian, R., Berthet, S., Yool, A. et al. Tracking Improvement in Simulated Marine Biogeochemistry Between CMIP5 and CMIP6. Curr Clim Change Rep 6, 95–119 (2020). https://doi.org/10.1007/s40641-020-00160-0

Skogen, M. D., Hjøllo, S. S., Sandø, A. B., and Tjiputra, J.: Future ecosystem changes in the Northeast Atlantic: a comparison between a global and a regional model system, ICES Journal of Marine Science, 75, 2355-2369, 10.1093/icesjms/fsy088, 2018.

Tagliabue, A., Kwiatkowski, L., Bopp, L., Butenschön, M., Cheung, W., Lengaigne, M., and Vialard, J.: Persistent Uncertainties in Ocean Net Primary Production Climate Change Projections at Regional Scales Raise Challenges for Assessing Impacts on Ecosystem Services, Frontiers in Climate, 3, 2021.

Terhaar, J., Orr, J. C., Ethé, C., Regnier, P., & Bopp, L. (2019). Simulated Arctic Ocean response to doubling of riverine carbon and nutrient delivery. Global Biogeochemical Cycles, 33, 1048– 1070. https://doi.org/10.1029/2019GB006200

Terhaar, J., Lauerwald, R., Regnier, P. et al. Around one third of current Arctic Ocean primary production sustained by rivers and coastal erosion. Nat Commun 12, 169 (2021). https://doi.org/10.1038/s41467-020-20470-z

Tjiputra, J. F., Polzin, D., and Winguth, A. M. E. (2007), Assimilation of seasonal chlorophyll and nutrient data into an adjoint three-dimensional ocean carbon cycle model: Sensitivity analysis and ecosystem parameter optimization, *Global Biogeochem. Cycles*, 21, GB1001, doi:10.1029/2006GB002745.

Tjiputra, J. F., Roelandt, C., Bentsen, M., Lawrence, D. M., Lorentzen, T., Schwinger, J., Seland, Ø., and Heinze, C.: Evaluation of the carbon cycle components in the Norwegian Earth System Model (NorESM), Geosci. Model Dev., 6, 301-325, 10.5194/gmd-6-301-2013, 2013.

van der Struijk, L. F. and Kroeze, C.: Future trends in nutrient export to the coastal waters of South America: Implications for occurrence of eutrophication, Global Biogeochemical Cycles, 24, https://doi.org/10.1029/2009GB003572, 2010.

Vancoppenolle, M., Bopp, L., Madec, G., Dunne, J., Ilyina, T., Halloran, P. R., and Steiner, N. (2013), Future Arctic Ocean primary productivity from CMIP5 simulations: Uncertain outcome, but consistent mechanisms, *Global Biogeochem. Cycles*, 27, 605– 619, doi:10.1002/gbc.20055.

---

## Author Response (AR1)

**Reply to Anonymous Referee #1**

**We are grateful for the comments and edits of the anonymous reviewer who invested time for the review of our manuscript. The reviewer's text is reported in Italic and our responses in roman.**

*The research topic is very timely, the manuscript is well written, and the model simulates seem to be well done. Furthermore, the authors have really made a lot of simulations and obviously went to great lengths. However, many things need to be addressed before the paper can be accepted. The most important points are: (a) Discussing Lacroix et al. (2021), (b) adding many more references to support the text, (c) evaluating the model results with observations and show where the model is good or bad, (d) evaluating the global news present-day river fluxes to understand how realistic they are, (e) discussing the far too low NPP in the Arctic Ocean, and (f) testing if results are statistically significant. In this state, the main conclusion that rivers might be of importance in coastal region has no underlying prove and large parts of the manuscript are not possible to review. I would focus more on regional coastal areas and discuss and analyze these further, but this is up to you.*

We thank the reviewer for this input and we have made a substantial effort to address her/his concerns in the updated manuscript. We have added more references to support the text and included comparison with Lacroix et al. (2021). We have added more discussion on the sensitivity of the results to the model performance, globally and for the Arctic Ocean in particular. We added statistical significance testing and robustness information. We reworked the text and clarified that "rivers might be of importance in coastal region" is not considered as a conclusion of this study. In our response to the specific comments, we provide arguments for why including an independent evaluation of the global news data product (which has been documented in literature) as well as an extensive model-data evaluation is beyond the scope of this short and focused scientific paper. Instead, we suggest extending the discussion to include specific aspects on uncertainties related to our model and data products we used. We have implemented most suggestions from the specific comments of the reviewer, such as including more detailed information of the global news scenarios and the riverine implementations in current Earth system models, and stated reasons if we did not implement a suggestion. More details are provided in our replies to the reviewer's specific comments below.

*Major comments*

- *References are often missing. One of the most important references missing is probably Lacroix et al. (2021) who performed very similar simulations. Comparison to their results would be essential.*
We have added comparison to their results at several places. However, they have applied temporally varying (increasing) riverine nutrients (only N and P) over 1905–2010, in which period our riverine inputs are mostly kept constant (even in the transient experiments). Additionally, they didn't add riverine carbon input, which makes it difficult to compare with. Their study focuses on the impact of changing (increasing) terrestrial nutrients on the marine primary production and carbon cycle, while our study explores

the impacts in different riverine configurations. One important addition to their study is, e.g., while the riverine nutrients and C input is kept constant over the whole simulation time, its impact on projected PP and C uptake expresses differently in future period from the historical period.

- *Please revise the entire manuscript for the many missing references. Here some examples:*
  - *Lines 30/31: no references for this statement*

This sentence together with the sentence after is one complete statement, and the references are therefore given after the sentence in Line 32. We have changed to use ";" to connect both sentences.

  - *Lines 33/34: no references for this statement*

Reference (Chester, 2012) added.

  - *Lines 34/35: no references for this statement*

References (Meybeck and Vörösmarty, 1999; Liu et al., 2021) added.

  - *Lines 37-48: Almost no references here!*

Line 37-38: Reference (Seitzinger et al., 2010) added.

Line 39-41: References (Bouwman et al., 2009; Garnier et al., 2021; Van Drecht et al., 2009; Eiriksdottir et al., 2016; Zhang et al., 2022) added.

Line 41-44: Reference was given.

Line 44-46: References (Liu et al., 2020; Frigstad et al., 2020; Wild et al., 2019; Pokrovsky et al., 2020; Mann et al., 2022) added.

  - *Lines 49/50: No references*

We have added a new paragraph with more detailed information for individual models with corresponding citations (please see the answer to next comment).

  - *…*
- *Lines 49-54: A more detailed explanation is missing on how ESMs simulate rivers. Seferian et al. (2020) gives a good overview. There were already 5 ESMs in CMIP5 that simulated riverine C, N, and P (3 also simulated Fe) fluxes and there are now 8 in CMIP6. Some of them even simulate dynamically changing riverine fluxes.*

We have added more information on the riverine implementation in the introduction, especially with respect to new developments in some of the CMIP6 models (e.g. CESM2, CNRM-ESM2-1, MIROC-ES2L, IPSL-CM6A-LR):

The latest generation of Earth system models (ESMs) have implemented some forms of riverine inputs in their ocean biogeochemistry modules (Séférian et al., 2020). The models that include riverine inputs use different implementations, from constant contemporary fluxes (e.g., IPSL-SM6A-LR and CESM2; Aumont et al., 2015; Danabasoglu et al., 2020) to interactive with terrestrial nutrient leaching transported by

dynamical river routing (e.g., CNRM-ESM2-1 and MIROC-ES2L; Séférian et al., 2019; Hajima et al., 2020), and typically use the Redfield ratio to convert from one chemical compound to the others. For instance, in the latest version of IPSL model (IPSL-SM6A-LR; Aumont et al., 2015) riverine nutrients (DIN, DIP, Si), dissolved organic nitrogen (DON), dissolved organic phosphorus (DOP), dissolved inorganic carbon (DIC) and total alkalinity (TA) are implemented as constant contemporary fluxes based on data sets from Global NEWS 2 (NEWS 2; Mayorga et al., 2010) and the Global Erosion Model of Ludwig et al. (1996). Similarly, in the CESM2 (Danabasoglu et al., 2020) riverine nutrients (except DIN and DIP), DIC and TA are held constant using data from NEWS 2 (Mayorga et al., 2010), but DIN and DIP are taken from a model (Beusen et al., 2015, 2016) and vary from 1900 to 2005, which is more sophisticated than using constant fluxes. Some ESMs have implemented interactive riverine nutrients input from terrestrial processes, e.g., in the CNRM-ESM2-1 the riverine dissolved organic carbon (DOC) is calculated actively from litter and soil carbon leaching in the land model, and the supply of the other nutrients, DIC and TA have been parameterized using the global average ratios to DOC from Mayorga et al. (2010) and Ludwig et al. (1996). In MIROC-ES2L model, N cycle is coupled between the ocean and land ecosystems, therefore, the inorganic N leached from the soil is transported by rivers and subsequently as an input to the ocean ecosystem. The riverine P is calculated from N using the Redfield ratio, but riverine carbon input is not implemented. Existing models with interactive riverine inputs typically do not consider biogeochemical processes in the freshwater system such as outgassing or sedimentation.

- *In the main manuscript, a large space is given to the Arctic Ocean. Please introduce the Arctic Ocean accordingly and explain in the Introduction already why it might matter if you want to keep it as one of the regional seas that you want to discuss (see Terhaar et al., 2019 & 2021 and citations within).*

Thank you for the suggestion and references. We have added a paragraph as below in the introduction.

Some regions such as the Arctic Ocean may receive a higher impact from changes in riverine inputs than regions. The Arctic Ocean accounts for only 4% of the global ocean area (Jakobsson, 2002), but takes 11% of the global river discharge (McClelland et al., 2012), and it is estimated that about one third of its net PP is sustained by nutrients originated from rivers and coastal erosion (Terhaar et al., 2021). Therefore, one can expect that Arctic PP will be affected by altered riverine transport of nutrients and carbon under future climate changes. Previous studies have shown that enhanced riverine nutrient input increases PP in the Arctic Ocean (Letscher et al., 2013; Le Fouest et al., 2013, 2015, 2018; Terhaar et al., 2019), while large riverine dissolved organic carbon (DOC) delivery reduces $CO_2$ uptake in Siberian shelf seas (Anderson et al., 2009; Manizza et al., 2011).

- *The model is not evaluated. Only a reference to previous publications is mentioned (lines 111-113). However, to understand and discuss the changes in the future and the sensitivity to riverine fluxes a much more detailed model evaluation is needed. For example, NPP is far too low in the Arctic Ocean: for the 2nd half of the 20th century the simulated NPP is around 100 TgC/yr. However, the observation-based NPP in the last years of the 20th century is slightly above 450 Tg C/yr (Arrigo and van Dijken, 2015). If the model is not capable of simulating NPP in a part of the*

*ocean, why should we trust any of the projections done by that model? Having demonstrated that this is the case in the Arctic Ocean, I cannot trust the other numbers. Especially, given that the model-obs differences (Fig 3b) are so much larger than the differences between rivers and no rivers (Fig. 3c). Please make a thorough comparison and evaluate your results on the background of the model performance and tell the reader about the models' strong points and weaknesses when it comes to ocean biogeochemistry.*

Thank you for the comment. The NorESM model has been evaluated in different publications. For the Arctic domain, its skill in simulating the observed primary production was done in Lee et al. (2016) and the reviewer is correct that the NorESM is biassed low against observations. However, it is on par with other global ocean models. For instance, in their paper, Lee et al. (2016) assessed the relative skills of 21 regional and global biogeochemical models in reproducing the observed contemporary Arctic primary production. The NorESM has a negative bias of -0.49, and is well within the multi-model mean bias of -0.31± 0.39. In another study, the NorESM model is compared with a regional model that comprises part of the Arctic region, and it shows that the NorESM simulates too late and too short bloom period than the regional model (Skogen et al., 2018), hence the annual integrated primary production is too low. Many coarse/intermediate resolution global models also show considerably lower NPP in the Arctic (e.g., 165Tg C/yr; Terhaar et al., 2019). Such common shortcomings in global scale marine biogeochemical models can partly be attributed by the simplified, not regionally adapted parameterization, which can be improved through data assimilation (Tjiputra et al., 2007; Gharamti et al., 2017).

As the reviewer suggested, we have added a subsection (2.2 model evaluation) to assess the model performance of the relevant part, and especially in the Arctic Ocean. We have also assessed the robustness of our results (please see the response to the comment on "result significance" below).

• *Four different scenarios for the future riverine fluxes are introduced in line 125. However, it is impossible to know which scenario is which. Please introduce the scenarios carefully so that the reader knows these scenarios are.*

We added a table (Table 1) to introduce these four future scenarios in more detail.

• *Riverine data in general: How good is the data that you use? I would like to see a comparison to observations of larger important rivers such as the Amazon or the rivers in the Arctic that are observed by ArcticGRO (https://arcticgreatrivers.org/). Nutrient fluxes from Global News 2 in the Arctic can be off by 300% (Kaiser et al., 2017; Thibodeau et al., 2017; Terhaar et al., 2019). Without knowing the quality of the present-day riverine fluxes, it is not possible to evaluate the results.*

The Global NEWS 2 riverine dataset has been calibrated and assessed against measured yields (Mayorga et al., 2010) and has been widely used and evaluated for different river estuaries (A number of publications can be found in a special section of Global Biogeochemical Cycles: https://agupubs.onlinelibrary.wiley.com/doi/toc/10.1002/(ISSN)1525-2027.NUTRIENT1). For example, van der Struijk et al. (2010) compared the Global NEWS nutrient yields to observed values for South American rivers. They stated that "For some rivers (such as the Amazon), the model performs better than for others. In

general, the model seems to do better for DIN, DON and DOC than for DIP, although for the Amazon also modeled DIP yields also compare well to measured values. The variations in yields among rivers are described well by the NEWS models… Nevertheless, we may argue that the NEWS models in general perform reasonably well for South American rivers."

For the Arctic rivers, as the reviewer suggested, we have checked the ArcticGRO dataset, however, we had to use the early dataset from PARTNERS Project (Holmes et al., 2012) rather than the ArcticGRO data in order to compare with the NEWS 2 data for the year 2000. In the paper by Terhaar et al. (2019), they have compared riverine total dissolved nitrogen and phosphorus (TDN, TDP) between NEWS 2 and PARTNERS Project and stated 41-169% overestimate of TDN by NEWS 2. However, the NEWS 2 dataset does not provide TDN and TDP data directly, and we did not find the information on how the TDN and TDP data from NEWS 2 are derived in the work by Terhaar et al. (2019). Therefore, we did our own comparison between the two datasets (see Table C1 in Appendix C) of dissolved inorganic nitrogen (DIN) and dissolved organic nitrogen (DON), which are directly provided by both datasets (DIP and DOP are not provided by Holmes et al. (2012), therefore we did not compare them). Our comparison shows that the NEWS 2 dataset compares fairly well with the measured data, especially for the Eurasian Arctic rivers with 3.5-28.6% deviation in DIN and 7.3-34.8% in DON, while the discrepancy is larger in the Canadian-Alaska Arctic rivers (i.e., Yukon and Mackenzie rivers) with upto 80.8% and 100% deviation in DIN and DON, respectively.

As the evaluation of the riverine data is out of the focus of our paper, we summarized the text above and added a brief evaluation of the dataset in section 2.3 and referred the readers to relevant publications.

- *Some of the organic nutrients are remineralized directly due to the fixed stoichiometric ratio in the marine organic matter (line 140). Please tell the reader how much is remineralized directly and discuss later if that influences the results. The lability of terrestrial organic matter is an important factor for the impact of riverine nutrient fluxes on NPP and carbon fluxes on air-sea CO2 fluxes (Terhaar et al., 2021).*

We have calculated the proportion of directly remineralized matter, including dissolved organic matter (DOM) and particulate (inorganic and organic) matter (PM), i.e., $X/(DOM_{riv}+PM_{riv})$ (X is the directly remineralized dissolved organic and particulate matter). The directly remineralized part on average accounts for 64.8%, 27.8% and 62.8% of the total riverine organic and particulate matter of phosphorus, nitrogen and carbon, respectively. This approach may lead to bias in the enhanced primary production. We further calculated the contribution of the directly remineralized part on the enhanced primary production, by comparing X with the corresponding total riverine-induced dissolved nutrient additions $[X/(X+DIXriv)*100\%]$, which accounts for 80.5%, 33.3%, and 41.1% for phosphate, nitrate, and carbon, respectively. Assuming that all coastal regions are nutrient limited, this direct remineralization could be responsible for 33.3%-80.5% of the enhanced primary production, depending on which nutrient species is limiting the primary production. In our model, phosphate is rarely limiting (Figure A1), therefore, the impact of this direct remineralization on primary production is likely on the lower end of this range (33.3%-80.5%). Given that the proportion of the direct remineralized organic matters in our model is comparable to those reported by Lacroix et al. (2021), who quantified that around 50% of the riverine DOM and 75% of

the POM are mineralized in global shelf waters, the impact on enhanced PP should be less than 33.3%.

We have added this discussion in section 4.3.

- *Is there a particular reason why you use fluxes from 1970 for the FIX run? Later you compare the NPP results to observation-based NPP from after 2000. Wouldn't it be better to use the 2000 fluxes for the FIX run?*

In Section 3.3, we assess the effect of future changes in riverine inputs by comparing the future period 2050–2099 to the recent past reference period 1950–1999. This assessment was one of our main objectives when we designed the experiments. We chose the 1970 fluxes because they are more representative for the 1950–1999 period than the 2000 fluxes (Beusen et al., 2016).

We agree with the reviewer that the use of 2000 fluxes for the FIX run would have been preferred when comparing with observation-based NPP from after 2000. In hindsight, choosing 1950–1999 as a present day reference was not an optimal choice considering better availability of observations estimates during the early 21st century.

We have mentioned the above as a caveat in the discussion of the revised manuscript version as below. To be more precise, we also changed the wording "contemporary" or "present day" to "recent past" in various places.

"Our spin-up experiment uses riverine nutrient and carbon inputs fixed at 1970 levels, as provided by NEWS 2. As a caveat, our post-1970 simulated changes in marine PP and $CO_2$ fluxes miss out any legacy effects from riverine transport changes that occurred before 1970. However, Beusen et al. (2016) found that changes in riverine N and P are relatively small before 1970 compared to changes after 1970. Therefore, we expect the error due to missing legacy effects to be minor.

In FIX, we applied riverine inputs at 1970 level over available inputs at 2000 level, because the former are more representative for the 1950–1999 baseline period that we used for future projections. However, the use of 1970 level inputs is suboptimal when evaluating simulated PP and $CO_2$ fluxes against observations obtained after 2000."

- *As mentioned above, nutrient fluxes often do not scale at all with runoff as concentrations can decrease strongly when discharge increases. Furthermore, apart from DOM and DIP the global news scenarios all give very different future scenarios compared to RUN. I think the simulation RUN hence really makes very little sense and I do not know what its value is. I would certainly not make such strong statements in the Discussion (line 321). I am happy to be convinced otherwise.*

We agree with the reviewer that the RUN experiment has its uncertainties. It is an idealistic scenario, which assumes that future changes in riverine carbon and nutrient transports are directly linked to changes in riverine freshwater transports, ignoring other anthropogenic effects related to land surface processes. The comparison of RUN to the GNS experiments demonstrates that the anthropogenic and natural changes in nutrients and carbon on land are equally or more important than the direct effect of the changes in the hydrological cycle. Including RUN along with the more realistic GNS scenarios thus still provide valuable information, which e.g., may caution other modelling groups against adopting an over-simplified coupling of riverine nutrient and

carbon transports to the hydrologic cycle. Another motivation for RUN was to introduce seasonal and interannual variability in nutrient and carbon transports that is linked to variability in riverine freshwater transport. Future work should explore if GNS and RUN can be integrated to produce more realistic long-term trends in riverine nutrient and carbon transports as well as short-term variability.

We have clarify that to explore the best practical way of implementing the riverine inputs for modelling groups is one of the aims of this work. We have modified the statement in line 321 in the original version (in section 4.2 in the updated version).

- *It seems that large changes are always simulated in the Black Sea. However, results from ESMs in enclosed or semi-enclosed seas usually make no sense. Did you mask these seas, including the Mediterranean Sea? If not, how does masking these seas change your results?*

The reviewer is right that in global models the results in those enclosed or semi-enclosed seas are often largely biassed. We did not mask those seas during simulation, but we masked Black Sea and Caspian Sea (not Mediterranean Sea) for plotting. The model result shows that the Black Sea, e.g., accounts for 1.7% of the global increase in PP due to riverine inputs (FIX-REF) during 2003–2012. As we said, this might be largely biased, therefore, we have excluded those seas from our calculation.

- *I am really struggling with the significance of the results. For example, in line 192, I would like to see the inter-annual variability as a measure of the standard deviation to see how much they are really different. Similar, is there an uncertainty estimate for the observation-based estimates in line 197? Overall, the differences in annual NPP and RMSE seem to be so small that I am not sure if it makes sense to use terms such as 'better' or 'improve'. Can you find any statistical way to evaluate if the changes are significant? This comment should be addressed to other numbers throughout the manuscript. Please be also careful with the word 'significant' as used in line 238 if it has no statistical meaning.*

We thank the reviewer for the comment. In the revised manuscript, we followed up the suggestions and added statistical robustness information for the numbers in the main text (global and regional integrated values) and to the spatial difference maps.

The robustness assessment is detailed in the new Appendix B. The new Tables B1 and B2 summarize the numbers of the main text (absolute values and absolute changes) and complement the values with standard-deviations. In this manner, we provide statistical robustness information to "all numbers throughout the manuscript". In the main text, we refer the reader to the tables B1 and B2 for supporting robustness information. The main text does not discuss any changes that are not found statistically significant.

As the reviewer suggested, we assessed the statistical robustness of the results with respect to sampling of interannual variability. We assessed the robustness of differences with Student's t-test and found all values stated in the manuscript text significantly different from zero at a 95% confidence level. Additionally, we provide the standard-deviation of time-mean that we computed as the standard-deviation of annual series divided by the square root of the number of years (if one wants to compare

directly against the interannual standard-deviation one merely needs to multiply with the square root of the number of years).

We have added robustness information also to the spatial difference maps. Time-mean difference values are plotted only if they passed a t-test along the year-dimension. Additionally, areas are marked with dots where the signal is larger than the standard-deviation of the mean of the absolute field featuring the full interannual variability (not the difference field which has interannual variability largely removed). In the marked regions, the signal is not only detectible in the idealized model world (in which the interannual variability removed) but also large enough to be competitive with real-world internal variability to have potential real-world implications and be detectible in observations.

We moved the last paragraph of original Section 2.3—on the lack of biogeochemistry feedback onto the physical climate and implications for the comparison of simulations—to Appendix B. As stated in this paragraph, we expect even small values to be statistically significant because the interannual climate variability is the same in all simulations and thus most of the interannual signal is removed in the computation of the experiment differences. In the revised manuscript, we have illustrated this behaviour with Figure B1 that shows two time series from two different experiments in one panel and the time-evolving difference of these series in another panel, together with confidence intervals of the time-means. The figure clearly shows the dramatic reduction of interannual variability following the computation of the difference time series.

Concerning the observation-based estimates, Carr et al. (2006) have assessed 24 ocean-color-based models (including different variations of VGPM) and reported that the mean global PP estimated from those models for six months of 1998 ranging ~35-70 Pg C yr$^{-1}$, i.e., varying by a factor of two between models. In the paper by Westberry et al. (2008), they have discussed the uncertainties of the CbPM based on the global PP values spanning a range of a factor of ~1.5, from 44 to 67 Pg C yr$^{-1}$. We have stated the inter-product range 55 to 61 Pg C yr$^{-1}$ between the 3 different satellite-based estimates that have been used in our study.

- *Please refrain from making strong claims about the Arctic. Indicating a ~76% increase in NPP is misleading giving how bad the model simulates the present-day NPP. Based on an observation-based NPP of 450 TgC/yr, a change of 70-80 TgC/yr is only an increase in 17%. Moreover, the very low present-day NPP suggests either strong light or strong nutrient limitation. If it is strong nutrient limitation, riverine fluxes would have an overly strong effect because all nutrients would be used immediately. So maybe even the 17% are still too high. This goes back to the point that the reader must know how good the model performs locally.*

We agree that stating a relative change (in percentage) could be misleading given the biassed low primary productivity simulated in our model. Therefore, we have removed the relative change in the text. As with other Earth system models, the increased Arctic primary production in NorESM, which is not strongly nutrient limited, is driven by light and temperature constrain, associated with sea-ice loss (stated in the discussion). Our projected PP increase in the reference run is roughly 70 Tg C yr$^{-1}$ by the end of the 21st century. This is in the same order of magnitude estimated from 11 ESMs with mean change of 59 Tg C yr$^{-1}$ (individual ESM ranges from -110 to +253 Tg C yr$^{-1}$;

Vancoppenolle et al., 2013). We have added a new subsection (Section 2.2) on model evaluation, especially in the Arctic Ocean.

*General comments*

- *Often 'biogeochemistry is used as a synonym for PP and air-sea CO2 fluxes. But the word biogeochemistry also includes acidification, carbon and nutrient cycles, and other things. Please just say PP and air-sea CO2 fluxes. ( for example line 250).*

We have made respective specifications throughout the manuscript in order to be clear in these occasions.

- *Please adhere to the best practice guide (https://www.ncei.noaa.gov/access/ocean-carbon-data-system/oceans/Handbook_2007/Guide_all_in_one.pdf) and use CT and AT instead of DIC and ALK.*

The best practices guide by Dickson et al. (2007) refers to best practices for measurements. We have checked 10 biogeochemical modelling studies (see the table below), and in nine of them DIC was used, while TCO2 was used in the remaining one. Concerning alkalinity, the acronym used in these studies are more diverse. ALK was used in two studies, while TA was used in three studies, and in the rest studies either no acronym was used for alkalinity or a mixture of TA, ALK or TALK was used. It seems to us that CT and AT are not widely used in modeling studies, although they are standard acronyms in observational fields. Therefore, we decided to replace ALK by TA but prefer to keep DIC as many of the modelling studies did.

| Model | Acronym used for dissolved inorganic carbon and total alkalinity | Reference |
|---|---|---|
| JCOPE_EC | DIC, ALK | Ishizu et al., 2020 |
| CMOC | DIC, TA | Zahariev et al., 2008 |
| CSIBv1 in NEMO | DIC, TA | Hayashida et al., 2019 |
| PISCES-v2 | DIC, alkalinity | Aumont et al., 2015 |
| TOPAZ2 in GFDL-ESM2.1 | DIC, ALK | Dunne et al., 2013 |
| COBALTv2 in GFDL-ESM4.1 | DIC, alkalinity | Stock et al., 2020 |
| NASA-GISS | DIC, alkalinity | Romanou et al., 2013 |
| Diat-HadOCC | DIC, TALK | Totterdell, 2019 |
| MEDUSA-1.0 | DIC, TA/ALK | Yool et al., 2013 |
| HAMOCC in MPI-ESM | TCO2, TA | Ilyina et al., 2013 |

- *I find the name 'reference run' misleading. It is rather a control run. Reference should be the best case or something.*

The term "reference experiment" is widely used in the Earth system modelling community in the way used in our study. It is standard for ESM modelling studies to have a "reference" experiment and one or several "sensitivity" experiments. The response is typically evaluated as the difference sensitivity minus reference. The term "control experiment" is typically reserved for a simulation where external forcings are fixed at pre-industrial (or some other) levels (e.g., in the work by Hajima et al., 2020). However, the external forcings used in this study are transient ones, therefore using "control" may confuse readers.

Furthermore, the original NorESM1-ME (Tjiputra et el., 2012) and many other ESMs do not include riverine nutrient and carbon transports to the oceans. Therefore, we find the use of "reference" for the experiment without transports appropriate in our study and prefer to keep it in the revised manuscript version.

- *Significant digits should always be the same. For example, in lines 204 and 205 you cite air-sea CO2 fluxes and use different number of digits.*

We have checked and changed all numbers accordingly.

*Minor comments*

- *Lines 19/20: Can one speak of improve based on such small changes in the RMSE? Is it significant?*

Referring to the response to major comment on the significance, we performed a statistical robustness analysis and have found the temporal sampling error well below the obtained change in RMSE (see new tables B1 and B2 and details in Appendix B).

- *Line 13: Suggest changing "not only regionally but also globally" to "regionally and globally"*

We changed it accordingly.

- *Line 18: Suggest changing "modelled" to "simulated"*

We changed it accordingly.

- *Line 22: Unclear what you mean by depending on the riverine configuration. There is no range in the numbers given in this sentence.*

We meant that by adding the riverine nutrients input, the projected future decline in PP can be alleviated maximum by 0.6 PgC yr-1 (27.3%) globally in our experiments. The range is given in Conclusion in line 367-369: "Riverine nutrient inputs into surface coastal waters alleviate the nutrient limitation and considerably lessen the projected future decline in PP from -5.4% without riverine inputs to -4.4%, -4.1% and -3.6% in FIX, RUN and GNS (averaged over four scenarios), respectively."

In order to avoid ambiguity, we removed "depending on the riverine configuration" from the original text.

- *Lines 23/24: the last part of the sentence should be rewritten*

We reformulated it to "The riverine impact on projected C uptake depends on the net effect of riverine nutrient induced PP increase and riverine C input induced outgassing."

- *Lines 22-25: A lot of words that do not tell much. Nutrients increase CO2 uptake, CT fluxes decreases it. But where does it increase it and where does it decrease it? Maybe shorten this or explain.*

We have reformulated it. Please see the response to the last comment.

- *Line 26: Can you be more quantitative?*

We have changed the later part of the abstract to "Riverine nutrient inputs lessen nutrient limitation under future warmer conditions as stratification increases, and thus lessen the projected future decline in PP by up to $0.7 \pm 0.02$ Pg C $yr^{-1}$ (29.5%) globally, when comparing 1950–1999 with 2050–2099 period.  The riverine impact on projected C uptake depends on the net effect of riverine nutrient induced C uptake and riverine C input induced $CO_2$ outgassing. These two opposite impacts are comparable in magnitudes when they are globally integrated. Therefore, in the two idealized riverine configurations the river inputs result in a weak net C sink of $0.03–0.04 \pm 0.01$ Pg C $yr^{-1}$, while in the more plausible riverine configurations the river inputs cause a net C source of $\sim 0.1 \pm 0.03$ Pg C $yr^{-1}$. The results are subject to model limitations related to resolution and process representations that potentially cause underestimation of impacts. High-resolution global or regional models with an adequate representation of shelf processes should be used to assess the impact of future riverine scenarios more accurately."

- *Line 31: Not sure if you can count runoff.*

We changed it to "river runoff plays an essential role in …".

- *Lines 31/32: transporting nutrients where? Suggest adding "into the ocean" after "transporting nutrients".*

We added it accordingly.

- *Line 34: What do you mean by "absolutely dominant" source? More than 50%? Please be clear.*

We added the information as follows. "For some substances such as total phosphorus (~90.0%) and total silicon (>70.0%), riverine input even acts as the absolutely dominant source."

- *Line 34: Suggest adding "into the ocean" after "transport of carbon".*

We added it accordingly.

- *Line 34: Suggest writing air-sea CO2 exchange instead of air-sea C exchange. It CO2 and not C that is exchanged across the air-sea interface.*

We changed it accordingly.

- *Line 36: What is "it"?*

We changed this sentence to "Despite our limited understanding on the  riverine carbon fluxes, they could play an important role in closing the global carbon budget (Friedlingstein et al., 2021) and could be very sensitive to regional and global changes

such as weathering, land cover and climate (Meybeck and Vörösmarty, 1999)." They refer to riverine carbon fluxes.

- *Line 36: Do you mean global ocean carbon cycle or really global carbon cycle?*

We have removed this term. Please see above.

- *Line 36: Global and regional changes of what?*

Please see the response above.

- *Line 37: Suggest starting a new paragraph here*

We followed the suggestion in the revised version.

- *Line 55: In the Arctic, Terhaar et al. (2019) started to assess future changes.*

Thank you for the reference. We have included it in the introduction. However, Terhaar et al. (2019) have assessed the impact of idealized changes (increased by 1% per year until doubling) in riverine carbon and nutrient delivery on PP, $CO_2$ fluxes and acidification, which is kind of sensitivity study, using historical climatology forcings. Their simulations did not extend to future time; therefore, they did not assess future projected changes in PP and $CO_2$ fluxes.

- *Line 56: Please say which datasets you are referring to.*

We added "Global NEWS 2 and GLORICH" here.

- *Line 58: Why does more data make the impact study more 'desirable'?*

"Taking the advantage of the latest improvement of global river nutrient/carbon export datasets, e.g., Global NEWS 2 (https://marine.rutgers.edu/globalnews/datasets.htm) and GLORICH (https://doi.pangaea.de/10.1594/PANGAEA.902360), and responding to the demand of development of ESMs with increasing model resolution, the assessment of the impact of riverine nutrients and carbon on future projections of marine biogeochemistry becomes feasible and desired, especially for impact studies along continental margins."

Here more data makes the assessment feasible, and increasing model resolution makes the assessment desirable.

- *Line 58: Please say why it is now feasible? One could argue that the CMIP6 horizontal resolution is still not good enough to resolve the global ocean.*

More feasible refers to the latest development of riverine datasets. Please see the response above.

- *Line 68: Please already say here why you use RCP4.5.*

We changed it to "the RCP4.5 (middle-of-the-road) scenario" here. We put the detailed reason for the choice of the future scenario in section 2.4 "Experimental design", which is a suitable place for that. We also added references to support our choice. Please also see the response to comment on Line 153 below.

- *Line 84: Configured sounds strange here.*

We changed "configured" to "implemented".

- *Line 88: 'd' is missing in 'based'.*

We have changed it.

- *Lines 88-113: Does the ocean biogeochemical component also have a name? It is very confusing that you say it is based on HAMOCC and then you only describe HAMOCC. That makes it impossible to understand what has changed.*

In the NorESM1-ME that we have used in this study, the biogeochemical model is still called HAMOCC5, even though its development diverts from the original HAMOCC5 version, which was developed in Hamburg. However, in the newer version of NorESM2, the biogeochemical model is renamed as iHAMOCC.

- *Line 129: What is the motivation to use CT and AT data from Hartmann (2009)? What kind of data is that? Modeled, observed, extrapolated? What is the underlying data? Please explain what you use.*

The reason that we used CT and AT data from Hartmann (2009) is that the Global NEWS2 dataset does not include CT and AT data. The CT and AT data from Hartmann (2009) are produced from a high-resolution model for global $CO_2$-consumption by chemical weathering. The dataset contains different forms of riverine carbon (dissolved and particulate, inorganic and organic), implemented to the Global NEWS2 river basin map. We have added this information in section 2.3.

- *Lines 129-133: Why do you use iron riverine fluxes from 1990? Is there nothing newer including observations since 1990? Does it make sense to weight iron river fluxes by runoff? Often nutrients do not scale at all with runoff (Holmes et al., 2012), so I would like to see some support for this assumption.*

To the best of our knowledge, the available global riverine iron dataset is rare. Previous studies have used various approximation approaches, e.g., constant Fe to dissolved inorganic carbon (DIC) ratio (Aumont et al., 2015), Fe to phosphorus ratio (Lacroix et al., 2020). In the study by Aumont et al. (2015), the Fe: DIC ratio is determined so that the total Fe supply also equals 1.45 Tg Fe $yr^{-1}$ as estimated by Chester (1990). We are aware that our approximation likely has bias in regional scales, especially in Fe limiting regions like the Arctic. However, it has likely a minor impact on the projected PP, since it is the light rather than riverine nutrients input which controls the projected PP in the Arctic in our model. We have discussed this in the Limitation section 4.3.

- *Line 135: Is there a reason why you use 1000 km and 300 km here?*

The short answer is that NorESM1 is based on NCAR's Community Earth System Model (CESM; Hurrell et al., 2013). The configuration for distributing riverine runoff into ocean grid cells along with the 1-degree resolution ocean grid have been both inherited from CESM without modification.

The exact reasoning behind the NCAR's choice of using a 1000 km e-folding length scale and 300 km cutoff value is not known to us. The effect of using an e-folding length scale that is considerably larger than the cutoff value is that approximately equal weights are used when distributing the runoff to ocean grid cells that lie within the cutoff range of the river mouth. The cutoff value must be large enough that at least one ocean grid cell midpoint lies within the range. For a 1-degree resolution ocean grid, a value of 100 km should be sufficient to satisfy this. Possibly, the large value of 300 km was chosen to also satisfy coarser grids, such as the 3-degree resolution grid used in NCAR's computationally efficient model configuration, and to avoid numerical instabilities by more smoothly distributing the runoff.

In this study, we ensured that riverine nutrients and carbon are distributed in the same way as riverine freshwater. We considered this important especially for the RUN

configuration that couples the variability of the riverine nutrient and carbon transports to the variability of the riverine freshwater transport. How sensitive the ocean biogeochemistry impacts are to the details of how riverine runoff is distributed into coastal grid cells and generally how well ocean shelf processes are represented warrants further investigation.

- *Line 145: Any reason why not GLODAPv2 is used?*

Our NorESM1-ME model configuration was finalised in the early 2010s, and GLODAPv2 was not available at that time. While there are improvements in the GLODAPv2 due to higher volume of data and improved interpolation scheme (Lauvset et al., 2016), since the model was then spun up for nearly 1000 years, we expect that these differences in initialization will not be significant for our purpose.

- *Line 147: Is the additional spin-up of 200 years is sufficient to get into a new equilibrium.*

The nutrient drift after 200 years spin-up is small (in the order of 1%/100 years).

- *Line 153: In what sense is RCP4.5 the most representative scenario? Most likely? Based on what?*

We rewrote this sentence as follows. "Here, we consider RCP4.5 as the representative future scenario following the $CO_2$ emission rate based on the submitted Intended Nationally Determined Contributions, which projects a median warming of 2.6-3.1°C by 2100 (Rogelj et al., 2016)." We have also added a sentence in the discussion (section 4.3) about its uncertainty.

- *Line 160: What means not considered? Deactivated?*

We changed "not considered" to "deactivated".

- *Lines 168-170: It is not entirely clear that you make 4 simulations. Can you be a little bit more explicit?*

We have changed it to "GNS: Four different transient inputs following future projections of NEWS 2."

- *Line 194: In the figure it does not look as if only 15% of the increase is in the coastal shelf seas. What do you mean by predominantly, can you be quantitative here?*

We have checked the number for increase on the shelves (it is 15.4% as we changed significant digits), and quantified the increased primary production in the North Atlantic and reformulated the text as follows: "The increase of PP in FIX occurs along continental margins (where seafloor is shallower than 300 m) and also in the North Atlantic region (0°N-65°N, 0°W-90°W), accounting for 15.4% and 24.9% of the global total increase, respectively (Figure 3c)."

- *Line 214: Is this a result or a speculation. If you cannot prove it, I suggest to either add it to the Discussion and add literature that supports this point or delete it.*

This is not a direct result. We removed it.

- *Line 244-245: Does it really make sense to say slightly higher if the difference is that small?*

We have added more information on the robustness of changes in the revised manuscript. Please also see the response to the comment on the significance of the results.

- *Line 272: Please be quantitative*

We changed it to "The projected global total PP shows up to 29.5% less decrease, if riverine inputs are present in the model."

- *Line 279: Is not nitrogen the limiting nutrient in the Arctic Ocean (Tremblay et al., 2015)?*

Our model shows that a large area of the Arctic Ocean is indeed nitrogen limiting, while in some parts iron is limiting.

- *Line 286: Why do you not add CMIP6 data?*

We have added comparison with CMIP6 results (Kwiatkowski et al., 2020; Tagliabue et al., 2021) in the revised version.

- *Why do you speculate in the Discussion in lines 290 to 293: You can show that with your model results.*

Thank you for the comment. The statement was inferred from model results, i.e., FIX simulation produces higher NPP than REF simulation, which suggests that riverine-induced nutrient addition alleviate the stronger nutrient limitation in the future. We have rephrased the sentence to clarify this and referred to the Figure A2b.

- *Figure 4c is almost impossible to read.*

We have improved this figure by smoothing the contour lines and using grey lines instead of green.

- *Figure 5a: Could you highlight the +38 more? I was confused first.*

We have changed "+38 [Pg C yr$^{-1}$]" in bold and added a note in the figure caption as follows: "note that the positive numbers in the y axis (marked with stars) are scaled by minus 38 Pg C yr$^{-1}$ so that the negative numbers are visible".

**Reply to Anonymous Referee #2**

**We are grateful for the comments and edits of the anonymous reviewer who invested time for the review of our manuscript. The reviewer's text is reported in Italic and our responses in roman.**

*Using a global ocean biogeochemical model, the authors assess the future impacts of changing riverine inputs while performing simulations with several scenarios of riverine inputs. In the paper, the authors show a slight reduction in root mean square error of primary production with respect to observations on the continental shelf and in parts of the ocean highly affected by rivers. They show that, overall, riverine nutrients may alleviate part of projected primary production decline. The topic is very relevant and potentially important for fully constraining the ocean carbon cycle. However, while the modelling work is very sound in its closed framework, I believe the author's review of literature seems to be lacking, which leads to, at times, questionable assumptions and too high confidence in a model setting not necessarily adequate for investing impacts of riverine tracers and their fate in the ocean. I do think the study should be published, but the authors should discuss the following points and, in my opinion, clearly point out uncertainties related to their framework in the abstract.*

We thank the reviewer for this input. We have extended the literature review with relevant global and regional studies and added more discussion on the sensitivity of our results to model and data limitations, including more discussion on the assumptions and adequacy of our model setting. The abstract now clearly points out that the results are subject to uncertainties related to model resolution and process representations and should be reassessed using models with improved spatial resolution and coastal process representations. We have implemented suggestions and addressed concerns from the specific comments of the reviewer. More details are provided in our replies to the reviewer's specific comments below.

*General Comments*

*It is debatable that the model configuration used here is actually adequate for the research question addressed in this paper. The model is limited in terms of resolution (~1 degree), and this is a strong constraint for representing fine-scale circulation features that are thought to be of particular importance in the coastal ocean. This is a topic that many studies have discussed previously, and these should be considered. Secondly, the model doesn't consider specific biogeochemical processes relevant to the coastal ocean. For instance, organic matter decomposition rates are shown to be much higher in the coastal sediment than in the open ocean (Ardnt et al., 2013), bed shear stress also re-suspend biogeochemical from the shelf seafloor. I think omitting both of these physical and biogeochemical limitations lead to the important underestimation of primary production on river-dominated continental shelves shown in this study, which seems to indicate the model is underrepresenting. This also affect exports of riverine biogeochemical compounds to the open ocean, and thus this has very important consequences for the main outcomes presented in this study. The authors do mention these limitations briefly in the limitations section, but this should be considered omni-present in attempts to interpretate the results, and at the least mentioned in the abstract.*

Thank you very much for the comment. We are fully aware that in a coarse resolution model, not all physical and biogeochemical processes can be reproduced in detail and this has consequences on the model results, which have been mentioned in the limitations section. However, we think that the study is useful nevertheless, in order to see the effect of an addition (or omission) of respective land-sea fluxes of chemical compounds within the coarse resolution (~1 degree) context. Especially since such models are still regularly employed in IPCC reports on projecting future ocean carbon cycle response to climate change. We will clarify that the goal of this study is not to simulate the distributions of geochemical tracers near the coast precisely, but rather to assess the large-scale impact of adding another layer of (relatively simplistic) continental margin-to-open ocean biogeochemical process. Further, to explore the best practical way of implementing the riverine inputs for NorESM is also one of the aims of this work.

We have modified the last part of the abstract to: "The riverine impact on projected C uptake depends on the net effect of riverine nutrient induced C uptake and riverine C input induced $CO_2$ outgassing. These two opposite impacts are comparable in magnitudes when they are globally integrated. Therefore, in the two idealized riverine configurations the river inputs result in a weak net C sink of 0.03–0.04 ± 0.01 Pg C yr$^{-1}$, while in the more plausible riverine configurations the river inputs cause a net C source of ~0.1 Pg C yr$^{-1}$. The results are subject to model limitations related to resolution and process representations that potentially cause underestimation of impacts. High-resolution global or regional models with an adequate representation of shelf processes should be used to assess the impact of future riverine scenarios more accurately."

At the end of the introduction, we haved added: "Another objective of the study is to explore the best practical way of implementing riverine inputs into newer versions of NorESM. Because of the coarse resolution of the version used here, a series of processes in the coastal zone cannot be represented in our study such as the high accumulation of organic sediment in shallow waters and respective remineralization rates of previously deposited material (Ardnt et al., 2013; Regnier et al., 2013). These processes can only be presented in models of much higher spatial resolution, which are at present too costly to be integrated long enough to simulate the large scale water masses adequately and project long-term scale climatic change. Given missing contributions from unresolved processes, our results are to be interpreted as lower bound estimates."

*A second, perhaps less central point, but still relevant to the study, is that the authors spin-up their model to present day fluxes, whereas these are actually more strongly perturbed over the historical time period (Beusen et al., 2016), than what is projected in terms of their future changes. Since time-scales of the ocean carbon cycle are notably long, this historical perturbation could have important legacy effects propagating into the future, potentially enhancing the primary production more than is estimated here. This should, in my opinion, also be discussed in the limitations section.*

Thank you for the comment. In the revised manuscript, we changed the text to "fixed at recent-past level", i.e., constant fluxes at 1970's level, instead of "contemporary level". Beusen et al. (2016) showed in their Figure 3 that the nutrient fluxes did not vary much before 1970. We agree that by using constant riverine fluxes rather than transient

fluxes to spin-up the model, our results will miss out potential legacy effects from flux changes that date before 1970. We have added following discussion about this in the limitations section: Our spin-up experiment uses riverine nutrient and carbon inputs fixed at 1970 levels, as provided by NEWS 2. As a caveat, our post-1970 simulated changes in marine PP and $CO_2$ fluxes miss out any legacy effects from riverine transport changes that occurred before 1970. The fixed inputs likely overestimate the accumulated inputs prior 1970, causing potential underestimation of the projected change impacts. However, Beusen et al. (2016) found that changes in riverine N and P are relatively small before 1970 compared to changes after 1970. Therefore, we expect the error due to missing legacy effects to be minor.

*Specific Comments:*

*Abstract*

- *L16 "With four riverine configurations: deactivated, fixed at a contemporary level, coupled to simulated freshwater runoff, and following four plausible future scenarios." Are only the nutrients (and if yes which ones) changing, or also carbon and alkalinity? This should be stated here.*

  We changed it to "...with four riverine transport configurations for nutrients (nitrogen, phosphorus, silicon and iron), carbon and total alkalinity: deactivated, fixed at a recent-past level, coupled to simulated freshwater runoff, and following four plausible future scenarios."

- *L17 "The inclusion of riverine nutrients and carbon…" Those numbers are valid for contemporary I guess?*

  We changed it to "The inclusion of riverine nutrients and carbon at 1970's level improves the simulated contemporary spatial distribution relative to observations…"

- *L20 "Riverine nutrient inputs alleviate nutrient limitation,…" Should be reformulated, since riverine nutrient inputs are unlikely alleviated nutrient limitation in general, but reduce (?) it in some regions (?).*

  We changed it to "Riverine nutrient inputs lessen nutrient limitation under future warmer conditions as stratification increases, and thus lessen the projected future decline in PP…". It is not feasible to put detailed regions in the abstract due to word limit, but we have explained the regions in detail in section 3.2.

*Introduction*

- *In general, there are very little citations in the introduction, and often the same ones are used repeatedly. There are some recent modeling studies of implications of riverine inputs in the ocean that would be very relevant for this study. These should, in my opinion, be considered in the introduction:*

*Lacroix, F., Ilyina, T., Mathis, M., Laruelle, G. G., & Regnier, P. (2021). Historical increases in land-derived nutrient inputs may alleviate effects of a changing physical climate on the oceanic carbon cycle. Global Change Biology, 27, 5491– 5513. https://doi.org/10.1111/gcb.15822*

*Liu, X., Stock, C. A., Dunne, J. P., Lee, M., Shevliakova, E., Malyshev, S., & Milly, P. C. D. (2021). Simulated global coastal ecosystem responses to a half-century increase in river nitrogen loads. Geophysical Research Letters, 48, e2021GL094367. https://doi.org/10.1029/2021GL094367*

*I would furthermore suggest citing some regional-scale studies that investigate implications of riverine inputs on biogeochemistry of specific shelves, literature is abundant here. In addition, I would read and refer to the last 2-3 Global Carbon Budget studies for potential importance of riverine carbon fluxes for the ocean.*

Thank you very much for the suggestion and the references. We have added a new paragraph on the modelling studies on the impact of riverine inputs in the introduction and thereby included the above mentioned literature as well as some other ones (e.g. Lacroix et al., 2020; Tivig et al., 2021), and have also added more references on regional studies, e.g., Arctic (Letscher et al., 2013; Le Fouest et al., 2013, 2015, 2018; Terhaar et al., 2019; Anderson et al., 2009; Manizza et al., 2011), South America rivers (van der Struijk and Kroeze, 2010) and Changjiang River (Yan et al., 2010).

- *L25 "The large range of the riverine input across our four riverine 26 configurations does not transfer to a large uncertainty of the projected global PP and ocean C uptake…" In terms of global PP, one could argue this could be due to the representation of continental shelf in the model, which leads to heavily underestimated continental shelf PP.*

  We agree completely with the reviewer on this point and we have changed this sentence to "Simulations with high-resolution global or regional models with an adequate representation of shelf processes are required to accurately assess the impact of future riverine scenarios."

  We are fully aware of the underrepresented shelf process issue and the underestimated coastal PP in coarse-resolution models. We have discussed those issues in section 4.3 (Firstly, poorly represented physical shelf processes, as well as uncertainties in biogeochemical dynamic. For example, conversion of organic to inorganic carbon occurs rapidly via remineralization in estuaries before they are transported to the open ocean. Secondly, coarse-resolution models tend to underestimate primary production along the coast. Such well-known model issues may limit the impact induced by riverine inputs). We have pointed out in section 4.2 that "However, the scenario differences might be of importance in regional projections, such as in seas surrounded by highly populated nations and/or near river estuaries."

- *L35 "Although riverine carbon only plays a minor role in the global carbon cycle, …" Recent Global Carbon Budget publications disagree with this (Friedlingstein et al., 2021). If the higher estimates of outgassing of riverine carbon are true (up to 0.8*

*Pg C yr-1), they could potentially play a large role in explaining discrepancies between CO2 estimates arising observation-based products and model-based results.*

We changed this sentence to "Despite our limited understanding on the riverine carbon fluxes, they could play an important role in closing the global carbon budget (Friedlingstein et al., 2021) and could be very sensitive to regional and global changes such as weathering, land cover and climate (Meybeck and Vörösmarty, 1999)."

- *L44 Maybe add the more recent Beusen et al. (2016) estimates to this for the historical time period?*

We added the following sentence in the revised version: "Beusen et al. (2016) estimated that river nutrient transport to the ocean increased from 19 to 37 Tg N yr$^{-1}$ and from 2 to 4 Tg P yr$^{-1}$ over the 20th century, taking into account of both increased nutrient input to rivers and intensified retention/removal of nutrients in freshwater systems."

*Methods*

- *L118 "The riverine influx includes carbon, nitrogen and phosphorus, each in dissolved inorganic, dissolved organic, and particulate forms, as well as alkalinity (ALK), dissolved silicon and iron (Fe)." Are there specific ocean variables for terrestrial dissolved and particulate organic matter? If not how does not model deal with organic P-N-C ratios that differ from those of the Redfield ratio? This is an important point to clarify because high C-to-nutrient ratios are thought to be largely responsible for ocean outgassing.*

In the model there is one dissolved organic pool (DOM) and one particulate organic pool (DET, detritus). First, we calculate the riverine organic P-N-C ratios for both dissolved and particulate forms, then add the least abundant species (scaled by the Redfield ratio) to the DOM and DET pools, respectively. The excess budget from the remaining two species both in dissolved and in particulate forms are assumed to be directly remineralized into inorganic form and added to the corresponding dissolved inorganic pools (i.e., DIP, DIN, and DIC) in the ocean. We have elaborated on this in the revised version in Section 2.3.

- *L140 "Any remaining riverine organic matter is then added to its inorganic pool" This is not really clear. Is excess organic carbon is added to the DIC pool? If not I think this might be the reason why river inputs cause a net sink in the model, and not source as is relatively well acknowledged (see e.g., Global Carbon Budget, 2021). Also keep in mind that organic carbon mineralization has a small effect on alkalinity (which I don't think would have a huge impact here).*

Please also see the response to the last point. Excess DOC is indeed added to the DIC pool, but yet the riverine fluxes in the model lead to a net C sink, which might be due to the overestimation of the organic-to-inorganic conversion of excess nutrients.

*Also, maybe more important here: are you assuming the large particulate fluxes (particulate P and N) from NEWS2 are organic? Because from my understanding,*

*these can be inorganic (for P bedrock erosion, occluded etc..), and this would not at all be bio-available in the coastal ocean.*

Although the Global NEWS2 data provide the total particulate N and P rather than differentiated inorganic and organic particulate forms, particulate N occurs largely as organic matter while particulate P is typically dominated by inorganic forms (Mayorga et al., 2010). The reviewer is completely correct in stating that particulate P is mostly inorganic and not directly bio-available. Thus, adding the remaining particulate P (after calculating the least abundant species according to the Redfield ratio) into dissolved inorganic P pool may lead to bias in the enhanced primary production. Along the same line, adding the remaining riverine dissolved organic matter into the corresponding dissolved inorganic pool may also partly lead to bias in the enhanced primary production. Therefore, we have calculated the range of the impact of the directly remineralized dissolved organic and particulate matter on the enhanced primary production, by comparing with the corresponding riverine dissolved nutrients [X/(X+DIXriv)*100%] (X is the directly remineralized dissolved organic and particulate matter). Assuming that all coastal regions are nutrient limited, this direct remineralization could be potentially responsible for approximately 33.3-80.5% of the enhanced primary production (33.3% and 80.5% for nitrate and phosphate limiting, respectively). However, in our model, phosphate is rarely limiting (Figure A1), therefore, the impact of this direct remineralization on primary production is likely on the lower end of this range.

We have elaborated on this point in the discussion Section 4.3 in the revised version.

- *L156 "REF: Reference run. Riverine nutrient and carbon supply is deactivated." Are there other sources of nutrients and carbon in the model? If riverine nutrients and carbon were the only inputs to the ocean model and their sediment loss is non-zero, I would expect all related variables to thrive to zero, which does not make a very interesting reference run. In the case there are other inputs, they should be given in numbers and explained.*

The only external inputs of nutrients are from aerial dust (iron) deposition and nitrogen fixation.

The REF simulates primary production and ocean $CO_2$ uptake evolution under climate change, without riverine input. It is interesting to see that the effect of riverine inputs on primary production is different between the historical and future time period (due to a different nutrient depletion level). This assessment is only possible to make by subtracting the climate effect on primary production in REF.

- *L179-L185 In my opinion the authors don't need to specifically defend themselves on this particular point, at least not to this extent. I would consider shortening or removing.*

We agree that the lack of feedback from biogeochemistry on climate in our model setup is not a major limitation of our study that requires much justification. We mainly mentioned it to explain why we were able to obtain statistically robust results from single simulation experiments. In the revised manuscript, we have moved this part from

the main part of the manuscript to the new Appendix B, where we (on the first reviewer's request) address in more detail the statistical robustness of our results.

*Results*

- *L198 "Although the total PP in FIX is still considerably lower than the satellite-based estimates, the inclusion of riverine nutrients and carbon does slightly improve the distribution of PP especially on continental margins (Figure 3), according to our area-weighted root mean square error (RMSE) analysis. „*

*Figure 3 really shows that a large part of the underestimation of PP is originating from the continental shelf, in particular regions of riverine inputs. The improvement is minor compared to the actual bias. In my opinion, this actually shows that the model underestimates the impacts of rivers on PP, which does have a strong implication for the conclusions of this paper, and should be assessed somehow.*

We agree with the reviewer that the riverine nutrient input (at 1970's level) does not largely improve the simulated contemporary primary production along the continental shelf. The reasons could be:

 1) The underestimation of primary production on continental margins due to coarse model resolution and unresolved shelf processes is larger than the impact of riverine nutrient input. Therefore, it could not be compensated.

2) The time period of contemporary primary production that we look into is 2003-2012, but the riverine nutrient input in the model is at 1970's level. Beusen et al. (2016) have shown that the riverine nitrogen and phosphorus has increased by ~40.0% and 28.6% from 1970 to 2000. Therefore, the riverine impact may be underestimated.

We have added this to the discussion Section 4.3.

*Figure 4: It's a bit concerning to me that considering riverine inputs lead to a sink of carbon in the ocean. It is relatively well acknowledged that river inputs are thought to cause a source of carbon (e.g., Regnier et al., 2013; Resplandy et al., 2018). The reason for this is that carbon to nutrient ratio of the (bio-available) terrestrial inputs is larger than the Redfield ratio. I guess the fact that most particulate P and N is thrown in as dissolved inorganic species might be the explanation for this. How is the alkalinity to DIC ratio of riverine inputs constrained? Either way, either explain the reason for this or I would consider not discussing the $CO_2$ flux for the "unperturbed" river simulation.*

Firstly, the riverine DIC to alkalinity ratio that we have applied is 1:1.

Our model shows the ocean as a weak carbon sink when including riverine inputs (in FIX and RUN), which can be partly due to our model assumption that the excess dissolved organic and particulate matter immediately remineralized into bioavailable inorganic nutrients from rivers. This is because we have only one dissolved organic pool (DOM) and one particulate organic pool (DET) in the model, and the Redfield ratio needs to be kept. We have assessed the impact of this approach on our results (please see the response to the comments in the Method section). However, the riverine inputs

cause a C source when we employ the GNS scenarios. Please also see the reply to the next comment.

We have discussed this point in Section 4.1 and 4.3.

- *L305 "Our experiments show that riverine nutrient inputs have a dominant role over the organic matter inputs in FIX, enhancing CO2 uptake along continental margins via sustaining PP in both historical and future time periods." This is however purely a consequence of the ratio of (bio-available) nutrients to organic matter that is added to the ocean, which as mentioned, I don't think is completely correct from a process-based perspective. In fact, I think most river-dominated shelves show C outgassing, see regional CO2 fluxes from Chen and Borges (2009) or regional-based studies.*

Thank you for your comment. We also note that it has been shown that the riverine inorganic nutrients input to the ocean enhances marine primary production, which induces $CO_2$ uptake (Tyrrell, 1999). The riverine organic nutrients (including DON, DOP, PON, POP) undergo remineralization in the ocean and release corresponding inorganic nutrients, which will at the end support primary production and $CO_2$ uptake. On the other hand, riverine input of inorganic and organic carbon (DOC and POC) release DIC, causing $CO_2$ outgassing. It is the competition between the riverine (inorganic and organic) nutrients input induced $CO_2$ uptake and the riverine carbon input induced $CO_2$ outgassing, which determines whether the shelf is a carbon sink or a carbon source. However, the composition of the riverine organic matter (i.e., carbon to nutrient ratio) and the degradation timescales which are the key factors, have been debated over the last 3 decades (Ittekkot, 1988; Hedges et al., 1997; Cai, 2010; Bianchi, 2011; Blair and Aller, 2011; Lalonde et al., 2014; Galy et al., 2015). It is generally agreed that the riverine organic carbon to nutrient ratio is high (e.g., C:P weight ratio larger than 700, Seitzinger et al., 2010) and the degradation and resuspension rates in shallow shelf seas/sediment are higher than the open ocean (Krumins et al., 2013). It suggests that at shallow and near-shore areas the riverine carbon input usually results in a $CO_2$ source for the atmosphere, while at deeper outer shelf areas the riverine nutrient input causes primary production increase and a $CO_2$ sink, and the magnitudes of the carbon source and sink on the continental shelves almost compensate each other. This phenomenon has been discussed by both measurement-based studies (Borges and Frankignoulle, 2005; Chen and Borges, 2009) and modelling studies (Lacroix et al., 2020). However, in our model, the spatial resolution is not fine enough to differentiate the near-shore and outer shelf physical and biogeochemical processes. This partly contributes to comparable $CO_2$ outgassing near shore (due to riverine C) and $CO_2$ ingassing on outer shelves (due to riverine inorganic and organic nutrients input), leading to a globally weak integrated C sink on the continental margins in FIX and RUN experiments for both historical and future time periods. On the other hand, in GNS the riverine inputs reduce globally integrated C uptake for both historical and future time periods. Simulations with high-resolution global or regional models with more realistic representation of shelf processes are required to accurately assess impact of riverine inputs on carbon cycling in the coastal ocean. In addition, future studies should also consider evaluating the sensitivity of $CO_2$ fluxes to the higher riverine carbon to nutrient ratio.

- *L337 "…do not transfer to large uncertainties in future global marine biogeochemistry projections in NorESM." Yes, but if you would have taken uncertainties related to the*

*coastal ocean into account, through e.g. sensitivity analysis of sediment degradation, I wonder if the conclusions would be different here, I would assume so.*

We changed the sentence to "A large range of the riverine inputs in GNS, e.g., temporal changes in DIN fluxes across scenarios ranging 24.8-63.0% of the annual flux in FIX, do not transfer to large uncertainties in future projections of global marine primary production in our model, which can be primarily attributed to unresolved shelf processes due to coarse model resolution."

*Minor edits*

- *L22 "and thus lessen the projected future decline in PP by up to 0.6 PgC yr-1 22 (27.3%) globally depending on the riverine configuration." -> , globally,*

We have changed it accordingly.

- *L55 "Taking the advantage of the latest improvement of global" Remove "the".*

We have changed it accordingly.

- *L171 "By comparing FIX versus REF…" Add comma here.*

We have changed it accordingly.

- *L332 ". Therefore, it is worth exploring the merits of using GNS in future projections of marine biogeochemistry." Not really sure what is meant here.*

This sentence was to express that using future scenarios of transient riverine fluxes can be explored by future modelling studies on projection of marine biogeochemistry, especially on high-resolution regional scales. We removed this sentence in the revised version, since it has been explained in the end of this paragraph.

**Reference**

[revised manuscript text omitted]

---

## Referee Report (RR1)

**Major Comments:**

In their manuscript, Gao et al. investigate the role of contemporary river fluxes for alleviating bias of the ocean biogeochemistry model, as well as the impacts of potential future changes in riverine fluxes for ocean biogeochemistry. The authors find improved model performance on the global continental shelf after introducing riverine fluxes, as well as counteracting effects of individual impacts of increased carbon and nutrients inputs to the ocean. As mentioned in the previous reviews, the material is new and the analysis was well performed, although the chosen setup is still clearly very limited. As one of the authors of one of the two rather critical reviews, I believe that the manuscript is now immensely improved in other aspects over the previous submitted version:

- The introduction is now a very nice review of state-of-art knowledge and implementations of riverine exports in the ocean (for which remain many uncertainties)
- The methods and results are better structured and clear to follow
- The interpretation and discussion of results are now more complete and clearer. In particular, a discussion on model limitations due to simplified relationships of riverine stoichiometries, degradation and shelf circulation are all now included. The authors included an well-thought back-of-the envelope calculation on the impacts of too low shelf mineralization rates on the change in primary productivity.

Overall, it was a very enjoyable read. I thus approve accepting the manuscript for publication and have only minor specific comments and edits.

**Specific Comments:**

**L28** *"while in the more plausible riverine configurations the river inputs cause a net C source of ~0.1 ± 0.03 Pg C yr-1 „*. In my opinion, this is quite an interesting result being that the effect of increased riverine carbon is stronger than increased nutrient inputs for the future projections. For the historical perturbations, it is usually assumed the nutrients are the dominant component (e.g. as the authors mention in the Lacroix et al., 2021) because of the large relative change in the past. But this indeed might not hold for the future, and there is very little work on the impacts of changing riverine C fluxes. I however leave it to the authors whether they would like to perhaps underline this more strongly.

Edits

L14 *"So far, this contribution is represented in the state-of-the-art Earth system models with limited effort."* Sounds a bit awkward.

L243 maybe formulate like this: *"…are aggregated within catchment basins defined by the NEWS 2 study for every river."*?

L245 *"..up to…"*

L251-253 *"First, we calculate the riverine organic P-N-C ratios for both dissolved and particulate forms, then add the least abundant species (scaled by the Redfield ratio) to the DOM and DET pools, respectively. „* I understand what is meant, I wonder however if this could be formulated more clearly however.

L384 if -> when

L447 and no riverine C input -> no varying riverine C input

L487 *"Given that the riverine nutrient and carbon inputs account for only a small proportion of the total amount of nutrients and carbon in the euphotic zone of the ocean"* I assume this refers to "yearly inputs", but the sentence also feels out of place and doesn't relate to what comes next.

---

## Author Response (AR2)

Reply to Anonymous referee #1

**We are grateful for the comments and edits of the anonymous reviewer who invested time for the review of our manuscript. The reviewer's text is reported in Italic and our responses in roman.**

*Suggestions for revision or reasons for rejection (will be published if the paper is accepted for final publication)*

*The authors did an amazing job in taking all the comments into account during the very thorough review. Thank you very much for working in all comments.*

*From my perspective, the paper is now almost ready for publications if the following comments would be implemented:*

*Larger comments:*

*I am still confused that the estimated effect of riverine influx and burial is an outgassing of 0.65 Pg C yr-1 (Regnier et al., 2022) but your model results suggest something different. This difference needs to be acknowledged and discussed somewhere. I understand that this is typical for most ocean biogeochemical models (riverine influx = burial) but it should not be the case if rivers and sedimentation are simulated as realistically as possible. Can you find a reason why adding the rivers does not lead to an outgassing (FIX-REF is 0.1 Pg C yr-1 but should be -0.65 Pg C yr-1).*

1) Thank you very much for pointing it out. One of the reasons for the discrepancy is the different definition of the extent of "ocean" used in the calculation of the air-sea $CO_2$ fluxes in both studies. In the study by Regnier et al. (2022), the 0.65 Pg C yr$^{-1}$ outgassing was calculated for open ocean (lateral flux from continental shelf water minus burial in open ocean sediment; 0.80-0.15=0.65). The carbon budget for estuaries, tidal wetlands and continental shelf water was calculated as input to the ocean. However, in our model, the air-sea $CO_2$ fluxes were calculated over the broader ocean areas, which implicitly include estuaries and continental shelves. Therefore, the carbon (C) uptake from estuaries and continental shelves in our study would partly balance the outgassing in the open ocean or elsewhere. If we take the numbers from their study and assume the burial is the same, our calculation of air-sea $CO_2$ fluxes would be 0.65-0.2-0.1=0.35 Pg C yr$^{-1}$, where 0.65 is outgassing from open ocean, 0.2 and 0.1 are C uptake from estuaries and continental shelf water. Therefore, the ocean outgassing would be 0.35 Pg C yr$^{-1}$ let alone the ±0.30 Pg C yr$^{-1}$ uncertainty.

2) If we understood correctly, by mentioning FIX-REF=0.1 Pg C yr$^{-1}$, the reviewer refers to the riverine impact on global C uptake during 2003–2012 (Table B2, 0.09±0.01 Pg C yr$^{-1}$). Our model results indeed suggest a weak C sink due to the riverine input at the level of 1970's. Note that the above mentioned 0.65±0.30 Pg C yr$^{-1}$ ocean outgassing was calculated for pre-industrial conditions. Although the anthropogenic perturbation of riverine load of C is uncertain, the riverine nutrients load in contemporary era is increased significantly. Riverine loads of phosphorus (P) and nitrogen (N) increased from the pre-industry values of 2 Tg P yr$^{-1}$ and 20 Tg N yr$^{-1}$ to 1970's level of 7.6 Tg P yr$^{-1}$ and 37 Tg N yr$^{-1}$, respectively (Beusen et al., 2016; Green et al., 2004; Seitzinger et al., 2010).

This can be the second reason why our results indicate an ocean C sink for contemporary time, due to the enhanced biological C uptake.

3) The third reason might be, as we mentioned in the manuscript, that our model overestimates the conversion of organic nutrients to dissolved inorganic nutrients, which probably leads to overestimate of biological C uptake.

*When I look at figure 5, it seems that the GN scenario has no effect on NPP globally and in the Arctic Ocean. Maybe consider mentioning that the scenario has almost no effect? I think it would be easier to understand and read the paper if you show only one scenario in the main part and move the other ones to an appendix. In general, the Discussion reads not very well. I am not sure what the information is that I should take away from this. Maybe try to restructure the paper as follows: Traditional simulation without any riverine input (REF), changes in PP and air-sea CO2 flux from climate change with fixed rivers (FIX), and changes in PP and air-sea CO2 fluxes due to changes in rivers (one GN scenario). I think this would be a clearer message that would be easier to transfer to the reader.*

1) In Figure 5a (see also Table B1) it shows that in GNS runs the projected decline in global PP is lessened by $0.66\pm0.02$ Pg C yr$^{-1}$ (29.5%) from $-2.24\pm0.37$ Pg C yr$^{-1}$ (REF) to $-1.57$ $\pm0.38$ Pg C yr$^{-1}$ (mean GNS). We mentioned that also in the Abstract. In the Arctic, the effect of GNS have the same magnitude as the FIX and RUN (Figure 5b). We agree with the reviewer that there is almost no difference in the effect among the four GN scenarios when we consider the global integrated PP or C uptake, and the spatial difference among the four scenarios is also marginal. We consider that this warrants further investigation with higher resolution models, which can resolve the effects of riverine input at hotspots in different scenarios. Since we did not focus on discussing the difference among the four scenarios, we think it is not necessary to separate them.

2) We have restructured the discussion in the following way: as reviewer suggested, we firstly discuss projection of PP and C uptake under climate change only (REF), followed by discussion on projected change due to riverine input (in the order of FIX, RUN and GNS).

*Please consider adding a discussion about what would be needed to improve modeling of the rivers following up on your study and other studies. Do you get insights why models do not find the same results as the one derived from observations (Regnier et al., 2022), i.e., a riverine carbon outgassing of 0.65 Pg C yr-1. When I look at Figure 4, the rivers seem to be a very minor problem compared to the rivers.*

Thanks for the suggestion. We have added a paragraph at the end of Conclusion.

1) Better resolve shelf processes with higher model resolution, to have more realistic remineralization rate for riverine organic matter in the coastal water and shelf sediment, as well as lateral transport

2) Better constrain the riverine carbon to nutrient ratios

3) Explore future scenarios of riverine input to the ocean and their impact on ocean PP and C uptake, especially on regional scales

*Can you explain the large overestimation of NPP in the Southern Ocean in NorESM and the large underestimation in the western Pacific and Indian Ocean? That might be very helpful to underestimate the general underestimation. Maybe nothing is exported north out of the Southern Ocean (Sarmiento et al., 2004)?*

Biases in physical and biogeochemical processes in NorESM contribute to the regional PP model-data discrepancies, and it is challenging to attribute specific processes to these biases. Nevertheless, two factors have been recognized to contribute to the PP bias in the Southern Ocean. In NorESM1 the large PP in the Southern Ocean can be attributed to a too weak top-down control, leading to large spring blooms in phytoplankton at high latitudes. With a re-tuning of the ecosystem parameterization this bias has been reduced in newer model version (Schwinger et al., 2016). Moreover, the bias high PP in the Southern Ocean can also be due to the strong winter mixing, which upwells too much nutrients for the proceeding spring bloom, based on the improvements simulated in the latest version of NorESM (Tjiputra et al., 2020). Otherwise, PP is relatively low because the isopycnic model might have a too low vertical diffusivity providing too little new nutrient to the euphotic zone at lower latitudes. This low bias in the nitrate-limited western Pacific and Indian Ocean has been alleviated through improvements in the nitrogen fixation and productivity parameterizations. Other processes such as the equatorward nutrient advection from the Southern Ocean (Sarmiento et al., 2004) could also play a role.

*Minor:*

*Line 19: simulated contemporary spatial distribution of what? A word seems to be missing.*

We added "of annual mean PP and air-sea $CO_2$ fluxes".

*Line 24: The best estimate and the uncertainty do not have the same number of digits behind the comma.*

We changed 0.7±0.02 to 0.66±0.02, as well as in Line 29, changed 0.1±0.03 to 0.11±0.03.

*Line 27: I am not sure what the therefore wants to say here. I cannot see the link but it might be my fault.*

We agree, "therefore" has been deleted.

*Lines 37-42: Maybe consider discussing Regnier et al. (2022) here.*

We have added "A recent study on global carbon cycle has emphasized the importance of the carbon transport through the land-to-ocean aquatic continuum (Regnier et al., 2022)."

*Line 83: What kind of model? Some more details would be good here.*

The name of the model (the Integrated Model to Assess the Global Environment-Global Nutrient Model (IMAGE-GNM)) has been added.

*Line 148: Is 'state-of-the-art' still the right wording? It is surely one of the best ESMs available but given that a new version is available, I would reconsider this wording. But it doesn't really matter.*

"State-of-the-art" has been deleted.

*Line 175 and afterwards: Given that the model is not exactly HAMOCC5 as you explained in the responses, I would not call it HAMOCC5 here and later. A possible alternative would be ocean biogeochemical model component.*

We have renamed it as HAMOCC$_{NorESM1}$ and edited it through the whole manuscript.

*Line 192: Please consider a quantitative assessment. Good agreement is very subjective.*

We have added "The simulated global annual mean PP is 40.1 Pg C yr$^{-1}$ during 2003–2012, which is lower than the satellite-based model estimates, ranging from 55 to 61 Pg C yr$^{-1}$."

*Lines 207-209: You might want to consider adding here also missing terrigenous input, which is at the centre of your study. Another important point would be lateral influx from the Atlantic and Pacific Ocean (Torres-Valdés et al. 2013), which is often underestimates in models with relatively coarse resolution (Terhaar et al., 2019) as ESMs.*

1) We have added "Additionally, lack of adequate representation of riverine input in some ESMs can also lead to underestimate of PP, since around one third of current Arctic marine PP is sustained by terrigenous nutrient input (Terhaar el al., 2021)."
2) We have acknowledged the underestimate of lateral influx (from coast to open ocean) in the first paragraph of Section 4.4. However, in Torres-Valdés et al. (2013)'s paper, they showed that there are statistically robust net silicate and phosphate exports out of the Arctic, while the net nitrate flux is zero. There is no net influx of nutrients to the Arctic from the Atlantic and Pacific, therefore we cannot use it as an argument for the model underestimate of PP.

*Lines 188-215: It would be interesting to also see an evaluation of nutrients and alkalinity in the model compared to observations.*

The simulated alkalinity, phosphate, nitrate and silicic acid have been evaluated in previous work by Tjiputra et al. (2013) and they have been compared between NorESM1 and NorESM2 in a more recent work by Tjiputra et al. (2020). We have added this information in the manuscript.

*Lines 342: Projections are always about the future, so I think future is not needed here.*

We have deleted it.

*Line 417-418: Consider citing Vancoppenelle et al. (2013) here.*

Thanks. It has been added.

*Lines 521-524: It seems a bit strange to compare model results to model results. There are many observational studies. You might want to consider discussion observational estimates of the lability of the riverine organic matter. Aumont et al (2001) showed the effect of lability in models and a comparison with Lacroix et al. (2021) suggests that the lability of organic matter might be on the high side. Adding observational studies here would probably help.*

We have changed to observation based estimate as "Given that the proportion of the direct remineralized organic N (27.8%, see the calculation above) in our model is comparable to or lower than the reported values by field studies (~38.8% of DON decomposed during transition from Arctic rivers to coastal ocean), which indicates that the bias on enhanced PP is likely less than 33.3%."

*Figure 3: Consider changing the colormap in a).*

We have changed the colormap.

*Aumont, O., Orr, J. C., Monfray, P., Ludwig, W., Amiotte-Suchet, P., and Probst, J.-L. (2001), Riverine-driven interhemispheric transport of carbon, Global Biogeochem. Cycles, 15( 2), 393– 405, doi:10.1029/1999GB001238.*

*Regnier, P., Resplandy, L., Najjar, R.G. et al. The land-to-ocean loops of the global carbon cycle. Nature 603, 401–410 (2022). https://doi.org/10.1038/s41586-021-04339-9*

*Sarmiento, J., Gruber, N., Brzezinski, M. et al. High-latitude controls of thermocline nutrients and low latitude biological productivity. Nature 427, 56–60 (2004). https://doi.org/10.1038/nature02127*

*Terhaar, J., Orr, J. C., Gehlen, M., Ethé, C., and Bopp, L.: Model constraints on the anthropogenic carbon budget of the Arctic Ocean, Biogeosciences, 16, 2343–2367, https://doi.org/10.5194/bg-16-2343-2019, 2019*

*Torres-Valdés, S., T. Tsubouchi, S. Bacon, A. C. Naveira-Garabato, R. Sanders, F. A. McLaughlin, B. Petrie,G. Kattner, K. Azetsu-Scott, and T. E. Whitledge (2013), Export of nutrients from the Arctic Ocean,J. Geophys. Res.Oceans,118, 1625–1644, doi:10.1002/jgrc.20063*
* * *
Beusen, A. H. W., Bouwman, A. F., Van Beek, L. P. H., Mogollón, J. M., and Middelburg, J. J.: Global riverine N and P transport to ocean increased during the 20th century despite increased retention along the aquatic continuum, Biogeosciences, 13, 2441-2451, 10.5194/bg-13-2441-2016, 2016.

Green, P. A., Vörösmarty, C. J., Meybeck, M., Galloway, J. N., Peterson, B. J., and Boyer, E. W.: Pre-industrial and contemporary fluxes of nitrogen through rivers: a global assessment based on typology, Biogeochemistry, 68, 71-105, 2004.

Schwinger, J., Goris, N., Tjiputra, J. F., Kriest, I., Bentsen, M., Bethke, I., Ilicak, M., Assmann, K. M., and Heinze, C.: Evaluation of NorESM-OC (versions 1 and 1.2), the ocean carbon-cycle stand-alone configuration of the Norwegian Earth System Model (NorESM1), Geosci. Model Dev., 9, 2589-2622, 10.5194/gmd-9-2589-2016, 2016.

Seitzinger, S. P., Mayorga, E., Bouwman, A. F., Kroeze, C., Beusen, A. H. W., Billen, G., Van Drecht, G., Dumont, E., Fekete, B. M., Garnier, J., and Harrison, J. A.: Global river nutrient export: A scenario analysis of past and future trends, Global Biogeochemical Cycles, 24, https://doi.org/10.1029/2009GB003587, 2010.

Terhaar, J., Lauerwald, R., Regnier, P., Gruber, N., and Bopp, L.: Around one third of current Arctic Ocean primary production sustained by rivers and coastal erosion, Nature Communications, 12, 169, 10.1038/s41467-020-20470-z, 2021.

Tjiputra, J. F., Roelandt, C., Bentsen, M., Lawrence, D. M., Lorentzen, T., Schwinger, J., Seland, Ø., and Heinze, C.: Evaluation of the carbon cycle components in the Norwegian Earth System Model (NorESM), Geosci. Model Dev., 6, 301-325, 10.5194/gmd-6-301-2013, 2013.

Tjiputra, J. F., Schwinger, J., Bentsen, M., Morée, A. L., Gao, S., Bethke, I., Heinze, C., Goris, N., Gupta, A., He, Y. C., Olivié, D., Seland, Ø., and Schulz, M.: Ocean biogeochemistry in the Norwegian Earth System Model version 2 (NorESM2), Geosci. Model Dev., 13, 2393-2431, 10.5194/gmd-13-2393-2020, 2020.

Reply to Referee #2 Fabrice Lacroix

**We are grateful for the comments and edits of Fabrice Lacroix who invested time for the review of our manuscript. The reviewer's text is reported in Italic and our responses in roman.**

*Suggestions for revision or reasons for rejection (will be published if the paper is accepted for final publication)*

*The authors have gone great lengths to improve their manuscript. While the chosen setup is still limited, the review of literature, as well as the analysis and discussion of the results is well done. I therefore suggest accepting the manuscript with only few technical corrections.*

**Major Comments:**

*In their manuscript, Gao et al. investigate the role of contemporary river fluxes for alleviating bias of the ocean biogeochemistry model, as well as the impacts of potential future changes in riverine fluxes for ocean biogeochemistry. The authors find improved model performance on the global continental shelf after introducing riverine fluxes, as well as counteracting effects of individual impacts of increased carbon and nutrients inputs to the ocean. As mentioned in the previous reviews, the material is new and the analysis was well performed, although the chosen setup is still clearly very limited. As one of the authors of one of the two rather critical reviews, I believe that the manuscript is now immensely improved in other aspects over the previous submitted version:*

- *The introduction is now a very nice review of state-of-art knowledge and implementations of riverine exports in the ocean (for which remain many uncertainties)*
- *The methods and results are better structured and clear to follow*
- *The interpretation and discussion of results are now more complete and clearer. In particular, a discussion on model limitations due to simplified relationships of riverine stoichiometries, degradation and shelf circulation are all now included. The authors included an well-thought back-of-the envelope calculation on the impacts of too low shelf mineralization rates on the change in primary productivity.*

*Overall, it was a very enjoyable read. I thus approve accepting the manuscript for publication and have only minor specific comments and edits.*

**Specific Comments:**

*L28 "while in the more plausible riverine configurations the river inputs cause a net C source of ~0.1 ± 0.03 Pg C yr-1 „. In my opinion, this is quite an interesting result being that the effect of increased riverine carbon is stronger than increased nutrient inputs for the future projections. For the historical perturbations, it is usually assumed the nutrients are the dominant component (e.g. as the authors mention in the Lacroix et al., 2021) because of the large relative change in the past. But this indeed might not hold for the future, and there is very little work on the impacts of changing riverine C fluxes. I however leave it to the authors whether they would like to perhaps underline this more strongly.*

Thank you very much for pointing it out. We have added one sentence in the Abstract: "It implies that the effect of increased riverine C may be larger than the effect of nutrient inputs in the future on the projections of ocean C uptake, while in historical period increased nutrient inputs are considered as the largest driver."

*Edits*

*L14 "So far, this contribution is represented in the state-of-the-art Earth system models with limited effort." Sounds a bit awkward.*

We have changed this sentence to "So far, this process has not been fully represented and evaluated in the state-of-the-art Earth system models."

*L243 maybe formulate like this: "...are aggregated within catchment basins defined by the NEWS 2 study for every river."?*

Changed accordingly.

*L245 "..up to..."*

Changed accordingly.

*L251-253 "First, we calculate the riverine organic P-N-C ratios for both dissolved and particulate forms, then add the least abundant species (scaled by the Redfield ratio) to the DOM and DET pools, respectively. „ I understand what is meant, I wonder however if this could be formulated more clearly however.*

We have added the following equations to make it more understandable.

$DOM_{riv}=min\ (DOP, DON/16, DOC/122)$

$DET_{riv}=min\ (POP, PON/16, POC/122)$

*L384 if -> when*

Changed accordingly.

*L447 and no riverine C input -> no varying riverine C input*

This part of the sentence has been deleted due to change in paragraph structure.

*L487 "Given that the riverine nutrient and carbon inputs account for only a small proportion of the total amount of nutrients and carbon in the euphotic zone of the ocean" I assume this refers to "yearly inputs", but the sentence also feels out of place and doesn't relate to what comes next.*

Thanks for pointing it out. We have deleted this part of the sentence and modified it as "We acknowledge several limitations of our study, particularly related to the resolution and complexity of our model."

---

## Author Response (AR3)

**Author response**

- In this submission, we have revised the color schemes of the Figure 2 as required.
- We have followed the suggestion to publish our datasets and model code, and included the citation in Code and data availability and added the reference in the list.